

# A systematic $1/c$-expansion of form factor sums for dynamical correlations in the Lieb-Liniger model

Etienne Granet⋆ and Fabian H. L. Essler

The Rudolf Peierls Centre for Theoretical Physics, Oxford University, Oxford OX1 3PU, UK

⋆ etienne.granet@physics.ox.ac.uk

## Abstract

We introduce a framework for calculating dynamical correlations in the Lieb-Liniger model in arbitrary energy eigenstates and for all space and time, that combines a Lehmann representation with a $1/c$ expansion. The $n^{\text{th}}$ term of the expansion is of order $1/c^n$ and takes into account all $\lfloor \frac{n}{2} \rfloor + 1$ particle-hole excitations over the averaging eigenstate. Importantly, in contrast to a "bare" $1/c$ expansion it is uniform in space and time. The framework is based on a method for taking the thermodynamic limit of sums of form factors that exhibit non integrable singularities. We expect our framework to be applicable to any local operator. We determine the first three terms of this expansion and obtain an explicit expression for the density-density dynamical correlations and the dynamical structure factor at order $1/c^2$. We apply these to finite-temperature equilibrium states and non-equilibrium steady states after quantum quenches. We recover predictions of (nonlinear) Luttinger liquid theory and generalized hydrodynamics in the appropriate limits, and are able to compute sub-leading corrections to these.



# 1 Introduction

The Lieb-Liniger model [1] is a key paradigm of integrable many-particle systems [3]. Moreover, it is directly relevant to a range of cold atom experiments both in and out of equilibrium,

see e.g. [4–9]. While the excitation spectrum at zero temperature [10] and thermodynamic properties [11] have been known for a long time, the exact solution does not provide easy access to correlations functions as these encode more detailed information about the exact energy eigenstates. An exception is the case of impenetrable bosons [12–25], which can be mapped onto non-interacting fermions. In absence of full analytic solutions valuable insights on the large space and time asymptotic behaviours of correlation functions at zero and low temperatures were gained by combining exact results on spectral properties obtained from the Bethe Ansatz with with conformal field theory (CFT) [26,27] and Luttinger liquid theory [28,29] and its recent extensions [30–34,36]. The last two decades then witnessed remarkable progress in the computation of zero temperature dynamical correlation functions by expressing them in terms of spectral representations over the energy eigenstates of the model. On the one hand it became possible to numerically evaluate the spectral sums to very high precision for large, finite systems [37,38]. On the other hand remarkable analytic progress led to a fairly complete understanding of the asymptotic behaviour at late times and large distances [39, 41, 42]. In contrast to ground state case and non-interacting theories [43–57] progress on determining finite temperature correlators in interacting integrable models has been much more limited. The basic idea in interacting integrable models has been to again use spectral representations and sum over "the most relevant" states, both for equal time [58–64] and dynamical correlators [65–76]. These summations can again be approached either numerically or analytically.

The numerical approach focuses on finite systems of about a hundred particles in the case of the Bose gas and works in momentum space, i.e. considers the dynamical structure factor as a function of frequency and momentum [71]. It then sums the dominant contributions to the dynamical structure factor in the sense that the f-sum rule is satisfied to a very high accuracy.

To make analytical progress it is essential to identify the classes of states that give the dominant contributions in a given range of frequencies and momenta or space and time [39]. Known results suggest that in interacting theories this generally requires the summation over an infinite number of states. Firstly, the large space and time asymptotics of zero temperature dynamical correlators in interacting models has been shown to be determined by an arbitrary number of (soft) particle-hole excitations over the ground state around the Fermi points and the saddle points of the dispersions of elementary excitations [39,77]. Secondly, it has been shown that the asymptotic behaviours of dynamical correlations of semi-local operators in thermal and other finite entropy states involves an arbitrary number of (soft) particle-hole excitations [78] over the macro state of interest. Truncating this sum to a finite number of particle-hole excitations leads to a result that diverges in time. In the zero temperature case it has been shown that it is possible to take the thermodynamic limit of (partial) spectral sums and obtain a representation in terms of (dressed) excitations in the thermodynamic limit [39, 77]. An analogous result for the finite temperature/entropy case would be highly desirable, but is not known at present. In Refs [73,75,76] such an expansion in terms of thermodynamic particle-hole excitations was conjectured. It is based an phenomenological assumptions on how partial sums over states in the finite volume combine into thermodynamic form factors. It also exhibits singularities, whose regularization is not presently known.

Given this state of affairs it is highly desirable to obtain explicit results through *ab initio* calculations that do not require any assumptions, i.e. carrying out the spectral sum in a finite volume and then taking the thermodynamic limit exactly. In order to make progress in this direction we consider the spectral sum in the framework of an expansion in the inverse interaction strength $c^{-1}$ around the impenetrable limit. Strong coupling expansions have previously been used at zero temperature and for static correlators at finite temperatures [79–82]. More recently the $1/c$ contribution for the finite temperature dynamical density-density correlation function was determined in [72]. This contribution has a particularly simple structure similar to that of the impenetrable limit, that does not carry over to the next orders, and as a conse-

quence until now it has been unclear how to determine higher orders in this expansion. In the following we develop a method for calculating the higher orders of this expansion and apply it to obtain the contribution to the dynamical density-density correlator at order $c^{-2}$. The general idea of the $1/c$ expansion, and more generally of strong coupling expansions in integrable models, is as follows. A consequence of integrability is that $N$-particle energy eigenstates in a finite volume can be labelled by $N$ rapidity variables

$$|\boldsymbol{\lambda}\rangle = |\lambda_1, \ldots, \lambda_N\rangle. \tag{1}$$

These rapidities are in a one-to-one correspondence with sets of (half-odd) integers $\{I_j\}$ through the quantization conditions in the finite volume

$$\{\lambda_1, \ldots, \lambda_N\} \longleftrightarrow \{I_1, \ldots, I_N\}. \tag{2}$$

The energy and momentum of these states are given by

$$E(\boldsymbol{\lambda}) = \sum_{j=1}^{N} \epsilon(\lambda_j), \quad P(\boldsymbol{\lambda}) = \sum_{j=1}^{N} p(\lambda_j), \tag{3}$$

where $\epsilon(\lambda)$ and $p(\lambda)$ parametrize the energy and momentum of a single-particle excitation over the vacuum (reference) state. For the Bose gas we have $\epsilon(\lambda) = \lambda^2$ and $p(\lambda) = \lambda$. Two-point correlation functions of a local operator $\mathcal{O}(x)$ in a given energy eigenstate $|\boldsymbol{\lambda}\rangle$ thus have spectral representations of the form

$$\frac{\langle\boldsymbol{\lambda}|\mathcal{O}(x,t)\mathcal{O}^\dagger(0,0)|\boldsymbol{\lambda}\rangle}{\langle\boldsymbol{\lambda}|\boldsymbol{\lambda}\rangle} = \sum_{M=0}^{\infty} \sum_{\{\mu_1,\ldots,\mu_M\}} \frac{|\langle\boldsymbol{\lambda}|\mathcal{O}(0,0)|\boldsymbol{\mu}\rangle|^2}{\langle\boldsymbol{\lambda}|\boldsymbol{\lambda}\rangle\langle\boldsymbol{\mu}|\boldsymbol{\mu}\rangle} e^{i\left(E(\boldsymbol{\lambda})-E(\boldsymbol{\mu})\right)t - i\left(P(\boldsymbol{\lambda})-P(\boldsymbol{\mu})\right)x}, \tag{4}$$

where the first sum runs over the particle number and the second over all $M$-particle energy eigenstates. The matrix elements

$$F_{\mathcal{O}}(\boldsymbol{\lambda}, \boldsymbol{\mu}) = \frac{\langle\boldsymbol{\lambda}|\mathcal{O}(0,0)|\boldsymbol{\mu}\rangle}{\sqrt{\langle\boldsymbol{\lambda}|\boldsymbol{\lambda}\rangle\langle\boldsymbol{\mu}|\boldsymbol{\mu}\rangle}}, \tag{5}$$

are also known as *form factors* and, as we will see, admit a $1/c$-expansion

$$F_{\mathcal{O}}(\boldsymbol{\lambda}, \boldsymbol{\mu}) = \sum_{n=0}^{\infty} \frac{F_{\mathcal{O},n}(\boldsymbol{I}, \boldsymbol{J})}{c^n}, \tag{6}$$

where $\boldsymbol{I} = \{I_1, \ldots, I_N\}$ and $\boldsymbol{J} = \{J_1, \ldots, J_M\}$ are the (half-odd) integers corresponding to the rapidities $\lambda_1, \ldots, \lambda_N$ and $\mu_1, \ldots, \mu_M$ respectively. Similarly $E(\boldsymbol{\lambda})$ and $P(\boldsymbol{\lambda})$ can be expanded in powers of $c^{-1}$

$$E(\boldsymbol{\lambda}) = \sum_{n=0}^{\infty} \frac{E_n(\boldsymbol{I})}{c^n}, \quad P(\boldsymbol{\lambda}) = \sum_{n=0}^{\infty} \frac{P_n(\boldsymbol{I})}{c^n}. \tag{7}$$

Denoting the truncation of the sums to order $\mathcal{O}(c^{-j})$ by $F_{\mathcal{O}}^{(j)}(\boldsymbol{I}, \boldsymbol{J})$, $E^{(j)}(\boldsymbol{I})$ and $P^{(j)}(\boldsymbol{I})$ respectively, the $1/c$-expansion at order $\mathcal{O}(c^{-j})$ is defined as

$$\sum_{M=0}^{\infty} \sum_{\{\mu_1,\ldots,\mu_M\}} |F_{\mathcal{O}}^{(j)}(\boldsymbol{I}, \boldsymbol{J})|^2 e^{i\left(E^{(j)}(\boldsymbol{I})-E^{(j)}(\boldsymbol{J})\right)t - i\left(P^{(j)}(\boldsymbol{I})-P^{(j)}(\boldsymbol{J})\right)x}. \tag{8}$$

We stress that the expansion sums certain $1/c$ contributions to all orders by virtue of the fact that although the (exactly known) energies and momenta are expanded inside the exponentials, the exponentials are *not* expanded in $1/c$. In this sense the expansion is non-perturbative,

and in fact rather different from more standard (diagrammatic) approaches pursued in [83]. As discussed in detail below (8) is in fact both a $1/c$ expansion and an expansion *in terms of number of particle-hole excitations*. At order $n$ in the expansion (i) only excitations that involve at most $\lfloor \frac{n}{2} \rfloor + 1$ particle-hole pairs contribute, and (ii) all terms up to $\mathcal{O}(c^{-n})$ contribute. Importantly, this "mixed" expansion has a well-defined thermodynamic limit and is uniform in space and time. This is in contrast to both the bare $1/c$ expansion that is non-uniform, and the bare expansion in the number of particle-hole excitations that is divergent in the thermodynamic limit.

Expectation values of the form (4) are relevant in two contexts.

1. By working in a micro-canonical ensemble dynamical response functions at finite temperature can be cast in this form. In the following we will use this to determine the finite temperature dynamical structure factor in the Lieb-Liniger model.

2. At late times after quantum quenches local observables relax to non-thermal stationary values [93–96]. It follows from the quench action approach [91, 92] to quantum quenches that expectation values in the stationary state in fact involve non-thermal energy eigenstates at finite energy densities. This has been used to study the stationary behaviour of certain one-point functions after (particular) quantum quenches [97–99,101]. A natural extension is then to consider linear response functions in such steady states [102, 103]. These can be expressed in the form (4), where $|\lambda\rangle$ corresponds to the non-equilibrium steady state relevant to the quench of interest.

In the following we will consider both these cases and evaluate (4) for the density operator and general $|\lambda\rangle$.

A brief summary of some of our key technical results is as follows. We show that the $1/c$-expansion corresponds to an expansion in the number of particle-hole excitations. This leads to a dramatic reduction in the complexity of the spectral sum that needs to be carried out. Interestingly, the contributions of one particle-hole and two particle-hole excitations are individually *divergent* in the infinite volume limit $L \to \infty$. Moreover they individually depend on details of the "averaging state" $|\lambda\rangle$ beyond the root distribution function in the thermodynamic limit. Crucially, their sum is not divergent and is independent of the choice of representative state $|\lambda\rangle$, and is well-defined.

The manuscript is organized as follows. In Section 2 we introduce the Lieb-Liniger model and recall the key elements of its Bethe Ansatz solution. In Section 3 we report some important intermediate results on the thermodynamic limit of expressions computed within the Bethe Ansatz. In Section 4 we discuss the $1/c$-expansion up to and including $\mathcal{O}(c^{-2})$ of the Bethe Ansatz equations, energy eigenvalues, form factors and the spectral representation of the density-density correlation function. These results are then used in Section 5 to obtain a fully explicit expression for the dynamical density-density correlator (and the related dynamical structure factor) in the thermodynamic limit, *cf.* equations (171), (166), (178) and (187). This constitutes the main result of our work. In Section 7 we obtain the asymptotic behaviour of the correlator and structure factor in various regimes. In particular we perform non-trivial consistency checks of our formulas, and recover known results from (nonlinear) Luttinger liquid theory and generalized hydrodynamics (GHD) [85, 86, 90].

## 2 Lieb-Liniger model

### 2.1 Definition

The Lieb-Liniger model [1,2] is a non-relativistic quantum field theory model with Hamiltonian

$$H = \int_0^L dx \left[ \psi^\dagger(x) \left( -\frac{\hbar^2}{2m} \frac{d^2}{dx^2} \right) \psi(x) + c \psi^\dagger(x) \psi^\dagger(x) \psi(x) \psi(x) \right], \tag{9}$$

where the canonical Bose field $\psi(x)$ satisfies equal-time commutation relations

$$[\psi(x), \psi^\dagger(y)] = \delta(x - y). \tag{10}$$

In the following we set $\hbar = 2m = 1$ and impose periodic boundary conditions. In first quantization (9) corresponds to a quantum mechanical system of $N$ particles with positions $0 \leq x_1, ..., x_N \leq L$ and Hamiltonian

$$H = \sum_{k=1}^N -\left( \frac{\partial}{\partial x_k} \right)^2 + 2c \sum_{j<k} \delta(x_j - x_k). \tag{11}$$

For later convenience we define the density operator at position $x$

$$\sigma(x) = \psi^\dagger(x) \psi(x), \tag{12}$$

and its time-$t$ evolved version $\sigma(x, t) = e^{iHt} \sigma(x) e^{-iHt}$.

### 2.2 The Bethe ansatz solution

#### 2.2.1 The spectrum

The Lieb-Liniger model is solvable by the Bethe ansatz: the energy $E$ and the momentum $P$ of an eigenstate $|\boldsymbol{\lambda}\rangle$ with $N$ bosons read

$$E(\boldsymbol{\lambda}) = \sum_{i=1}^N \lambda_i^2, \qquad P(\boldsymbol{\lambda}) = \sum_{i=1}^N \lambda_i, \tag{13}$$

where the rapidities $\boldsymbol{\lambda} = \{\lambda_1, .., \lambda_N\}$ satisfy the following set of "Bethe equations"

$$e^{iL\lambda_k} = \prod_{\substack{j=1 \\ j \neq k}}^N \frac{\lambda_k - \lambda_j + ic}{\lambda_k - \lambda_j - ic}, \quad k = 1, \ldots, N. \tag{14}$$

It is convenient to express them in logarithmic form

$$\frac{\lambda_k}{2\pi} = \frac{I_k}{L} - \frac{1}{L} \sum_{j=1}^N \frac{1}{\pi} \arctan \frac{\lambda_k - \lambda_j}{c}, \tag{15}$$

with $I_k$ an integer if $N$ is odd, a half-integer if $N$ is even. For $c > 0$, which we will assume in this paper, all the solutions to this equation are real [3].

### 2.2.2 The density form factors

As set out in the introduction, our aim is to calculate the density-density correlation function in an eigenstate $|\boldsymbol{\lambda}\rangle$

$$\langle \sigma(x,t)\sigma(0,0)\rangle = \frac{\langle \boldsymbol{\lambda}|\sigma(x,t)\sigma(0,0)|\boldsymbol{\lambda}\rangle}{\langle \boldsymbol{\lambda}|\boldsymbol{\lambda}\rangle}. \tag{16}$$

Our strategy is to use a Lehman representation in terms of energy eigenstates $|\boldsymbol{\mu}\rangle = |\mu_1,...,\mu_{N'}\rangle$, where $\{\mu_1,...,\mu_{N'}\}$ are solutions to the Bethe equations (15)

$$\begin{aligned}
\langle \sigma(x,t)\sigma(0,0)\rangle &= \sum_{\boldsymbol{\mu}} \frac{|\langle \boldsymbol{\lambda}|\sigma(0)|\boldsymbol{\mu}\rangle|^2}{\langle \boldsymbol{\lambda}|\boldsymbol{\lambda}\rangle \langle \boldsymbol{\mu}|\boldsymbol{\mu}\rangle} e^{it(E(\boldsymbol{\lambda})-E(\boldsymbol{\mu}))+ix(P(\boldsymbol{\mu})-P(\boldsymbol{\lambda}))} \\
&= \sum_{N'=0}^{\infty} \sum_{\mu_1<...<\mu_{N'}} \frac{|\langle \boldsymbol{\lambda}|\sigma(0)|\boldsymbol{\mu}\rangle|^2}{\langle \boldsymbol{\lambda}|\boldsymbol{\lambda}\rangle \langle \boldsymbol{\mu}|\boldsymbol{\mu}\rangle} e^{it(E(\boldsymbol{\lambda})-E(\boldsymbol{\mu}))+ix(P(\boldsymbol{\mu})-P(\boldsymbol{\lambda}))}.
\end{aligned} \tag{17}$$

The (normalized) form factors of local operators between two Bethe states have been derived in Refs [104–109]. In the case of the density operator $\sigma$, the (square of the normalized) form factor between two eigenstates $|\boldsymbol{\lambda}\rangle, |\boldsymbol{\mu}\rangle$ with respective numbers of Bethe roots $N, N'$ reads

$$\frac{|\langle \boldsymbol{\lambda}|\sigma(0)|\boldsymbol{\mu}\rangle|^2}{\langle \boldsymbol{\lambda}|\boldsymbol{\lambda}\rangle \langle \boldsymbol{\mu}|\boldsymbol{\mu}\rangle} = \delta_{N,N'} \frac{\left(\sum_{i=1}^{N}\mu_i-\lambda_i\right)^2}{L^{2N}\mathcal{N}_{\boldsymbol{\lambda}}\mathcal{N}_{\boldsymbol{\mu}}} \frac{\prod_{i\neq j}(\lambda_i-\lambda_j)(\mu_i-\mu_j)}{\prod_{i,j}(\lambda_i-\mu_j)^2} \prod_{i\neq j}\frac{\lambda_i-\lambda_j+ic}{\mu_i-\mu_j+ic}$$

$$\times \left| \det_{i,j\neq p} \left[ (V_i^+ - V_i^-)\delta_{ij} + i(\mu_i-\lambda_i)\prod_{k\neq i}\frac{\mu_k-\lambda_i}{\lambda_k-\lambda_i}\left( \frac{2c}{(\lambda_i-\lambda_j)^2+c^2} - \frac{2c}{(\lambda_p-\lambda_j)^2+c^2} \right) \right] \right|^2. \tag{18}$$

Here $p \in \{1,...,N\}$ can be freely chosen,

$$V_i^{\pm} = \prod_{k=1}^{N}\frac{\mu_k-\lambda_i \pm ic}{\lambda_k-\lambda_i \pm ic}, \tag{19}$$

and $\mathcal{N}_{\boldsymbol{\lambda}}$ is given by [110]

$$\mathcal{N}_{\boldsymbol{\lambda}} = \det_{i,j=1,...,N} \left[ \delta_{ij}\left( 1+\frac{1}{L}\sum_{k=1}^{N}\frac{2c}{c^2+(\lambda_i-\lambda_k)^2} \right) - \frac{1}{L}\frac{2c}{c^2+(\lambda_i-\lambda_j)^2} \right]. \tag{20}$$

## 3 Thermodynamic description of eigenstates

In a finite system of size $L$ all eigenstates of the Hamiltonian are fully characterized by a set of $N$ Bethe numbers $I_k$, or equivalently a set of $N$ Bethe roots $\lambda_k$. The purpose of this section is to explain how to turn this description into one based on (continuous) distribution functions of these roots in the thermodynamic limit $L \to \infty$ when $N$ scales like $L$. In particular, contrary to a common misconception, we emphasize that the usual "root densities" defined below *do not* fully characterize an eigenstate in the thermodynamic limit; this observation turns out to be of crucial importance in our calculation.

### 3.1 Root density

In the thermodynamic limit, any sum of a *non-singular* (piece-wise continuous) function $f$ over the Bethe roots or Bethe numbers

$$S_L[f] = \frac{1}{L}\sum_k f(\lambda_k), \qquad \tilde{S}_L[f] = \frac{1}{L}\sum_k f(\tfrac{I_k}{L}), \tag{21}$$

is independent of the precise values taken individually by each $I_k$ or $\lambda_k$, and depends only on the *number of Bethe roots or Bethe numbers in any given interval*. This information is encoded in the so-called root density $\rho(\lambda) \geq 0$ and filling function $0 \leq \chi(\iota) \leq 1$. They are defined by the requirement that in the large $L$ limit

$$L\rho(\lambda)d\lambda = \text{number of Bethe roots in } [\lambda, \lambda + d\lambda]\,,$$
$$L\chi(\iota)d\iota = \text{number of Bethe numbers } I_k/L \text{ per length in } [\iota, \iota + d\iota]. \tag{22}$$

In the thermodynamic limit the sums (21) can be turned into integrals over these functions

$$S_\infty[f] = \int_{-\infty}^{\infty} f(\lambda)\rho(\lambda)\mathrm{d}\lambda\,, \qquad \tilde{S}_\infty[f] = \int_{-\infty}^{\infty} f(\iota)\chi(\iota)\mathrm{d}\iota\,. \tag{23}$$

The same holds for multidimensional sums of a multivariate *non-singular* function $f$, with

$$S_L[f] = \frac{1}{L^n} \sum_{k_1,\dots,k_n} f(\lambda_{k_1},\dots,\lambda_{k_n})\,, \tag{24}$$

converging to

$$S_\infty[f] = \int_{-\infty}^{\infty}\cdots\int_{-\infty}^{\infty} f(\lambda_1,...,\lambda_n)\rho(\lambda_1)\dots\rho(\lambda_n)\mathrm{d}\lambda_1\dots\mathrm{d}\lambda_n\,. \tag{25}$$

As far as expressions of the form (21) and (24) are concerned, an eigenstate in the thermodynamic limit is entirely characterized by the root density $\rho(\lambda)$, or equivalently the filling function $\chi(\iota)$. To relate these two equivalent quantities, we introduce the function $\vartheta(\lambda)$ as the $L \to \infty$ limit of a function $\vartheta(\lambda_k) \equiv \chi(\frac{I_k}{L})$ of the Bethe roots, where $I_k$ is the integer associated with $\lambda_k$. Using the Bethe equations (15) $\vartheta(\lambda)$ can be expressed in terms of $\chi$ and $\rho$ as

$$\vartheta(\lambda) = \chi\left(\frac{\lambda}{2\pi} + \frac{1}{\pi}\int_{-\infty}^{\infty} \arctan\left(\frac{\lambda-\mu}{c}\right)\rho(\mu)\mathrm{d}\mu\right). \tag{26}$$

The filling function $\chi(\iota)$ and the root density are then related through

$$\frac{\rho(\lambda)}{\vartheta(\lambda)} = \frac{1}{2\pi} + \frac{1}{2\pi}\int_{-\infty}^{\infty} \frac{2c}{c^2 + (\lambda-\mu)^2}\rho(\mu)\mathrm{d}\mu\,. \tag{27}$$

It is customary to introduce the so-called hole density $\rho_h(\lambda)$ defined by

$$\frac{\rho(\lambda)}{\vartheta(\lambda)} = \rho(\lambda) + \rho_h(\lambda)\,, \tag{28}$$

which again contains equivalent information to $\rho(\lambda)$ or $\chi(\iota)$. When expressed in terms of the particle and hole densities (27) is known as the *thermodynamic limit of the Bethe Ansatz equations* [111]. Finally, the particle density is given by

$$D \equiv \int_{-\infty}^{\infty} \rho(x)\mathrm{d}x = \lim_{L\to\infty} \frac{N}{L}\,. \tag{29}$$

We introduce the Fermi momentum $q_F$ defined by

$$q_F = \pi D\,. \tag{30}$$

Although there is a simple relation between $D$ and $q_F$, we will in the following sometimes use $D$ and sometimes $q_F$, depending on the physical context at hand. We also denote (in the units where $\hbar = 2m = 1$)

$$\omega_F = q_F^2\,. \tag{31}$$

## 3.2 Pair distribution function

### 3.2.1 Definition

Root densities entirely characterize the value of sums of the type (21) and (24) in the thermo-dynamic limit. However, some functions of the Bethe roots *cannot* be expressed solely in terms of root densities in the thermodynamic limit, and as a consequence can take *different values in the thermodynamic limit* for states that have the same root density. An example is provided by

$$\Sigma_L[g] = \frac{1}{L^3} \sum_{i \neq j} \frac{g(\lambda_i, \lambda_j)}{(\lambda_i - \lambda_j)^2}, \tag{32}$$

that we will encounter below [1]. The sum in (32) by definition depends on the *joint distribution function* of pairs of roots separated by $\mathcal{O}(L^{-1})$, and the latter clearly contains information beyond that contained in the root density (which does not distinguish between roots separated by $\mathcal{O}(L^{-1})$).

We first note that if we impose the constraint $|\lambda_i - \lambda_j| > \epsilon$ for a $\epsilon > 0$ then $\Sigma_L[g]$ vanishes in the thermodynamic limit. Hence, it only depends on $g(\lambda, \lambda)$ and its derivatives at $\lambda$. Taylor expanding $g(\lambda_i, \lambda_j)$ for $\lambda_i$ close to $\lambda_j$ reduces the order of the pole and makes the next terms vanish in the thermodynamic limit, so it depends only on $g(\lambda, \lambda)$. Being a linear functional of $g$ it can be written in the thermodynamic limit in the form

$$\Sigma_\infty[g] = \int_{-\infty}^{\infty} g(\lambda, \lambda) \gamma_{-2}(\lambda) d\lambda, \tag{33}$$

where the function $\gamma_{-2}(\lambda)$ depends on the state. We call $\gamma_{-2}(\lambda)$ a *pair distribution function* as it encodes information about the joint distribution of pairs of Bethe roots. The index $-2$ relates to the fact that we are summing over the inverse square of the difference between two Bethe roots. The pair distribution function $\gamma_{-2}(\lambda)$ characterizes certain properties of the thermodynamic limit of an eigenstate and is *unrelated* to the root density $\rho(\lambda)$. Two states can have the same $\rho(\lambda)$ but different $\gamma_{-2}(\lambda)$.

The simplest example is that of (translationally invariant) free fermions, where the Bethe roots reduce to the single-particle momenta. Here we may construct two sequences of eigen-states labelled by an integer $n$, with momenta $\{\lambda_i = \frac{ni}{L} | i = 1, \ldots, N\}$ and $\{\mu_{2i} = \frac{2ni}{L}, \mu_{2i+1} = \frac{2ni+1}{L} | i = 1, \ldots, N/2\}$ respectively. In the thermodynamic limit both states are described by a root density $\rho(\lambda) = 1/n$, but the pair distribution functions are different: $\gamma_{-2}(\lambda) = \frac{\pi^2}{3n^2}$ for the first state and $\gamma_{-2}(\lambda) = 1 + \frac{\pi^2}{12n^2} + \sum_{m \neq 0} \frac{1}{(2nm+1)^2}$ for the second one.

### 3.2.2 (Generalized) micro-canonical ensemble and representative states

The (generalized) micro-canonical ensemble average of a local operator $\mathcal{O}(x)$ is a priori defined as

$$\frac{1}{C_L} \sum_{\nu} \frac{\langle \nu | \mathcal{O}(x) | \nu \rangle}{\langle \nu | \nu \rangle}, \tag{34}$$

where the sum is over an appropriate "shell" of simultaneous eigenstates of the Hamiltonian and the local conservation laws of the theory. $C_L$ is the number of terms in the sum. In a large but finite volume this means that for thermal averages we fix the energy within a window that contains an exponential (in system size) number of eigenstates. In the case of generalized

---

[1]The summand does not need to be singular for this to happen: Another example is $L \sum_i g(\lambda_i)(\lambda_{i+1} - \lambda_i)^2$ if the Bethe roots are ordered $\lambda_1 < \lambda_2 < \ldots < \lambda_N$.

micro-canonical ensembles we fix the eigenvalues of (some or all) of the local conservation laws in an analogous fashion [91, 112]. It is believed that almost all states in the sum in (34) have identical local properties, and hence the sum over states can be replaced by an expectation value with respect to a single typical state $|\lambda\rangle$ in the thermodynamic limit

$$\lim_{L\to\infty}\frac{1}{C_L}\sum_{\nu}\frac{\langle\nu|\mathcal{O}(x)|\nu\rangle}{\langle\nu|\nu\rangle}=\lim_{L\to\infty}\frac{\langle\lambda|\mathcal{O}(x)|\lambda\rangle}{\langle\lambda|\lambda\rangle}\,. \tag{35}$$

The state $|\lambda\rangle$ is sometimes called a *representative state* and we follow this terminology here. We note that in practice there is a great deal of freedom in choosing a representative state in a large, finite volume.

### 3.2.3  Average over representative states

As we have seen above, the thermodynamic limit of the sum (32) cannot generally be expressed as an integral over the root density, but depends on the choice of representative state in the finite volume. The thermodynamic limit of these sums involves the separate function $\gamma_{-2}(\lambda)$ defined in (33). As we will see in the following, in our calculations of the density-density correlation function the dependence of certain intermediate quantities on $\gamma_{-2}(\lambda)$ eventually compensate and the end result depends only on the root density. However, it is a priori possible that in other calculations involving sums of form factors no such cancellations will occur and the end result will indeed depend on the choice of representative state through $\gamma_{-2}(\lambda)$ or an analogous quantity.

We now make the following observation. As we have discussed above, averages with respect to a Bethe state $|\lambda\rangle$ often emerge upon simplifying averages over exponentially (in system size) many representative states corresponding to a given root density $\rho(\lambda)$. By construction such averages will depend only on the density. This then poses the question what value (32) takes after averaging over all representative states with same root density in the thermodynamic limit. We now address this issue.

First, we need to define properly which states in a large finite volume $L$ are acceptable representative states for a given root density. We define a sequence of *sets of states* to be *complete* for the root density $\rho$ if the corresponding sequence of sets of solutions to the Bethe equations $(\mathfrak{S}_L)_{L\in\mathbb{N}}$ all give rise to the density $\rho$ in the thermodynamic limit, and if the number of elements of the set $\mathfrak{S}_L$ satisfies

$$\log|\mathfrak{S}_L|=LS_{\mathrm{YY}}[\rho]+o(L)\,, \tag{36}$$

where $S_{\mathrm{YY}}[\rho]$ is the Yang-Yang entropy [11]

$$S_{\mathrm{YY}}[\rho]=\int\Big[\big(\rho(\lambda)+\rho_h(\lambda)\big)\log\big(\rho(\lambda)+\rho_h(\lambda)\big)-\rho(\lambda)\log\rho(\lambda)-\rho_h(\lambda)\log\rho_h(\lambda)\Big]\mathrm{d}\lambda\,. \tag{37}$$

In order to build such a set in a large finite volume let us consider a root density $\rho(\lambda)$ at a given particle density $D=\int\rho(\lambda)\mathrm{d}\lambda$. Given the root density we may introduce a particle counting function by

$$z(\lambda)=\int_{-\infty}^{\lambda}\rho(x)\mathrm{d}x\,. \tag{38}$$

Next we choose a "coarsening function" $\epsilon_L$ with the property that $\epsilon_L\to 0$ and $L\epsilon_L\to\infty$ when $L\to\infty$ — for example one can take $\epsilon_L=\frac{1}{\sqrt{L}}$. We now split the real axis into $n_L$ "bins" $[x_{L,j},x_{L,j+1}]$ containing $\lfloor L\epsilon_L\rfloor$ Bethe roots by defining $x_{L,1},...,x_{L,n_L+1}$ such that $z(x_{L,i})=i\epsilon_L$ for $1\le i\le n_L+1=\lfloor D/\epsilon_L\rfloor$.

Finally we define $\mathfrak{S}_L$ as the set containing all the states in a finite volume $L$ that contain exactly $\lfloor L\epsilon_L \rfloor$ Bethe roots in each of the $n_L$ bins $[x_{L,i}, x_{L,i+1}]$. All states in $\mathfrak{S}_L$ have $N_L = (\lfloor D/\epsilon_L \rfloor - 1)\lfloor L\epsilon_L \rfloor$ Bethe roots, which for $L \to \infty$ by construction are distributed with density $\rho(\lambda)$. The number of elements of $\mathfrak{S}_L$ will depend on the number of "vacancies" in each of the bins, which in turn depend on the values of all the Bethe roots since they interact via the Bethe equations. However, asymptotically in $L$, we have $K_{L,i} = \lfloor L(x_{L,i+1} - x_{L,i})(\rho(x_{L,i}) + \rho_h(x_{L,i})) \rfloor$ vacancies in each of the bins, so that

$$|\mathfrak{S}_L| = \prod_{i=1}^{n_L} \binom{K_{L,i} + \mathcal{O}(L^0)}{\lfloor L\epsilon_L \rfloor}. \tag{39}$$

Using Stirling's formula in the large-$L$ limit one has

$$\log |\mathfrak{S}_L| = L S_{\mathrm{YY}}[\rho] + o(L), \tag{40}$$

which shows that $\mathfrak{S}_L$ is indeed a complete set of representative states for a given root density $\rho(\lambda)$.

We can now state our result for the average of (32) over all representative states with root density $\rho(\lambda)$:

$$\lim_{L\to\infty} \frac{1}{|\mathfrak{S}_L|} \sum_{\{\lambda_i\}_i \in \mathfrak{S}_L} \frac{1}{L^3} \sum_{i\neq j} \frac{g(\lambda_i, \lambda_j)}{(\lambda_i - \lambda_j)^2} = \frac{\pi^2}{3} \int_{-\infty}^{\infty} g(\lambda, \lambda)(\rho(\lambda) + \rho_h(\lambda))\rho(\lambda)^2 \mathrm{d}\lambda. \tag{41}$$

A proof of (41) is given in Appendix B.

We note that if we instead sum over rapidities distributed regularly according to the inverse of the counting function $z^{-1}(\lambda)$ without imposing that the rapidities are solutions of the Bethe equations, the sum takes a different value:

$$\lim_{L\to\infty} \frac{1}{L^3} \sum_{i\neq j} \frac{g(z^{-1}(i/L), z^{-1}(j/L))}{(z^{-1}(i/L) - z^{-1}(j/L))^2} = \frac{\pi^2}{3} \int_{-\infty}^{\infty} g(\lambda, \lambda)\rho(\lambda)^3 \mathrm{d}\lambda. \tag{42}$$

If we sum over rapidities distributed regularly according to the inverse of the counting function $z^{-1}(\lambda)$ and impose the Bethe equations, the sum (32) is not easily expressed in terms of $\rho$, but takes a value different from either (42) or (41). Hence formula (41) is both non-trivial and non-intuitive.

## 3.3 Principal values

### 3.3.1 Single principal value

The sums (24) can be expressed in terms of root densities in the thermodynamic limit, provided $f$ is non-singular. We have seen in the previous section that for functions $f$ with a quadratic singularity the thermodynamic limit value of the sum cannot be expressed in terms of the root density. We now turn to functions that are singular but integrable in a principal value sense. This is the case of the sum

$$\tilde{\Sigma}_L[g] = \frac{1}{L^2} \sum_{\substack{i,j \\ i\neq j}} \frac{g(\lambda_i, \lambda_j)}{\lambda_i - \lambda_j}. \tag{43}$$

We will assume that $g$ and $\rho$ are continuous. Symmetrizing the sum, we have

$$\tilde{\Sigma}_L[g] = \frac{1}{2L^2} \sum_{\substack{i,j \\ i\neq j}} \frac{g(\lambda_i, \lambda_j) - g(\lambda_j, \lambda_i)}{\lambda_i - \lambda_j}. \tag{44}$$

The function $F(x, y) = \frac{g(x,y)-g(y,x)}{x-y}$ is regular, so that it has the form of (24) and its thermodynamic limit can be expressed in terms of $\rho$ according to

$$\tilde{\Sigma}_\infty[g] = \frac{1}{2} \int_{-\infty}^{\infty} \int_{-\infty}^{\infty} \frac{g(\lambda, \mu) - g(\mu, \lambda)}{\lambda - \mu} \rho(\lambda)\rho(\mu)\mathrm{d}\lambda\mathrm{d}\mu. \tag{45}$$

Since the integrand is finite, one can remove a small shell $|\lambda - \mu| < \epsilon$ with an error of $\mathcal{O}(\epsilon)$, and then un-symmetrize the sum. This yields

$$\tilde{\Sigma}_\infty[g] = \fint \frac{g(\lambda, \mu)}{\lambda - \mu} \rho(\lambda)\rho(\mu)\mathrm{d}\lambda\mathrm{d}\mu, \tag{46}$$

with the following usual definition of the principal value integral

$$\fint \frac{F(\lambda)}{\lambda - \mu}\mathrm{d}\lambda = \lim_{\epsilon \to 0} \int_{|\lambda-\mu|>\epsilon} \frac{F(\lambda)}{\lambda - \mu}\mathrm{d}\lambda. \tag{47}$$

Hence, sums of type (43) can indeed be expressed in terms of root densities.

In contrast partial sums like

$$\frac{1}{L} \sum_{\substack{i \\ i \neq j}} \frac{g(\lambda_i, \lambda_j)}{\lambda_i - \lambda_j}, \tag{48}$$

at fixed $j$ cannot be expressed in terms of the root density in the thermodynamic limit.

### 3.3.2 Double principal values

Higher-dimensional sums of the form

$$\tilde{\Sigma}_L[g] = \frac{1}{L^3} \sum_{\substack{i,j,k \\ i \neq j \\ j \neq k}} \frac{g(\lambda_i, \lambda_j, \lambda_k)}{(\lambda_i - \lambda_j)(\lambda_j - \lambda_k)}, \tag{49}$$

can be treated likewise, but with subtleties hiding in the fact that $i$ can be equal to $k$. Separating out the term with $i = k$ and symmetrizing the remaining sum gives

$$\tilde{\Sigma}_L[g] = \frac{1}{6L^3} \sum_{\substack{i \neq j \\ j \neq k \\ i \neq k}} \sum_{\sigma \in \mathfrak{S}_3} \frac{g(\lambda_{\sigma(i)}, \lambda_{\sigma(j)}, \lambda_{\sigma(k)})(\lambda_{\sigma(k)} - \lambda_{\sigma(i)})\,\mathrm{sgn}(\sigma)}{(\lambda_i - \lambda_j)(\lambda_j - \lambda_k)(\lambda_k - \lambda_i)} - \frac{1}{L^3} \sum_{i \neq j} \frac{g(\lambda_i, \lambda_j, \lambda_i)}{(\lambda_i - \lambda_j)^2}. \tag{50}$$

The first term is regular so that (25) can be used, while the second term is of the type (32) and can be expressed in terms of $\gamma_{-2}(\lambda)$. In the first term we can remove the region where $|\lambda - \mu| < \epsilon$ or $|\lambda - \nu| < \epsilon$ or $|\nu - \mu| < \epsilon$ with an error that is $\mathcal{O}(\epsilon)$, and then un-symmetrize the integral. One obtains

$$\tilde{\Sigma}_\infty[g] = \fint \frac{g(\lambda, \mu, \nu)\rho(\lambda)\rho(\mu)\rho(\nu)}{(\lambda - \mu)(\mu - \nu)}\mathrm{d}\lambda\mathrm{d}\mu\mathrm{d}\nu - \int_{-\infty}^{\infty} g(\lambda, \lambda, \lambda)\gamma_{-2}(\lambda)\mathrm{d}\lambda, \tag{51}$$

where the simultaneous principal value in the triple-integral is defined as

$$\fint \frac{F(\lambda, \mu, \nu)}{(\lambda - \mu)(\mu - \nu)}\mathrm{d}\lambda\mathrm{d}\mu\mathrm{d}\nu = \lim_{\epsilon \to 0} \int_{\substack{|\lambda-\mu|>\epsilon \\ |\mu-\nu|>\epsilon \\ |\lambda-\nu|>\epsilon}} \frac{F(\lambda, \mu, \nu)}{(\lambda - \mu)(\mu - \nu)}\mathrm{d}\lambda\mathrm{d}\mu\mathrm{d}\nu. \tag{52}$$

As shown in Appendix A.1 this can be expressed in terms of the successive principal value triple-integral according to a Poincaré-Bertrand-like formula

$$\oiiint \frac{F(\lambda,\mu,\nu)}{(\lambda-\mu)(\mu-\nu)}\mathrm{d}\lambda\mathrm{d}\mu\mathrm{d}\nu = \oiiint \frac{F(\lambda,\mu,\nu)}{(\lambda-\mu)(\mu-\nu)}\mathrm{d}\lambda\mathrm{d}\mu\mathrm{d}\nu + \frac{\pi^2}{3}\int_{-\infty}^{\infty}F(\lambda,\lambda,\lambda)\mathrm{d}\lambda, \qquad (53)$$

where we defined

$$\oiiint \frac{F(\lambda,\mu,\nu)}{(\lambda-\mu)(\mu-\nu)}\mathrm{d}\lambda\mathrm{d}\mu\mathrm{d}\nu = \int \mathrm{d}\mu \fint \mathrm{d}\nu \frac{1}{\mu-\nu}\fint \mathrm{d}\lambda \frac{F(\lambda,\mu,\nu)}{\lambda-\mu}$$

$$= \int \mathrm{d}\mu \lim_{\epsilon\to0}\int_{|\nu-\mu|>\epsilon}\mathrm{d}\nu\frac{1}{\mu-\nu}\lim_{\epsilon'\to0}\int_{|\mu-\lambda|>\epsilon'}\mathrm{d}\lambda\frac{F(\lambda,\mu,\nu)}{\lambda-\mu}. \qquad (54)$$

It can also be expressed as

$$\oiiint \frac{F(\lambda,\mu,\nu)}{(\lambda-\mu)(\mu-\nu)}\mathrm{d}\lambda\mathrm{d}\mu\mathrm{d}\nu = \int \mathrm{d}\nu \fint \mathrm{d}\mu \frac{1}{\mu-\nu}\fint \mathrm{d}\lambda \frac{F(\lambda,\mu,\nu)}{\lambda-\mu}$$

$$= \int \mathrm{d}\lambda \fint \mathrm{d}\mu \frac{1}{\lambda-\mu}\fint \mathrm{d}\nu \frac{F(\lambda,\mu,\nu)}{\mu-\nu} \qquad (55)$$

$$= \int \mathrm{d}\mu \fint \mathrm{d}\lambda \frac{1}{\lambda-\mu}\fint \mathrm{d}\nu \frac{F(\lambda,\mu,\nu)}{\mu-\nu},$$

and

$$\oiiint \frac{F(\lambda,\mu,\nu)}{(\lambda-\mu)(\mu-\nu)}\mathrm{d}\lambda\mathrm{d}\mu\mathrm{d}\nu = \lim_{\epsilon,\epsilon'\to0}\int_{\substack{|\lambda-\mu|>\epsilon \\ |\mu-\nu|>\epsilon'}}\frac{F(\lambda,\mu,\nu)}{(\lambda-\mu)(\mu-\nu)}\mathrm{d}\lambda\mathrm{d}\mu\mathrm{d}\nu, \qquad (56)$$

as shown in Appendices A.2. Using these principal value integral identities we can rewrite (51) in the form

$$\tilde{\Sigma}_{\infty}[g] = \oiiint \frac{g(\lambda,\mu,\nu)\rho(\lambda)\rho(\mu)\rho(\nu)}{(\lambda-\mu)(\mu-\nu)}\mathrm{d}\lambda\mathrm{d}\mu\mathrm{d}\nu + \int_{-\infty}^{\infty}g(\lambda,\lambda,\lambda)\left[\frac{\pi^2}{3}\rho(\lambda)^3 - \gamma_{-2}(\lambda)\right]\mathrm{d}\lambda. \qquad (57)$$

### 3.4 Examples of root densities

The calculations presented in this paper hold for a generic piece-wise continuous root density $\rho(\lambda)$. Two applications we have in mind is to thermal states and non-equilibrium steady states after quantum quenches, and we now discuss specific root densities that arise in these contexts.

### 3.4.1 Thermal states

Thermal states are characterized by root densities that maximise the Yang-Yang entropy at inverse temperature $\beta$ [11]. Defining the so-called dressed energy $\varepsilon_{\mathrm{dr}}(\lambda)$ by

$$\vartheta(\lambda) = \frac{1}{1+e^{\beta\varepsilon_{\mathrm{dr}}(\lambda)}}, \qquad (58)$$

the filling function $\vartheta(\lambda)$ of a thermal state is such that

$$\varepsilon_{\mathrm{dr}}(\lambda) = \lambda^2 - h - \frac{1}{2\pi\beta}\int_{-\infty}^{\infty}\frac{2c}{c^2+(\lambda-\mu)^2}\log(1+e^{-\beta\varepsilon_{\mathrm{dr}}(\mu)})\mathrm{d}\mu. \qquad (59)$$

Here $h$ is a chemical potential that is used to fix the desired particle density $D$. In practice one first solves the nonlinear integral equation (59) and then uses (58) to determine $\rho(\lambda)$ from the linear integral equation (27).

A particular case of thermal states is the zero temperature ground state, obtained in the limit $\beta \to \infty$. Its root density satisfies

$$\rho(\lambda) = \frac{1}{2\pi} + \frac{c}{\pi} \int_{-Q}^{Q} \frac{\rho(\mu)}{c^2 + (\lambda - \mu)^2} d\mu, \tag{60}$$

with $Q$ defined such that

$$\int_{-Q}^{Q} \rho(\lambda) d\lambda = D. \tag{61}$$

### 3.4.2 Non-equilibrium steady states

Refs [97, 99] considered a particular interaction quench in the Lieb-Liniger model, where the system is initially in the ground state of (9) for $c = 0$, and is subsequently time-evolved with the Lieb-Liniger Hamiltonian at a finite value of $c$. The root density characterizing the steady state reached at late times was determined in [99] and remarkably allows for a closed form solution

$$\rho_{ss}(\lambda) = \frac{\tau}{4\pi(1 + a(\lambda/c))} \frac{da(\lambda/c)}{d\tau}, \tag{62}$$

where $\tau = \frac{1}{c} \int_{-\infty}^{\infty} \rho_{ss}(x) dx$ and

$$a(x) = \frac{2\pi\tau}{x \sinh(2\pi x)} I_{1-2ix}(4\sqrt{\tau}) I_{1+2ix}(4\sqrt{\tau}), \tag{63}$$

with $I$ the modified Bessel function.

## 4 $1/c$ expansion of the Lieb-Liniger model

In this section we perform an expansion around the limit $c \to \infty$ at order $1/c^2$ of the energy levels and form factors in the Lieb-Liniger model, at fixed $L$ and fixed Bethe numbers. We then expose the consequences it has on the spectral sum (17) in Section 4.3.3.

### 4.1 The Bethe equations

The Bethe equations (15) admit a regular $1/c$ expansion at large $c$. In the following, in order to expand the form factor at order $1/c^2$ we will need the value of the Bethe roots at order $1/c^3$. The Bethe equations (15) at order $1/c^3$ read

$$\lambda_i = \frac{2\pi I_i}{L} - \frac{2}{cL} \sum_{k=1}^{N} (\lambda_i - \lambda_k) + \frac{2}{3c^3 L} \sum_{k=1}^{N} (\lambda_i - \lambda_k)^3 + \mathcal{O}(c^{-5}). \tag{64}$$

This gives the following expression for the Bethe roots in terms of the Bethe numbers at order $1/c^3$

$$\lambda_i = \frac{2\pi}{1 + \frac{2D}{c}} \frac{I_i}{L} + \frac{4\pi}{c(1 + \frac{2D}{c})} \frac{1}{L} \sum_{j=1}^{N} \frac{I_j}{L} + \frac{1}{3\pi c^3} \left(\frac{2\pi}{1 + \frac{2D}{c}}\right)^4 \frac{1}{L} \sum_{j=1}^{N} \left(\frac{I_i - I_j}{L}\right)^3 + \mathcal{O}(c^{-4}). \tag{65}$$

The alert reader will have noticed that some of the terms contain higher powers of $1/c$ than the order at which we are working, that is $1/c^3$. We find it useful throughout the manuscript to retain certain "resummed" expressions of $1/c$ as they appear in calculations, both for clarity and convenience since they often happen to compensate each other. In any case, keeping these resummed expressions in $1/c$ does not affect the validity of the equations at the order considered.

### 4.2   The form factors

#### 4.2.1   Leading order in $1/c$ of the form factor between two generic states

The behaviour of the $1/c$ expansion of a form factor (18) between states $|\boldsymbol{\lambda}\rangle$ and $|\boldsymbol{\mu}\rangle$ depends on the "relative positions" of the Bethe numbers of one state to the other. To see this, let us determine the leading order in $1/c$ of the form factor (18) without making any assumptions on the eigenstates $|\boldsymbol{\lambda}\rangle$ and $|\boldsymbol{\mu}\rangle$. It is then straightforward to see that when $c \to \infty$

$$V_j^+ - V_j^- = \frac{2}{ic} \sum_{k=1}^{N} (\mu_k - \lambda_k) + \mathcal{O}(c^{-2}) \tag{66}$$
$$\mathcal{N}_{\boldsymbol{\lambda}} = 1 + \mathcal{O}(c^{-1}),$$

while the non-diagonal term in the determinant appearing in the form factor is of order $\mathcal{O}(c^{-3})$. We conclude that

$$\frac{|\langle \boldsymbol{\lambda}|\sigma(0)|\boldsymbol{\mu}\rangle|^2}{\langle \boldsymbol{\lambda}|\boldsymbol{\lambda}\rangle \langle \boldsymbol{\mu}|\boldsymbol{\mu}\rangle} = \frac{\left(\sum_i \mu_i - \lambda_i\right)^{2N}}{L^{2N}} \left(\frac{2}{c}\right)^{2N-2} \frac{\prod_{i \neq j}(\lambda_i - \lambda_j)(\mu_i - \mu_j)}{\prod_{i,j}(\lambda_i - \mu_j)^2} (1 + \mathcal{O}(c^{-1})). \tag{67}$$

We see that the order in $1/c$ of this expression entirely depends on the roots $\lambda_k$ and $\mu_k$. To be specific, let us now denote by $I_k$ and $J_k$ the Bethe numbers of $\boldsymbol{\lambda}$ and $\boldsymbol{\mu}$ respectively, and define

$$\nu = N - |\{I_k\} \cap \{J_k\}|, \tag{68}$$

the number of Bethe numbers present in $\boldsymbol{\lambda}$ and absent from $\boldsymbol{\mu}$. If $\lambda_i$ and $\mu_j$ have different Bethe numbers, then from (65) we have $\lambda_i - \mu_j = \mathcal{O}(c^0)$, whereas if they have the same Bethe number then at least $\lambda_i - \mu_j = \mathcal{O}(c^{-1})$. It follows that

$$\frac{|\langle \boldsymbol{\lambda}|\sigma(0)|\boldsymbol{\mu}\rangle|^2}{\langle \boldsymbol{\lambda}|\boldsymbol{\lambda}\rangle \langle \boldsymbol{\mu}|\boldsymbol{\mu}\rangle} = \mathcal{O}(c^{-2(\nu-1)}). \tag{69}$$

Hence expanding in $1/c$ naturally orders the Lehman representation (17) into an expansion in terms of number of *particle-hole excitations* of $\boldsymbol{\mu}$ above $\boldsymbol{\lambda}$, i.e. of the number of changes in the Bethe numbers of $\boldsymbol{\mu}$ compared to those of $\boldsymbol{\lambda}$. This means that if one considers (17) at order $c^{-m}$, then only intermediate states $\boldsymbol{\mu}$ with $\nu \leq \frac{m}{2} + 1$ contribute to the sum. We note however that the converse is not true: restricting (17) to e.g. one-particle-hole excitations would still involve arbitrarily high orders in $1/c$.

Since our goal is to compute correlations at order $1/c^2$, we only need to investigate the restriction of (67) to one- and two-particle-hole excitations.

#### 4.2.2   Order $1/c^2$ of form factors involving a single particle-hole excitation

In this section we consider one-particle-hole excitations of the state $|\boldsymbol{\mu}\rangle$ above $|\boldsymbol{\lambda}\rangle$. Up to reordering the roots, we can assume that the Bethe numbers $I_k$ of $\boldsymbol{\lambda}$ differ from those $J_k$ of $\boldsymbol{\mu}$ only at a single position $a$:

$$\forall i \neq a \quad I_i = J_i, \qquad J_a - I_a \equiv n \neq 0. \tag{70}$$

Since the excited particle cannot coincide with an already existing particle, we also have the constraint

$$\forall i \neq a \quad I_a + n \neq I_i. \tag{71}$$

This has the following consequences at order $1/c^3$ on the value of the Bethe roots. Using (65) we have

$$\mu_a = \lambda_a + \frac{2\pi n}{L(1 + 2D/c)} \left(1 + \frac{2}{cL}\right) + \mathcal{O}(c^{-3}), \tag{72}$$

while for $i \neq a$, we obtain

$$\mu_i = \lambda_i + \frac{4\pi n}{cL^2(1+2D/c)}\left(1 - \frac{(\lambda_i - \lambda_a)^2}{c^2} + \frac{2\pi n}{c^2 L}(\lambda_i - \lambda_a) - \frac{1}{3c^2}\left(\frac{2\pi n}{L}\right)^2\right) + \mathcal{O}(c^{-4}). \quad (73)$$

Using (72) and (73) we can determine the various terms entering the expression of the form factor at order $c^{-2}$

$$\prod_{i\neq j} \frac{\lambda_i - \lambda_j + ic}{\mu_i - \mu_j + ic} = 1 - \frac{1}{c^2}\left(\frac{2\pi n}{L}\right)^2 \sum_{i\neq a} 1 + \frac{4\pi n}{Lc^2}\sum_{i\neq a} \lambda_i - \lambda_a ,$$

$$V_i^+ - V_i^- = \frac{4\pi n}{icL}\left(1 - \frac{(\lambda_i - \lambda_a)^2}{c^2}\right),$$

$$\prod_{\substack{i\neq j \\ i\neq a \\ j\neq a}} \frac{(\lambda_i - \lambda_j)(\mu_i - \mu_j)}{(\lambda_i - \mu_j)^2} = 1 + \left(\frac{4\pi n}{cL^2}\right)^2 \sum_{\substack{i\neq j \\ i\neq a \\ j\neq a}} \frac{1}{(\lambda_i - \lambda_j)^2} ,$$

$$\prod_{i\neq a} \frac{(\lambda_i - \lambda_a)^2}{(\mu_i - \lambda_a)^2} = 1 - \frac{8\pi n}{cL^2(1+2D/c)}\sum_{i\neq a} \frac{1}{\lambda_i - \lambda_a}$$

$$+ \left(\frac{4\pi n}{cL^2}\right)^2\left[2\left(\sum_{i\neq a} \frac{1}{\lambda_i - \lambda_a}\right)^2 + \sum_{i\neq a} \frac{1}{(\lambda_i - \lambda_a)^2}\right], \quad (74)$$

$$\prod_{i\neq a} \frac{(\mu_i - \mu_a)^2}{(\lambda_i - \mu_a)^2} = 1 + \frac{8\pi n}{cL^2(1+\frac{2D}{c})}\sum_{i\neq a} \frac{1}{\lambda_i - \lambda_a - \frac{2\pi n}{L(1+\frac{2D}{c})}}$$

$$+ \left(\frac{4\pi n}{cL^2}\right)^2\left[2\left(\sum_{i\neq a} \frac{1}{\lambda_i - \lambda_a - \frac{2\pi n}{L(1+\frac{2D}{c})}}\right)^2 + \sum_{i\neq a} \frac{1}{(\lambda_i - \lambda_a - \frac{2\pi n}{L(1+\frac{2D}{c})})^2}\right],$$

$$\prod_{i=1}^{N} \frac{1}{(\lambda_i - \mu_i)^2} = \frac{4(1+\frac{2D}{c})^{2N}}{(1+\frac{2}{cL})^2} \frac{c^{2N-2}L^{4N-2}}{n^{2N}(4\pi)^{2N}}$$

$$\times \left[1 + \frac{2}{c^2}\sum_{i\neq a}\left((\lambda_i - \lambda_a)^2 - \frac{2\pi n}{L}(\lambda_i - \lambda_a) + \frac{1}{3}\left(\frac{2\pi n}{L}\right)^2\right)\right],$$

$$\mathcal{N}_{\boldsymbol{\lambda}} = \mathcal{N}_{\boldsymbol{\mu}} = \left(1 + \frac{2D}{c}\right)^{N-1}, \quad (75)$$

$$i(\mu_l - \lambda_l)\prod_{k\neq l} \frac{\mu_k - \lambda_l}{\lambda_k - \lambda_l}\left(\frac{2c}{(\lambda_l - \lambda_j)^2 + c^2} - \frac{2c}{(\lambda_p - \lambda_j)^2 + c^2}\right) = \mathcal{O}(c^{-4}). \quad (76)$$

Putting everything together we have at order $c^{-2}$

$$\frac{|\langle \boldsymbol{\lambda}|\sigma(0)|\boldsymbol{\mu}\rangle|^2}{\langle \boldsymbol{\lambda}|\boldsymbol{\lambda}\rangle\langle \boldsymbol{\mu}|\boldsymbol{\mu}\rangle} = \frac{(1+\frac{2D}{c})^2}{(1+\frac{2}{cL})^2}\frac{1}{L^2}\left[1 + \frac{4}{cL(1+\frac{2D}{c})}\frac{2\pi n}{L}\sum_{i\neq a}\left(\frac{1}{\lambda_i - \lambda_a - \frac{2\pi n}{L(1+\frac{2D}{c})}} - \frac{1}{\lambda_i - \lambda_a}\right)\right.$$

$$+ \frac{4}{c^2 L^2}\left(\frac{2\pi n}{L}\right)^2\left(-\frac{L^2}{12}\sum_{i\neq a} 1 + \sum_{\substack{i\neq j \\ j\neq a}}\frac{1}{(\lambda_i - \lambda_j)^2} + 2\left(\sum_{i\neq a}\frac{1}{\lambda_i - \lambda_a - \frac{2\pi n}{L(1+\frac{2D}{c})}} - \frac{1}{\lambda_i - \lambda_a}\right)^2\right.$$

$$\left.\left.+ \sum_{i\neq a}\frac{1}{(\lambda_i - \lambda_a - \frac{2\pi n}{L(1+\frac{2D}{c})})^2}\right)\right] + \mathcal{O}(c^{-3}). \quad (77)$$

### 4.2.3 Order $1/c^2$ of form factors involving two particle-hole excitations

We now consider two particle-hole excitations. Up to re-ordering the roots of $\boldsymbol{\mu}$, we can assume its Bethe numbers differ from those of $\boldsymbol{\lambda}$ only at positions $a$ and $b \neq a$, and thus assume

$$\forall i \neq a, b \quad I_i = J_i, \qquad J_a - I_a \equiv n \neq 0, \qquad J_b - I_b \equiv m \neq 0. \tag{78}$$

Since the excited particles cannot coincide with an already existing particle, we also have the constraints

$$\forall i \neq a, b \quad I_a + n \neq I_i, \qquad \forall i \neq a, b \quad I_a + m \neq I_i. \tag{79}$$

Moreover we must also exclude the case where one of the excited particles fill the hole left by the other, since this reduces to a single particle-hole excitation and is therefore already covered. The corresponding constraint is

$$I_a + n \neq I_b, \qquad I_b + m \neq I_a. \tag{80}$$

Finally we have to exclude the case where the two excited particles coincide

$$I_a + n \neq I_b + m. \tag{81}$$

From (65) we obtain

$$\mu_i = \begin{cases} \lambda_i + \frac{4\pi(n+m)}{c\left(1+\frac{2D}{c}\right)L^2} + \mathcal{O}(c^{-3}) & \text{if } i \neq a, b \\ \lambda_a + \frac{2\pi n}{L\left(1+\frac{2D}{c}\right)} + \frac{4\pi(n+m)}{c\left(1+\frac{2D}{c}\right)L^2} + \mathcal{O}(c^{-3}) & \text{if } i = a \\ \lambda_b + \frac{2\pi m}{L\left(1+\frac{2D}{c}\right)} + \frac{4\pi(n+m)}{c\left(1+\frac{2D}{c}\right)L^2} + \mathcal{O}(c^{-3}) & \text{if } i = b. \end{cases} \tag{82}$$

We can now investigate the form taken by (67) for these values of roots. At leading order in $1/c$ we have

$$\frac{\prod_{i \neq j}(\lambda_i - \lambda_j)(\mu_i - \mu_j)}{\prod_{i,j}(\lambda_i - \mu_j)^2} = \frac{(\lambda_a - \lambda_b)^2(\mu_a - \mu_b)^2}{(\lambda_a - \mu_b)^2(\lambda_b - \mu_a)^2}\frac{1}{\prod_i(\lambda_i - \mu_i)^2}(1 + \mathcal{O}(c^{-1})), \tag{83}$$

which, when substituted in (67) yields the following leading order expression of the form factor for two-particle-hole excitations

$$\frac{|\langle\boldsymbol{\lambda}|\sigma(0)|\boldsymbol{\mu}\rangle|^2}{\langle\boldsymbol{\lambda}|\boldsymbol{\lambda}\rangle\langle\boldsymbol{\mu}|\boldsymbol{\mu}\rangle} = \frac{4}{c^2 L^4}\frac{(n+m)^4}{n^2 m^2}\frac{(\lambda_a - \lambda_b)^2(\lambda_a - \lambda_b + \frac{2\pi(n-m)}{L(1+2D/c)})^2}{(\lambda_a - \lambda_b + \frac{2\pi n}{L(1+2D/c)})^2(\lambda_a - \lambda_b - \frac{2\pi m}{L(1+2D/c)})^2} + \mathcal{O}(c^{-3}). \tag{84}$$

### 4.3 The Lehmann representation

We can now write the Lehmann representation (17) for the density-density correlation functions at order $1/c^2$. As explained in the previous section, only one and two particle-hole excitations contribute to (17) at order $1/c^2$, and the corresponding form factors were computed at this order in the previous subsections. This leaves us with working out the phases in the corresponding terms in (17) at order $1/c^2$.

### 4.3.1 The phase for a single particle-hole excitation

For excitations with one particle and one hole, it follows from (72) and (73) that

$$x\big(P(\boldsymbol{\mu})-P(\boldsymbol{\lambda})\big)+t\big(E(\boldsymbol{\lambda})-E(\boldsymbol{\mu})\big)=x\frac{2\pi n}{L}-t\Bigg[\frac{8\pi n}{cL^2(1+\frac{2D}{c})}\sum_i\lambda_i$$
$$+\Big(\frac{2\pi n}{L(1+\frac{2D}{c})}\Big)^2(1+\tfrac{4}{cL}+\tfrac{4D}{c^2L})+\frac{4\pi n}{L(1+\frac{2D}{c})}\lambda_a\Bigg]+\mathcal{O}(c^{-3}). \tag{85}$$

It will be convenient to perform the following change of variable $x'$ defined as

$$x'=x(1+\tfrac{2D}{c})-\frac{4\delta_L}{c}t, \tag{86}$$

where

$$\delta_L=\frac{1}{L}\sum_{i=1}^N\lambda_i. \tag{87}$$

Then the phase becomes

$$x(P(\boldsymbol{\mu})-P(\boldsymbol{\lambda}))+t(E(\boldsymbol{\lambda})-E(\boldsymbol{\mu}))=-t\frac{2\pi n}{L(1+\frac{2D}{c})}\Bigg[\frac{2\pi n}{L(1+\frac{2D}{c})}+2\lambda_a+\mathcal{O}(L^{-1})\Bigg]$$
$$+x'\frac{2\pi n}{L(1+\frac{2D}{c})}+\mathcal{O}(c^{-3}). \tag{88}$$

For later convenience we define

$$\delta\equiv\lim_{L\to\infty}\delta_L=\int_{-\infty}^{\infty}x\rho(x)\mathrm{d}x. \tag{89}$$

### 4.3.2 The phase for two particle-hole excitations

Using (82)

$$x(P(\boldsymbol{\mu})-P(\boldsymbol{\lambda}))+t(E(\boldsymbol{\lambda})-E(\boldsymbol{\mu}))=x\frac{2\pi(n+m)}{L}+t\Bigg[\lambda_a^2-\Big(\lambda_a+\frac{2\pi n}{L\big(1+\frac{2D}{c}\big)}\Big)^2$$
$$+\lambda_b^2-\Big(\lambda_b+\frac{2\pi m}{L\big(1+\frac{2D}{c}\big)}\Big)^2-\frac{8\pi(n+m)}{cL^2\big(1+\frac{2D}{c}\big)}\sum_j\lambda_j$$
$$-\frac{16\pi^2}{L^4(1+\frac{2D}{c})^2c^2}(n+m)^2\sum_i1-\frac{16\pi^2}{L^3(1+\frac{2D}{c})^2c}(n+m)^2\Bigg]+\mathcal{O}(c^{-3}).$$

We can express this in terms of $x'$ as well

$$x(P(\boldsymbol{\mu})-P(\boldsymbol{\lambda}))+t(E(\boldsymbol{\lambda})-E(\boldsymbol{\mu}))=x'\frac{2\pi(n+m)}{L\big(1+\frac{2D}{c}\big)}+t\Bigg[\lambda_a^2-\Big(\lambda_a+\frac{2\pi n}{L\big(1+\frac{2D}{c}\big)}\Big)^2$$
$$+\lambda_b^2-\Big(\lambda_b+\frac{2\pi m}{L\big(1+\frac{2D}{c}\big)}\Big)^2+\mathcal{O}(L^{-1})\Bigg]+\mathcal{O}(c^{-3}). \tag{90}$$

### 4.3.3 The sum over intermediate states

So far we have expanded all the terms arising in (17) at order $1/c^2$, at a fixed $L$ for arbitrary eigenstates $|\boldsymbol{\lambda}\rangle$ and $|\boldsymbol{\mu}\rangle$ with fixed Bethe numbers. We have shown that the sum truncates to one- and two-particle-hole excitations, and that the resulting terms are well-defined functions of the excitation parameters $n$ and $m$.

However, as the Lieb Liniger model is a field theory and not a lattice model it features an *infinite* number of particle-hole states even if $L$ is finite, so that (17) is still an infinite sum even if it involves only one- and two-particle-hole excitations. This creates two notable problems. The first one is that we encounter infinite sums of the type $\sum_{k=-\infty}^{\infty} k^n e^{ik^2 t + ikx}$ for $n = 0, 1, 2$ which are *ill-defined as functions of $x, t$* (except if $n = 0$ and $t \neq 0$). The explanation for this behaviour is that $\langle \sigma(x, t)\sigma(0, 0)\rangle$, similarly to the propagator of a quantum particle, should be understood as a *probability amplitude* that is meant to be integrated against a smooth and localized function of $x$ and $t$, or, stated differently, that it must be understood as a distribution in $x, t$. The second problem is that the $1/c$ expansion of a form factor $\langle \boldsymbol{\lambda}|\sigma(0)|\boldsymbol{\mu}\rangle$ has been performed for fixed Bethe numbers, whereas in the spectral sum at fixed $c$ there are always excited states with Bethe roots larger than $c$. This poses a potential problem of commuting two limits.

In order to address these problems we are going to impose that all the rapidities involved in the spectral sum (17) are smaller than a certain *cut-off* $\Lambda$, that can be taken as large as desired. Firstly, this imposes a restriction of the state $|\boldsymbol{\lambda}\rangle$, in which we are calculating our expectation value. We require that for all roots $|\lambda_j| < \Lambda$, i.e. that the density $\rho(\lambda)$ vanishes for $|\lambda| > \Lambda$; this is a mild restriction in the following sense. In practice we are interested in the dynamical response in macro states characterized by root distributions $\rho(\lambda)$ that decay faster than $|\lambda|^{-2}$ for $\lambda \to \infty$, which is a necessary condition for the energy density of the state to be finite (for example in the thermal state the decay is Gaussian). We therefore can always approximate $\rho(\lambda)$ to any given accuracy by a root density $\rho_\Lambda(\lambda)$, which vanishes outside the interval $[-\Lambda, \Lambda]$. Moreover this truncation can be done in an infinitely differentiable way, so it does not affect the regularity of the root density $\rho(\lambda)$. Secondly, this cut-off also restrains the sum (17) to excited states $|\boldsymbol{\mu}\rangle$ such that the $|\mu_i| < \Lambda$, which removes the problem of possible excited rapidities becoming larger than $c$. Hence we define a $\Lambda$-regularized correlation function $\langle \sigma(x, t)\sigma(0, 0)\rangle_\Lambda$ as

$$\langle \sigma(x, t)\sigma(0, 0)\rangle_\Lambda = \sum_{\substack{\boldsymbol{\mu} \\ \forall i, |\mu_i| < \Lambda}} \frac{|\langle \boldsymbol{\lambda}|\sigma(0)|\boldsymbol{\mu}\rangle|^2}{\langle \boldsymbol{\lambda}|\boldsymbol{\lambda}\rangle \langle \boldsymbol{\mu}|\boldsymbol{\mu}\rangle} e^{it(E(\boldsymbol{\lambda}) - E(\boldsymbol{\mu})) + ix(P(\boldsymbol{\mu}) - P(\boldsymbol{\lambda}))}. \tag{91}$$

The correlator $\langle \sigma(x, t)\sigma(0, 0)\rangle_\Lambda$ defined in this way and expanded in $1/c$ has a regular thermodynamic limit $L \to \infty$, as we will see below. Now, in order to recover the true correlation functions (17), one would like to then take the limit $\Lambda \to \infty$. It turns out that such a limit of $\langle \sigma(x, t)\sigma(0, 0)\rangle_\Lambda$ seen as a function of $x, t$ does not exist. To be more specific one encounters problematic terms of the form

$$I_n(\Lambda|t, x) = \int_{-\Lambda}^{\Lambda} \mu^n e^{-it\mu^2 + ix\mu} d\mu, \quad n = 0, 1, 2, \dots \tag{92}$$

for which the limit $\Lambda \to \infty$ does not exist (except for $n = 0$ if $t \neq 0$). However, the limit exists in a distribution sense, i.e. the integral of $I_n(\Lambda|t, x)$ over any smooth localized function of $x, t$ has a well-defined limit when $\Lambda \to \infty$. This is all we require, since the correlation function is in any case meant to be integrated with a smooth localized function of $x, t$.

To take the limit we perform an integration by part and obtain

$$I_n(\Lambda|t, x) = \frac{x I_{n-1}(\Lambda|t, x)}{2t} + \frac{(n-1) I_{n-2}(\Lambda|t, x)}{2it} + \frac{e^{-ix\Lambda}(-1)^{n-1} - e^{ix\Lambda}}{2it} e^{-it\Lambda^2} \Lambda^{n-1}. \tag{93}$$

In particular we have

$$
\begin{aligned}
I_1(\Lambda|t,x) &= \frac{x}{2t}I_0(\Lambda|t,x) + \frac{e^{-ix\Lambda}-e^{ix\Lambda}}{2it}e^{-it\Lambda^2} \\
I_2(\Lambda|t,x) &= \left(\left(\frac{x}{2t}\right)^2 + \frac{1}{2it}\right)I_0(\Lambda|t,x) - \frac{e^{-ix\Lambda}+e^{ix\Lambda}}{2it}e^{-it\Lambda^2}\Lambda + x\frac{e^{-ix\Lambda}-e^{ix\Lambda}}{4it^2}e^{-it\Lambda^2} ,
\end{aligned}
\tag{94}
$$

where

$$
\begin{aligned}
\lim_{\Lambda\to\infty}I_0(\Lambda|t,x) &= \int_{-\infty}^{\infty}e^{-it\mu^2+i\mu x}\mathrm{d}\mu, \qquad \text{if } t \neq 0 , \\
I_0(\Lambda|t,x) &= \frac{e^{ix\Lambda}-e^{-ix\Lambda}}{ix} , \qquad\quad \text{if } t = 0.
\end{aligned}
\tag{95}
$$

Terms like $e^{-it\Lambda^2 \mp ix\Lambda}\Lambda^n$ and $e^{ix\Lambda}$ do not have limits when $\Lambda\to\infty$ as a function of $x,t$. In a distribution sense however, they vanish when $\Lambda\to\infty$ in the sense that their integral with any smooth localized function of $x,t$ vanishes when $\Lambda\to\infty$. Hence we obtain that when $\Lambda\to\infty$ $I_n(\Lambda|t,x)$ tends to $I_n(t,x)$ with $I_n(0,x)=0$ and

$$
\begin{aligned}
I_1(t,x) &= \frac{x}{2t}I_0(t,x) , \\
I_2(t,x) &= \left(\left(\frac{x}{2t}\right)^2 + \frac{1}{2it}\right)I_0(t,x) , \\
I_0(t,x) &= \int_{-\infty}^{\infty}e^{-it\mu^2+i\mu x}\mathrm{d}\mu , \qquad t \neq 0.
\end{aligned}
\tag{96}
$$

One notices that these limits are exactly those obtained by introducing a small imaginary part in time and taking $\Lambda\to\infty$

$$
I_n(t,x) = \lim_{\epsilon\to 0^+}\int_{-\infty}^{\infty}\mu^n e^{-i(t-i\epsilon)\mu^2+ix\mu}\mathrm{d}\mu .
\tag{97}
$$

However, such a small imaginary part cannot be incorporated from the beginning in (91), since $E(\boldsymbol{\lambda})-E(\boldsymbol{\mu})$ can take both signs when $|\boldsymbol{\lambda}\rangle$ is not the ground state.

These limits will be useful in the following sections in order to take the limit $\Lambda\to\infty$ of the $\Lambda$-regularized correlation functions.

At order $1/c^2$ we therefore have the following decomposition

$$
\langle\sigma(x,t)\sigma(0,0)\rangle_\Lambda = D^2 + \mathcal{C}_1^\Lambda(x,t) + \mathcal{C}_2^\Lambda(x,t) + \mathcal{O}(c^{-3}) ,
\tag{98}
$$

where $\mathcal{C}_{1,2}^\Lambda(x,t)$ are defined in the following. Introducing the convenient notations

$$
\lambda_{a,n} \equiv \lambda_a + \frac{2\pi n}{L(1+\frac{2D}{c})} ,
\tag{99}
$$

and

$$
L' = L\left(1 + \frac{2D}{c}\right) ,
\tag{100}
$$

we have the following contribution at order $c^{-2}$ of the one-particle-hole excitations

$$
\mathcal{C}_1^{\Lambda}(x,t) = \frac{(1+\frac{2D}{c})^2}{L^2} \sum_{a=1}^{N} \sum_{\substack{n \\ \forall k, \lambda_{a,n} \neq \lambda_k \\ |\lambda_{a,n}| < \Lambda}} \left[ 1 + \frac{4}{cL}\frac{2\pi n}{L'} \sum_{i \neq a}\left( \frac{1}{\lambda_i - \lambda_a - \frac{2\pi n}{L'}} - \frac{1}{\lambda_i - \lambda_a} \right) \right.
$$
$$
+ \frac{4}{c^2 L^2}\left(\frac{2\pi n}{L'}\right)^2\left( -\frac{L^2}{12}\sum_{i \neq a}1 + \sum_{\substack{i \neq j \\ j \neq a}}\frac{1}{(\lambda_i - \lambda_j)^2} + 2\left(\sum_{i \neq a}\frac{1}{\lambda_i - \lambda_a - \frac{2\pi n}{L'}} - \frac{1}{\lambda_i - \lambda_a}\right)^2 \right.
$$
$$
\left. \left. + \sum_{i \neq a}\frac{1}{(\lambda_i - \lambda_a - \frac{2\pi n}{L'})^2} \right) \right] \exp\left( ix'(\lambda_{a,n} - \lambda_a) + it\left(\lambda_a^2 - \lambda_{a,n}^2\right) \right).
$$

We already neglected a global factor $(1+\frac{2}{cL})^2$ that is 1 in the thermodynamic limit, as well as a $\mathcal{O}(L^{-1})$ contribution in the exponential. We also used $\frac{1}{L^2 c^2} = \frac{1}{L'^2 c^2} + \mathcal{O}(c^{-3})$ at order $c^{-2}$.

In Figure 1 we show the distribution of Bethe numbers for the particle-hole excitations that are summed over in (101). Compared to the representative state we have changed a single integer.

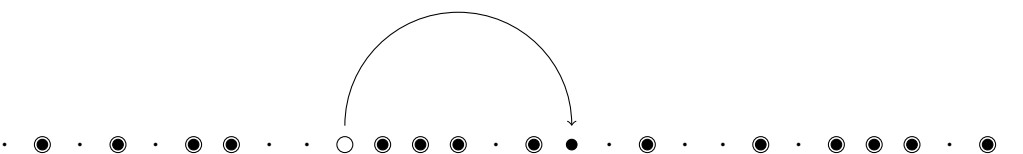

Figure 1: Sketch of a one-particle-hole excitation: positions of the momenta of the representative state (empty circles) and the intermediate state (filled circles) respectively, and position of the holes (dots).

For the two particle-hole excitations the sum in (17) is over the set $\{a, b\}$ and over $n, m$ with the constraint $\mu_a < \mu_b$. Since the form factor is symmetric upon swapping $a, b$ and $n, m$ simultaneously, this constraint can be taken into account with a factor $1/2$ and with imposing $\mu_a \neq \mu_b$. The sum over $\{a, b\}$ as a set can be transformed into a sum over $a \neq b$ as a couple with a factor $1/2$ as well. Hence we have the leading contribution of the two particle-hole excitations

$$
\mathcal{C}_2^{\Lambda}(x,t) = \frac{1}{c^2 L^4} \sum_{a \neq b} \sum_{\substack{n \\ \forall i, \lambda_{a,n} \neq \lambda_i \\ |\lambda_{a,n}| < \Lambda}} \sum_{\substack{m \\ \forall i, \lambda_{b,m} \neq \lambda_i \\ \lambda_{b,m} \neq \lambda_{a,n} \\ |\lambda_{b,m}| < \Lambda}} \frac{(n+m)^4}{n^2 m^2}\frac{(\lambda_a - \lambda_b)^2(\lambda_a - \lambda_b + \frac{2\pi}{L'}(n-m))^2}{(\lambda_a - \lambda_b + \frac{2\pi}{L'}n)^2(\lambda_a - \lambda_b - \frac{2\pi}{L'}m)^2}
$$
$$
\times \exp\left( it\left[\lambda_a^2 - \lambda_{a,n}^2 + \lambda_b^2 - \lambda_{b,m}^2\right] + ix'[\lambda_{a,n} - \lambda_a + \lambda_{b,m} - \lambda_b] \right). \quad (101)
$$

In Figure 2 we show the distribution of Bethe numbers for the two particle-hole excitations that are summed over in (101). Compared to the representative state we have changed two integers.

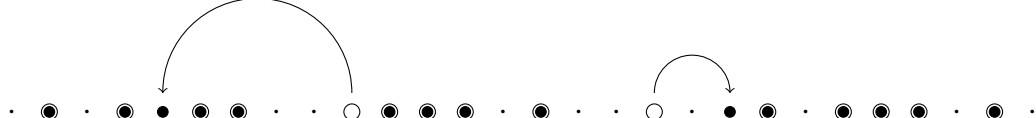

Figure 2: Sketch of a two particle-hole excitation: position of the momenta of the representative state (empty circles) and the intermediate state (filled circles) respectively, and position of the holes (dots).

## 4.4 Examples of root densities

In this subsection we complete the $1/c$ expansion of the model by determining the expansions of the root densities introduced in Section 3.4.

### 4.4.1 Hole density

We introduced earlier the hole density $\rho_h(\lambda)$ in (28) with $\vartheta(\lambda)$ given in terms of $\rho(\lambda)$ in (27). The hole density is a function of the root density $\rho(\lambda)$, and for a generic $\rho(\lambda)$ it reads at order $c^{-2}$

$$\rho_h(\lambda) = \frac{1 + \frac{2D}{c}}{2\pi} - \rho(\lambda) + \mathcal{O}(c^{-3}), \tag{102}$$

where we recall that $D$ is defined in (29).

### 4.4.2 Thermal states

Thermal states at finite inverse temperature $\beta < \infty$ are defined in terms of the nonlinear integral equation for the dressed energy (59) and the thermodynamic limit of the Bethe Ansatz equations (27). These can be expanded in $1/c$ without difficulty, and we obtain the following result for the particle density at order $1/c^2$

$$\rho(x) = \frac{1}{2\pi} \frac{A(c, \beta)}{1 + e^{\beta x^2 + B(c, \beta)}}, \tag{103}$$

where

$$A(c, \beta) = 1 - \frac{\text{Li}_{1/2}(-e^{\beta h})}{\sqrt{\pi \beta} c} + \frac{\text{Li}^2_{1/2}(-e^{\beta h}) + \text{Li}_{-1/2}(-e^{\beta h}) \text{Li}_{3/2}(-e^{\beta h})}{\pi \beta c^2} + \mathcal{O}(c^{-3}),$$

$$B(c, \beta) = -\beta h + \frac{\text{Li}_{3/2}(-e^{\beta h})}{\sqrt{\pi \beta} c} - \frac{\text{Li}_{1/2}(-e^{\beta h}) \text{Li}_{3/2}(-e^{\beta h})}{\pi \beta c^2} + \mathcal{O}(c^{-3}). \tag{104}$$

We recall that $h$ is the chemical potential used to fix the particle number $D$. In order to derive (104) we used the following relations

$$\int_{-\infty}^{\infty} \frac{\mathrm{d}x}{1 + e^{x^2 + y}} = -\sqrt{\pi} \, \text{Li}_{1/2}\left[-e^{-y}\right]$$

$$\int_{-\infty}^{\infty} \log(1 + e^{-x^2 - y}) \mathrm{d}x = -\sqrt{\pi} \, \text{Li}_{3/2}\left[-e^{-y}\right]. \tag{105}$$

### 4.4.3 Zero temperature ground state

Equation (60) for the ground state root density can be expanded in $1/c$ to yield

$$
\begin{aligned}
\rho(\lambda) &= \frac{1+\frac{2D}{c}}{2\pi} \mathbf{1}_{|\lambda|<Q} + \mathcal{O}(c^{-3}), \\
Q &= \frac{q_F}{1+\frac{2D}{c}} + \mathcal{O}(c^{-3}).
\end{aligned}
\tag{106}
$$

Here $\mathbf{1}_{\mathcal{P}}$ is the indicator function, equal to 1 if the affirmation $\mathcal{P}$ is true and 0 if it is false. The Luttinger parameter $K = (2\pi\rho(Q))^2$ [3] is

$$
K = 1 + \frac{4D}{c} + \frac{4D^2}{c^2} + \mathcal{O}(c^{-3}).
\tag{107}
$$

# 5 The thermodynamic limit of correlation functions

In this section we perform explicitly the sum over intermediate states in (98) at order $1/c^2$ in the infinite volume limit $L \to \infty$.

## 5.1 One particle-hole excitations

Our starting point is $\mathcal{C}_1^\Lambda(x,t)$ as defined in (101). In the following we consider the different orders in the $1/c$-expansion and derive integral representations of the corresponding contributions to $\mathcal{C}_1^\Lambda(x,t)$ in the thermodynamic limit. As we have noted before, we retain certain resummed expressions in this expansion for convenience, an example being the factor $(1+\frac{2D}{c})$. When we refer to a given order of the $1/c$-expansion this should be understood modulo such factors.

### 5.1.1 Order $c^0$

Let us focus first on the leading order $\mathcal{O}(c^0)$, namely

$$
\mathcal{A}_0 = \frac{(1+\frac{2D}{c})^2}{L^2} \sum_a \sum_{\substack{n \\ \forall k, \, \lambda_{a,n} \neq \lambda_k \\ |\lambda_{a,n}|<\Lambda}} e^{ix'(\lambda_{a,n}-\lambda_a)+it\left(\lambda_a^2-\lambda_{a,n}^2\right)}.
\tag{108}
$$

We rewrite this as

$$
\begin{aligned}
\mathcal{A}_0 ={}& \frac{(1+\frac{2D}{c})^2}{L^2} \sum_a \sum_{\substack{n \\ |\lambda_{a,n}|<\Lambda}} e^{ix'(\lambda_{a,n}-\lambda_a)+it\left(\lambda_a^2-\lambda_{a,n}^2\right)} \\
&- \frac{(1+\frac{2D}{c})^2}{L^2} \sum_a \sum_{\substack{k \\ |\lambda_k|<\Lambda}} e^{ix'(\lambda_k-\lambda_a)+it\left(\lambda_a^2-\lambda_k^2\right)}.
\end{aligned}
\tag{109}
$$

Using (25) the sums over $a$ and $k$ can be turned into integrals over the root density $\rho(\lambda)$, and the sum over $n$ into an integral with density $\frac{1+\frac{2D}{c}}{2\pi}$. Altogether we find

$$
\mathcal{A}_0 = (1+\tfrac{2D}{c})^2 \int_{-\infty}^{\infty} d\lambda \, \rho(\lambda) \int_{-\Lambda}^{\Lambda} d\mu \, \rho_h(\mu) e^{it(\lambda^2-\mu^2)+ix'(\mu-\lambda)} + \mathcal{O}(L^{-1}),
\tag{110}
$$

where we used the expression (102) for the hole density $\rho_h$ at order $c^{-2}$.

### 5.1.2 Order $c^{-1}$

We next turn to the $c^{-1}$ term

$$\mathcal{A}_1 = 4\frac{(1+\frac{2D}{c})^2}{cL^3} \sum_a \sum_{\substack{n \\ \forall k, \lambda_{a,n} \neq \lambda_k \\ |\lambda_{a,n}|<\Lambda}} \frac{2\pi n}{L'} \sum_{i \neq a} \left( \frac{1}{\lambda_i - \lambda_a - \frac{2\pi n}{L'}} - \frac{1}{\lambda_i - \lambda_a} \right)$$
$$\times e^{ix'(\lambda_{a,n}-\lambda_a)+it\left(\lambda_a^2-\lambda_{a,n}^2\right)}. \tag{111}$$

We rewrite this as

$$\mathcal{A}_1 = 4\frac{(1+\frac{2D}{c})^2}{cL^3} \sum_a \sum_{i \neq a} \sum_{\substack{n \\ \lambda_{a,n} \neq \lambda_i \\ |\lambda_{a,n}|<\Lambda}} \frac{2\pi n}{L'} \left( \frac{1}{\lambda_i - \lambda_a - \frac{2\pi n}{L'}} - \frac{1}{\lambda_i - \lambda_a} \right)$$
$$\times e^{ix'(\lambda_{a,n}-\lambda_a)+it\left(\lambda_a^2-\lambda_{a,n}^2\right)}$$
$$-4\frac{(1+\frac{2D}{c})^2}{cL^3} \sum_a \sum_{i \neq a} \sum_{\substack{k \\ k \neq i \\ |\lambda_k|<\Lambda}} (\lambda_k - \lambda_a) \left( \frac{1}{\lambda_i - \lambda_k} - \frac{1}{\lambda_i - \lambda_a} \right)$$
$$\times e^{ix'(\lambda_k-\lambda_a)+it\left(\lambda_a^2-\lambda_k^2\right)}. \tag{112}$$

This term involves either a sum over regularly spaced integers $n$ that becomes an integral with density $\frac{1+2D/c}{2\pi}$ in the thermodynamic limit, or sums of the type (43) that can be expressed as principal part integrals over the root density. We obtain

$$\mathcal{A}_1 = \frac{4(1+\frac{2D}{c})^2}{c} \int_{-\infty}^{\infty} d\lambda \, \rho(\lambda) \int_{-\Lambda}^{\Lambda} d\mu(\mu-\lambda) \left[ \fint \frac{\rho(u)}{u-\mu} du - \fint \frac{\rho(u)}{u-\lambda} du \right]$$
$$\times e^{it(\lambda^2-\mu^2)+ix'(\mu-\lambda)} \left( \frac{1+\frac{2D}{c}}{2\pi} - \rho(\mu) \right) + \mathcal{O}(L^{-1}). \tag{113}$$

Introducing the Hilbert transform $\tilde{\rho}$ of $\rho$ by

$$\tilde{\rho}(\lambda) = \fint \frac{\rho(u)}{\lambda-u} du, \tag{114}$$

permits us to rewrite this contribution in the form

$$\mathcal{A}_1 = -\frac{4(1+\frac{2D}{c})^2}{c} \int_{-\infty}^{\infty} d\lambda \rho(\lambda) \int_{-\Lambda}^{\Lambda} d\mu \rho_h(\mu)(\mu-\lambda) \left[ \tilde{\rho}(\mu) - \tilde{\rho}(\lambda) \right] e^{it(\lambda^2-\mu^2)+ix'(\mu-\lambda)}$$
$$+ \mathcal{O}(L^{-1}). \tag{115}$$

### 5.1.3 Order $c^{-2}$: first contribution

We now consider contributions involving the factor

$$\left( \sum_{i \neq a} \frac{1}{\lambda_i - \lambda_a - \frac{2\pi n}{L'}} - \frac{1}{\lambda_i - \lambda_a} \right)^2 = \sum_{\substack{i \neq a \\ j \neq a}} \frac{1}{(\lambda_i - \lambda_a)(\lambda_j - \lambda_a)}$$
$$+ \sum_{\substack{i \neq a \\ j \neq a}} \frac{1}{(\lambda_i - \lambda_a - \frac{2\pi n}{L'})(\lambda_j - \lambda_a - \frac{2\pi n}{L'})} - 2 \sum_{\substack{i \neq a \\ j \neq a}} \frac{1}{(\lambda_i - \lambda_a - \frac{2\pi n}{L'})(\lambda_j - \lambda_a)}, \tag{116}$$

which are more delicate. The first term on the right hand side gives rise to a contribution

$$
\mathcal{A}_2 = 8 \frac{(1+\frac{2D}{c})^2}{c^2 L^4} \sum_a \sum_{\substack{\forall k, \lambda_{a,n}\neq\lambda_k \\ |\lambda_{a,n}|<\Lambda}}^{n} \left(\frac{2\pi n}{L'}\right)^2 \sum_{\substack{i,j \\ i\neq a \\ j\neq a}} \frac{1}{(\lambda_i-\lambda_a)(\lambda_j-\lambda_a)}
$$
$$
\times e^{ix'(\lambda_{a,n}-\lambda_a)+it\left(\lambda_a^2-\lambda_{a,n}^2\right)}. \tag{117}
$$

We rewrite this as

$$
\mathcal{A}_2 = 8 \frac{(1+\frac{2D}{c})^2}{c^2 L^4} \sum_a \sum_{\substack{n \\ |\lambda_{a,n}|<\Lambda}} \left(\frac{2\pi n}{L'}\right)^2 \sum_{\substack{i,j \\ i\neq a \\ j\neq a}} \frac{e^{ix'(\lambda_{a,n}-\lambda_a)+it\left(\lambda_a^2-\lambda_{a,n}^2\right)}}{(\lambda_i-\lambda_a)(\lambda_j-\lambda_a)}
$$
$$
- 8 \frac{(1+\frac{2D}{c})^2}{c^2 L^4} \sum_a \sum_{\substack{k \\ |\lambda_k|<\Lambda}} (\lambda_k-\lambda_a)^2 \sum_{\substack{i,j \\ i\neq a \\ j\neq a}} \frac{e^{ix'(\lambda_k-\lambda_a)+it\left(\lambda_a^2-\lambda_k^2\right)}}{(\lambda_i-\lambda_a)(\lambda_j-\lambda_a)}. \tag{118}
$$

The two terms are of the form (49) and we apply (57) to express them in terms of the root density $\rho(\lambda)$ and the pair distribution function $\gamma_{-2}(\lambda)$ defined in (33), with a triple integral with successive principal values defined in (54). This yields

$$
\mathcal{A}_2 = -\frac{8}{c^2} \int_{-\Lambda}^{\Lambda} d\mu \rho_h(\mu) e^{-it\mu^2+ix'\mu} \int \frac{(\mu-\lambda)^2 \rho(\lambda)\rho(u)\rho(v)}{(u-\lambda)(\lambda-v)} e^{it\lambda^2-ix'\lambda} d\lambda du dv
$$
$$
- \frac{8}{c^2} \int_{-\infty}^{\infty} d\lambda \int_{-\Lambda}^{\Lambda} d\mu (\mu-\lambda)^2 \rho_h(\mu) \left[\frac{\pi^2}{3}\rho(\lambda)^3 - \gamma_{-2}(\lambda)\right] e^{it(\lambda^2-\mu^2)+ix'(\mu-\lambda)} d\lambda d\mu
$$
$$
+ \mathcal{O}(c^{-3}) + \mathcal{O}(L^{-1}). \tag{119}
$$

The definition of the successive principal value integral allows us to rewrite it in terms of $\tilde{\rho}$, to give

$$
\mathcal{A}_2 = \frac{8}{c^2} \int_{-\infty}^{\infty} d\lambda \rho(\lambda) \int_{-\Lambda}^{\Lambda} d\mu \rho_h(\mu) (\mu-\lambda)^2 \tilde{\rho}(\lambda)^2 e^{it(\lambda^2-\mu^2)+ix'(\mu-\lambda)}
$$
$$
- \frac{8}{c^2} \int_{-\infty}^{\infty} d\lambda \int_{-\Lambda}^{\Lambda} d\mu (\mu-\lambda)^2 \rho_h(\mu) \left[\frac{\pi^2}{3}\rho(\lambda)^3 - \gamma_{-2}(\lambda)\right] e^{it(\lambda^2-\mu^2)+ix'(\mu-\lambda)} d\lambda d\mu
$$
$$
+ \mathcal{O}(c^{-3}) + \mathcal{O}(L^{-1}). \tag{120}
$$

### 5.1.4 Order $c^{-2}$: second contribution

The second term on the right hand side of (116) is particularly cumbersome to deal with. We first treat the case $i = j$ separately, and for all terms with $i \neq j$ we apply a partial fraction decomposition with respect to $n$, so that we have only one $n$ appearing in the denominator. Finally we split the sum over $n$ as the difference of sums over vacancies and particles. Specif-

ically, we have for $f(u) = u^2 e^{ix'u + it(\lambda_a^2 - (\lambda_a + u)^2))}$

$$\sum_{\substack{i,j,n \\ \forall k, \lambda_{a,n} \neq \lambda_k \\ |\lambda_{a,n}| < \Lambda \\ i,j \neq a}} \frac{f(\frac{2\pi n}{L'})}{(\lambda_i - \lambda_a - \frac{2\pi n}{L'})(\lambda_j - \lambda_a - \frac{2\pi n}{L'})} = \sum_{\substack{i,n \\ \lambda_{a,n} \neq \lambda_i \\ |\lambda_{a,n}| < \Lambda \\ i \neq a}} \frac{f(\frac{2\pi n}{L'})}{(\lambda_i - \lambda_a - \frac{2\pi n}{L'})^2} - \sum_{\substack{i,k \\ i \neq k \\ |\lambda_k| < \Lambda \\ i \neq a}} \frac{f(\lambda_k - \lambda_a)}{(\lambda_i - \lambda_k)^2}$$

$$+ \sum_{\substack{i,j,n \\ \lambda_{a,n} \neq \lambda_i \\ |\lambda_{a,n}| < \Lambda \\ i,j \neq a, \, i \neq j}} \frac{f(\frac{2\pi n}{L'})}{(\lambda_j - \lambda_i)(\lambda_i - \lambda_a - \frac{2\pi n}{L'})} - \sum_{\substack{i,j,n \\ \lambda_{a,n} \neq \lambda_j \\ |\lambda_{a,n}| < \Lambda \\ i,j \neq a, \, i \neq j}} \frac{f(\frac{2\pi n}{L'})}{(\lambda_j - \lambda_i)(\lambda_j - \lambda_a - \frac{2\pi n}{L'})}$$

$$- \sum_{\substack{i,j,k \\ i \neq j, \, i \neq k \\ |\lambda_k| < \Lambda \\ i,j \neq a}} \frac{f(\lambda_k - \lambda_a)}{(\lambda_j - \lambda_i)(\lambda_i - \lambda_k)} + \sum_{\substack{i,j,k \\ i \neq j, \, j \neq k \\ |\lambda_k| < \Lambda \\ i,j \neq a}} \frac{f(\lambda_k - \lambda_a)}{(\lambda_j - \lambda_i)(\lambda_j - \lambda_k)}. \tag{121}$$

In all these terms, the conditions $i, j \neq a$ only give rise to subleading contributions in $L$, so that they can be discarded. The first term on the right hand side of (121) gives rise to a contribution to $\mathcal{C}_1^\Lambda(x,t)$ of the form

$$\mathcal{A}_3 = 8 \frac{(1 + \frac{2D}{c})^2}{c^2 L^4} \sum_a \sum_{\substack{n \\ \lambda_{a,n} \neq \lambda_i \\ |\lambda_{a,n}| < \Lambda}} \left(\frac{2\pi n}{L'}\right)^2 \sum_{\substack{i \\ i \neq a}} \frac{e^{ix'(\lambda_{a,n} - \lambda_a) + it(\lambda_a^2 - \lambda_{a,n}^2)}}{(\lambda_i - \lambda_a - \frac{2\pi n}{L'})^2}. \tag{122}$$

As this is proportional to $L^{-4}$ and only involves three sums the dominant contribution arises from the double pole. Using $\sum_{n \neq 0} \frac{1}{n^2} = \frac{\pi^2}{3}$ for the sum over $n$, we obtain

$$\mathcal{A}_3 = \frac{8}{c^2} \int_{-\infty}^{\infty} (\lambda - \mu)^2 \rho(\lambda) \frac{\pi^2}{3} \frac{\rho(\mu)}{(2\pi)^2} e^{it(\lambda^2 - \mu^2) + ix'(\mu - \lambda)} d\lambda d\mu + \mathcal{O}(c^{-3}) + \mathcal{O}(L^{-1}). \tag{123}$$

### 5.1.5 Order $c^{-2}$: third contribution

The second term on the right hand side of (121) gives rise to a contribution

$$\mathcal{A}_4 = -8 \frac{(1 + \frac{2D}{c})^2}{c^2 L^4} \sum_a \sum_{\substack{i,k \\ i,k \neq a \\ i \neq k}} \frac{(\lambda_k - \lambda_a)^2}{(\lambda_i - \lambda_k)^2} e^{ix'(\lambda_k - \lambda_a) + it(\lambda_a^2 - \lambda_k^2)}. \tag{124}$$

The sum is of the form (32) and according to (33) in the thermodynamic limit gives rise to integrals over the pair distribution function

$$\mathcal{A}_4 = -\frac{8}{c^2} \int_{-\infty}^{\infty} (\lambda - \mu)^2 \rho(\lambda) \gamma_{-2}(\mu) e^{it(\lambda^2 - \mu^2) + ix'(\mu - \lambda)} d\lambda d\mu + \mathcal{O}(c^{-3}) + \mathcal{O}(L^{-1}). \tag{125}$$

### 5.1.6 Order $c^{-2}$: fourth contribution

The third and fourth terms in (121) are "hybrid" terms mixing sums over $\lambda_i$'s and sums over regularly distributed $n$'s. They give rise to a contribution to $\mathcal{C}_1^\Lambda(x,t)$ of the form

$$\mathcal{A}_5 = 16 \frac{(1 + \frac{2D}{c})^2}{c^2 L^4} \sum_a \sum_{\substack{i,j \\ i,j \neq a \\ i \neq j}} \sum_{\substack{n \\ \lambda_{a,n} \neq \lambda_i \\ |\lambda_{a,n}| < \Lambda}} \left(\frac{2\pi n}{L'}\right)^2 \frac{e^{ix'(\lambda_{a,n} - \lambda_a) + it(\lambda_a^2 - \lambda_{a,n}^2)}}{(\lambda_i - \lambda_a - \frac{2\pi n}{L'})(\lambda_j - \lambda_i)}. \tag{126}$$

By symmetrizing over $i, j$, one obtains a sole pole in $n$, but since $n$ is regularly distributed and avoids only the pole one can convert the sum into a principal value integral. This leads to integrals with two successive principal values

$$
\mathcal{A}_5 = \frac{16}{c^2} \int_{-\infty}^{\infty} \mathrm{d}\lambda \rho(\lambda) \int_{-\Lambda}^{\Lambda} \mathrm{d}\mu \frac{1}{2\pi} (\mu - \lambda)^2 e^{it(\lambda^2 - \mu^2) + ix'(\mu - \lambda)} \fint \mathrm{d}u \frac{\rho(u)}{u - \mu} \fint \mathrm{d}v \frac{\rho(v)}{v - u}
$$
$$
+ \mathcal{O}(c^{-3}) + \mathcal{O}(L^{-1}).
\tag{127}
$$

This can be simplified further by expressing the rightmost double integral in terms of $\tilde{\rho}(\lambda)$. To that end, let us consider the integral of this term as a function of $\mu$ with an arbitrary continuous function $\varphi(\mu)$. Using (53) we have

$$
\int \mathrm{d}\mu \varphi(\mu) \fint \mathrm{d}u \frac{\rho(u)}{\mu - u} \fint \mathrm{d}v \frac{\rho(v)}{u - v} = \fiint \frac{\varphi(\mu)\rho(u)\rho(v)}{(\mu - u)(u - v)} \mathrm{d}\mu \mathrm{d}u \mathrm{d}v - \frac{\pi^2}{3} \int \varphi(\mu)\rho(\mu)^2 \mathrm{d}\mu.
\tag{128}
$$

Under the simultaneous principal value triple integral it is legitimate to decompose $\frac{1}{(\mu-u)(u-v)} = \frac{1}{\mu-v}(\frac{1}{\mu-u} + \frac{1}{u-v})$ and split the integral into two since $|\mu - v| > \epsilon$:

$$
\fiint \frac{\varphi(\mu)\rho(u)\rho(v)}{(\mu - u)(u - v)} \mathrm{d}\mu \mathrm{d}u \mathrm{d}v = \fiint \frac{\varphi(\mu)\rho(u)\rho(v)}{(\mu - v)(\mu - u)} \mathrm{d}\mu \mathrm{d}u \mathrm{d}v + \fiint \frac{\varphi(\mu)\rho(u)\rho(v)}{(\mu - v)(u - v)} \mathrm{d}\mu \mathrm{d}u \mathrm{d}v.
\tag{129}
$$

We then use (53) to express the two simultaneous principal value triple integrals in terms of successive principal value integrals

$$
\int \mathrm{d}\mu \varphi(\mu) \fint \mathrm{d}u \frac{\rho(u)}{\mu - u} \fint \mathrm{d}v \frac{\rho(v)}{u - v} = \int \mathrm{d}\mu \varphi(\mu) \fint \mathrm{d}v \frac{\rho(v)}{\mu - v} \fint \mathrm{d}u \frac{\rho(u)}{\mu - u} - \pi^2 \int \varphi(\mu)\rho(\mu)^2 \mathrm{d}\mu
$$
$$
+ \int \mathrm{d}\mu \varphi(\mu) \fint \mathrm{d}v \frac{\rho(v)}{\mu - v} \fint \mathrm{d}u \frac{\rho(u)}{u - v}.
\tag{130}
$$

The first integral on the right hand side is $\int \varphi(\mu)\tilde{\rho}(\mu)^2 \mathrm{d}\mu$ while the third equals minus the left hand side. Using that this identity holds for any continuous function $\varphi(\mu)$ we conclude that

$$
\fint \mathrm{d}\lambda \frac{\rho(\lambda)}{\mu - \lambda} \fint \mathrm{d}u \frac{\rho(u)}{\lambda - u} = \frac{1}{2} \tilde{\rho}(\mu)^2 - \frac{\pi^2}{2} \rho(\mu)^2.
\tag{131}
$$

Putting everything together we obtain

$$
\mathcal{A}_5 = \frac{8}{c^2} \int_{-\infty}^{\infty} \mathrm{d}\lambda \rho(\lambda) \int_{-\Lambda}^{\Lambda} \mathrm{d}\mu \frac{1}{2\pi} (\mu - \lambda)^2 \tilde{\rho}(\mu)^2 e^{it(\lambda^2 - \mu^2) + ix'(\mu - \lambda)}
$$
$$
- \frac{8\pi^2}{c^2} \int_{-\infty}^{\infty} \mathrm{d}\lambda \rho(\lambda) \int_{-\Lambda}^{\Lambda} \mathrm{d}\mu \frac{1}{2\pi} (\mu - \lambda)^2 \rho(\mu)^2 e^{it(\lambda^2 - \mu^2) + ix'(\mu - \lambda)} + \mathcal{O}(c^{-3}) + \mathcal{O}(L^{-1}).
\tag{132}
$$

### 5.1.7 Order $c^{-2}$: fifth contribution

The fifth and fourth terms on the right hand side of (121) are of the form (49) and give rise to a contribution

$$
\mathcal{A}_6 = -16 \frac{(1 + \frac{2D}{c})^2}{c^2 L^4} \sum_{a} \sum_{\substack{i,j,k \\ i,j,k \neq a \\ i \neq j, \ i \neq k}} \frac{(\lambda_k - \lambda_a)^2}{(\lambda_j - \lambda_i)(\lambda_i - \lambda_k)} e^{ix'(\lambda_k - \lambda_a) + it(\lambda_a^2 - \lambda_k^2)}.
\tag{133}
$$

Applying (57) with successive principal values and then using (131) we find

$$
\begin{aligned}
\mathcal{A}_6 = {} & -\frac{8}{c^2} \int_{-\infty}^{\infty} \mathrm{d}\lambda \rho(\lambda) \int_{-\Lambda}^{\Lambda} \mathrm{d}\mu \rho(\mu)(\mu-\lambda)^2 \tilde{\rho}(\mu)^2 e^{it(\lambda^2-\mu^2)+ix'(\mu-\lambda)} \\
& + \frac{8\pi^2}{c^2} \int_{-\infty}^{\infty} \mathrm{d}\lambda \rho(\lambda) \int_{-\Lambda}^{\Lambda} \mathrm{d}\mu (\mu-\lambda)^2 \rho(\mu)^3 e^{it(\lambda^2-\mu^2)+ix'(\mu-\lambda)} \\
& - \frac{16}{c^2} \int_{-\infty}^{\infty} \int_{-\infty}^{\infty} (\lambda-\mu)^2 \rho(\lambda)\left[\frac{\pi^2}{3}\rho(\mu)^3 - \gamma_{-2}(\mu)\right] e^{it(\lambda^2-\mu^2)+ix'(\mu-\lambda)} \mathrm{d}\lambda \mathrm{d}\mu \\
& + \mathcal{O}(c^{-3}) + \mathcal{O}(L^{-1}).
\end{aligned}
\tag{134}
$$

### 5.1.8 Order $c^{-2}$: sixth contribution

Finally, the last term in (116) gives rise to a contribution

$$
\mathcal{A}_7 = -16 \frac{(1+\frac{2D}{c})^2}{c^2 L^4} \sum_a \sum_{\substack{\forall k,\, \lambda_{a,n}\neq\lambda_k \\ n \\ |\lambda_{a,n}|<\Lambda}} \left(\frac{2\pi n}{L'}\right)^2 \sum_{\substack{i,j \\ i\neq a \\ j\neq a}} \frac{e^{ix'(\lambda_{a,n}-\lambda_a)+it\left(\lambda_a^2-\lambda_{a,n}^2\right)}}{(\lambda_i-\lambda_a-\frac{2\pi n}{L'})(\lambda_j-\lambda_a)}.
\tag{135}
$$

By again decomposing the sum over $n$ as a sum over vacancies minus a sum over particles we find

$$
\begin{aligned}
\mathcal{A}_7 = {} & -\frac{16}{c^2} \int_{-\infty}^{\infty} \mathrm{d}\lambda \rho(\lambda) \int_{-\Lambda}^{\Lambda} \mathrm{d}\mu \rho_h(\mu)(\mu-\lambda)^2 \tilde{\rho}(\lambda)\tilde{\rho}(\mu) e^{it(\lambda^2-\mu^2)+ix'(\mu-\lambda)} \\
& + \mathcal{O}(c^{-3}) + \mathcal{O}(L^{-1}).
\end{aligned}
\tag{136}
$$

### 5.1.9 Result for the contribution of one particle-hole excitations

We leave the remaining contributions to $\mathcal{C}_1^\Lambda(x,t)$ untouched, i.e. in sum form, since they will be cancelled by contributions from two particle-hole excitations to the correlator. Our final result for $\mathcal{C}_1^\Lambda(x,t)$ is thus given by

$$
\begin{aligned}
\mathcal{C}_1^\Lambda(x,t) = {} & \Omega_1^\Lambda + (1+\tfrac{2D}{c})^2 \int_{-\infty}^{\infty} \mathrm{d}\lambda\, \rho(\lambda) \int_{-\Lambda}^{\Lambda} \mathrm{d}\mu\, \rho_h(\mu)\left[1 - \frac{4}{c}(\mu-\lambda)(\tilde{\rho}(\mu)-\tilde{\rho}(\lambda))\right. \\
& \left. + \frac{8}{c^2}(\mu-\lambda)^2(\tilde{\rho}(\mu)-\tilde{\rho}(\lambda))^2 - \frac{8\pi^2}{c^2}(\mu-\lambda)^2[\rho(\mu)]^2\right] e^{it(\lambda^2-\mu^2)+ix'(\mu-\lambda)} \\
& -\frac{8}{c^2} \int_{-\infty}^{\infty} \mathrm{d}\lambda \int_{-\Lambda}^{\Lambda} \mathrm{d}\mu (\lambda-\mu)^2 \rho_h(\mu)\left[\frac{\pi^2}{3}[\rho(\lambda)]^3 - \gamma_{-2}(\lambda)\right] e^{it(\lambda^2-\mu^2)+ix'(\mu-\lambda)} \\
& +\frac{8}{c^2} \int_{-\infty}^{\infty} \mathrm{d}\lambda \int_{-\infty}^{\infty} \mathrm{d}\mu (\lambda-\mu)^2 \rho(\lambda)\left[\frac{\rho(\mu)}{12} - \frac{2\pi^2}{3}[\rho(\mu)]^3 + \gamma_{-2}(\mu)\right] e^{it(\lambda^2-\mu^2)+ix'(\mu-\lambda)} \\
& + \mathcal{O}(c^{-3}) + \mathcal{O}(L^{-1}),
\end{aligned}
\tag{137}
$$

where we have defined

$$
\begin{aligned}
\Omega_1^\Lambda = {} & -\frac{1}{c^2 L^2} \sum_a \sum_{\substack{\forall k,\, \lambda_{a,n}\neq\lambda_k \\ |\lambda_{a,n}|<\Lambda}} \left[\frac{1}{3}\sum_{\substack{i \\ i\neq a}} 1 - \frac{4}{L^2}\sum_{\substack{i,j \\ i\neq j \\ j\neq a}} \frac{1}{(\lambda_i-\lambda_j)^2} - \frac{4}{L^2}\sum_{\substack{i \\ i\neq a}} \frac{1}{(\lambda_i-\lambda_a-\frac{2\pi n}{L'})^2}\right] \\
& \times \left(\frac{2\pi n}{L'}\right)^2 e^{it[\lambda_a^2-\lambda_{a,n}^2]+ix'[\lambda_{a,n}-\lambda_a]}.
\end{aligned}
\tag{138}
$$

## 5.2 Two-particle-hole excitations

### 5.2.1 A partial fraction decomposition

The computation of $\mathcal{C}_2^\Lambda(x,t)$ defined in (101) is slightly different. In order to proceed we decompose the form factor into partial fractions with respect to $n$, and then $m$:

$$
\frac{(n+m)^4}{n^2 m^2} \frac{(\lambda_a - \lambda_b)^2 (\lambda_a - \lambda_b + \frac{2\pi(n-m)}{L(1+\frac{2D}{c})})^2}{(\lambda_a - \lambda_b + \frac{2\pi n}{L(1+\frac{2D}{c})})^2 (\lambda_a - \lambda_b - \frac{2\pi m}{L(1+\frac{2D}{c})})^2} =
$$
$$
\left(\frac{2\pi n}{L'}\right)^2 \left[ \frac{1}{(\frac{2\pi m}{L'})^2} + \frac{2}{(\lambda_a - \lambda_b)\frac{2\pi m}{L'}} + \frac{1}{(\lambda_a - \lambda_b - \frac{2\pi m}{L'})^2} + \frac{2}{(\lambda_a - \lambda_b)(\lambda_a - \lambda_b - \frac{2\pi m}{L'})} \right]
$$
$$
+ \frac{2\pi n}{L'} \left[ \frac{2}{\frac{2\pi m}{L'}} + \frac{2(\lambda_a - \lambda_b)}{(\lambda_a - \lambda_b - \frac{2\pi m}{L'})^2} + \frac{2}{\lambda_a - \lambda_b - \frac{2\pi m}{L'}} \right]
$$
$$
+ \left[ \frac{2(\lambda_a - \lambda_b)}{\frac{2\pi m}{L'}} + \frac{(\lambda_a - \lambda_b)^2}{(\lambda_a - \lambda_b - \frac{2\pi m}{L'})^2} + \frac{2(\lambda_a - \lambda_b)}{\lambda_a - \lambda_b - \frac{2\pi m}{L'}} \right]
$$
$$
+ \left(\frac{2\pi n}{L'}\right)^{-1} \left[ -2(\lambda_a - \lambda_b) + 2\frac{2\pi m}{L'} - \frac{2(\frac{2\pi m}{L'})^2}{\lambda_a - \lambda_b} + \frac{2(\lambda_a - \lambda_b)^2}{\lambda_a - \lambda_b - \frac{2\pi m}{L'}} \right]
$$
$$
+ \left(\frac{2\pi n}{L'}\right)^{-2} \left(\frac{2\pi m}{L'}\right)^2
$$
$$
+ \left(\lambda_a - \lambda_b + \frac{2\pi n}{L'}\right)^{-1} \left[ 2(\lambda_a - \lambda_b) - \frac{2(\lambda_a - \lambda_b)^2}{\frac{2\pi m}{L'}} - 2\frac{2\pi m}{L'} + \frac{2(\frac{2\pi m}{L'})^2}{\lambda_a - \lambda_b} \right]
$$
$$
+ \left(\lambda_a - \lambda_b + \frac{2\pi n}{L'}\right)^{-2} \left[ (\lambda_a - \lambda_b)^2 - 2(\lambda_a - \lambda_b)\frac{2\pi m}{L'} + \left(\frac{2\pi m}{L'}\right)^2 \right].
\tag{139}
$$

We now use that the sum is invariant under the simultaneous reparametrisations $n' = m - \frac{L'(\lambda_a - \lambda_b)}{2\pi}$ and $m' = n + \frac{L'(\lambda_a - \lambda_b)}{2\pi}$ (which corresponds to swapping the position of the two excited particles) to bring all the poles into poles in $n$ or $m$, the only exceptions being $[\frac{2\pi n}{L'}(\lambda_a - \lambda_b - \frac{2\pi m}{L'})]^{-1}$ and $[\frac{2\pi n}{L'}(\lambda_a - \lambda_b + \frac{2\pi n}{L'})]^{-1}$ which cannot be transformed further. Next we use the invariance under swapping $n, m$ and $a, b$ simultaneously (which corresponds to renaming dummy variables) to bring all the poles into poles in $m$ only, with the exception of $[\frac{2\pi m}{L'}(\lambda_a - \lambda_b + \frac{2\pi n}{L'})]^{-1}$. We obtain

$$
\mathcal{C}_2^\Lambda(x,t) = \frac{4}{c^2 L^4} \sum_{a \neq b} \sum_{\substack{n \\ \forall i, \lambda_{a,n} \neq \lambda_i \\ |\lambda_{a,n}| < \Lambda}} \sum_{\substack{m \\ \forall j, \lambda_{b,m} \neq \lambda_j \\ \lambda_{b,m} \neq \lambda_{a,n} \\ |\lambda_{b,m}| < \Lambda}} e^{ix'[\lambda_{a,n} - \lambda_a + \lambda_{b,m} - \lambda_b] + it[\lambda_a^2 - \lambda_{a,n}^2 + \lambda_b^2 - \lambda_{b,m}^2]}
$$
$$
\times \left[ \frac{n^2}{m^2} + 2\frac{(\frac{2\pi n}{L'})^2}{(\lambda_a - \lambda_b)\frac{2\pi m}{L'}} + 2\frac{n}{m} + 2\frac{\lambda_a - \lambda_b}{\frac{2\pi m}{L'}} - \frac{(\lambda_a - \lambda_b)^2}{\frac{2\pi m}{L'}(\lambda_a - \lambda_b + \frac{2\pi n}{L'})} \right].
\tag{140}
$$

We now carry out the sums over $m$ and $b$ in order to bring this to a form similar to the contribution from one particle hole excitations. We will denote the resulting five terms by $\Sigma_i$ for $i = 1, \ldots, 5$ and treat them one at a time.

### 5.2.2  First term $\Sigma_1$

In this subsection we take the thermodynamic limit of

$$\Sigma_1 = \frac{4}{c^2 L^4} \sum_{a \neq b} \sum_{\substack{n \\ \forall k, \lambda_{a,n} \neq \lambda_k \\ |\lambda_{a,n}| < \Lambda}} \sum_{\substack{m \\ \forall i, \lambda_{b,m} \neq \lambda_i \\ \lambda_{b,m} \neq \lambda_{a,n} \\ |\lambda_{b,m}| < \Lambda}} \frac{n^2}{m^2} e^{it\left[\lambda_a^2 - \lambda_{a,n}^2 + \lambda_b^2 - \lambda_{b,m}^2\right] + ix'\left[\lambda_{a,n} - \lambda_a + \lambda_{b,m} - \lambda_b\right]}. \tag{141}$$

We begin by splitting the exponential factor

$$e^{it\left[\lambda_b^2 - \lambda_{b,m}^2\right] + ix'\left[\lambda_{b,m} - \lambda_b\right]} = \left(e^{it\left[\lambda_b^2 - \lambda_{b,m}^2\right] + ix'\left[\lambda_{b,m} - \lambda_b\right]} - 1\right) + 1. \tag{142}$$

Performing the sum over $m$ for the second term in (142) gives

$$\Sigma_1 = \tilde{\Sigma}_1 + \frac{1}{c^2 L^2} \sum_a \sum_{\substack{n \\ \forall k, \lambda_{a,n} \neq \lambda_k \\ |\lambda_{a,n}| < \Lambda}} \left(\frac{2\pi n}{L'}\right)^2 e^{it[\lambda_a^2 - \lambda_{a,n}^2] + ix'[\lambda_{a,n} - \lambda_a]}$$

$$\times \left[\frac{1}{3} \sum_{\substack{i \\ i \neq a}} 1 - \frac{4}{L^2} \left\{ \sum_{\substack{i,j \\ i \neq j,\, j \neq a \\ |\lambda_j| < \Lambda}} \frac{1}{(\lambda_i - \lambda_j)^2} + \sum_{\substack{j \\ j \neq a}} \frac{1}{(\lambda_a - \lambda_j + \frac{2\pi n}{L'})^2} \right\} \right], \tag{143}$$

where

$$\tilde{\Sigma}_1 = \frac{4}{c^2 L^4} \sum_{a \neq b} \sum_{\substack{n \\ \forall k, \lambda_{a,n} \neq \lambda_k \\ |\lambda_{a,n}| < \Lambda}} \sum_{\substack{m \\ \forall i, \lambda_{b,m} \neq \lambda_i \\ \lambda_{b,m} \neq \lambda_{a,n} \\ |\lambda_{b,m}| < \Lambda}} \frac{n^2}{m^2} e^{it\left[\lambda_a^2 - \lambda_{a,n}^2\right] + ix'\left[\lambda_{a,n} - \lambda_a\right]} \left[e^{it\left[\lambda_b^2 - \lambda_{b,m}^2\right] + ix'\left[\lambda_{b,m} - \lambda_b\right]} - 1\right]. \tag{144}$$

The advantage of this representation is that the pole in $m$ is now only of order 1. Writing the sum over $m$ as a sum over $m \neq 0$ minus sums over particles one obtains

$$\tilde{\Sigma}_1 = \frac{4}{c^2 L^4} \sum_{a \neq b} \sum_{\substack{n \\ \forall k, \lambda_{a,n} \neq \lambda_k \\ |\lambda_{a,n}| < \Lambda}} \sum_{\substack{m \neq 0 \\ |\lambda_{b,m}| < \Lambda}} \frac{n^2}{m^2} \left[e^{it\left[\lambda_b^2 - \lambda_{b,m}^2\right] + ix'\left[\lambda_{b,m} - \lambda_b\right]} - 1\right] e^{it\left[\lambda_a^2 - \lambda_{a,n}^2\right] + ix'\left[\lambda_{a,n} - \lambda_a\right]}$$

$$- \frac{4}{c^2 L^4} \sum_{a \neq b} \sum_{\substack{n \\ \forall k, \lambda_{a,n} \neq \lambda_k \\ |\lambda_{a,n}| < \Lambda}} \sum_{\substack{i \\ i \neq b \\ |\lambda_i| < \Lambda}} \frac{\left(\frac{2\pi n}{L'}\right)^2}{(\lambda_b - \lambda_i)^2} \left[e^{it\left[\lambda_b^2 - \lambda_i^2\right] + ix'\left[\lambda_i - \lambda_b\right]} - 1\right] e^{it\left[\lambda_a^2 - \lambda_{a,n}^2\right] + ix'\left[\lambda_{a,n} - \lambda_a\right]}$$

$$- \frac{4}{c^2 L^4} \sum_{a \neq b} \sum_{\substack{n \\ \forall k, \lambda_{a,n} \neq \lambda_k \\ |\lambda_{a,n}| < \Lambda}} \frac{\left(\frac{2\pi n}{L'}\right)^2}{(\lambda_b - \lambda_a - \frac{2\pi n}{L'})^2} \left[e^{it\left[\lambda_b^2 - \lambda_{a,n}^2\right] + ix'\left[\lambda_{a,n} - \lambda_b\right]} - 1\right]$$

$$\times e^{it\left[\lambda_a^2 - \lambda_{a,n}^2\right] + ix'\left[\lambda_{a,n} - \lambda_a\right]}. \tag{145}$$

The first term is a sum over regularly spaced integers $m$ with only a simple pole. In the thermodynamic limit it can therefore be expressed in terms of a principal value integral with a constant density $\frac{1 + \frac{2D}{c}}{2\pi}$. The second term is of type (43) and gives rise to an integral over the root density $\rho(\lambda)$ in the thermodynamic limit. The last term is negligible in $L$. We find

$$\Sigma_1 = \Omega_2^\Lambda + \frac{4A_{x',t}}{c^2} \int_{-\infty}^{\infty} d\lambda \rho(\lambda) \int_{-\Lambda}^{\Lambda} d\mu \rho_h(\mu)(\mu-\lambda)^2 e^{it(\lambda^2-\mu^2)+ix'(\mu-\lambda)}$$
$$+ \mathcal{O}(\Lambda^{-1}L^0) + \mathcal{O}(L^{-1}), \tag{146}$$

where we defined

$$A_{x,t} = \int_{-\infty}^{\infty} du \int_{-\infty}^{\infty} dv \frac{\rho(u)\rho_h(v)}{(u-v)^2}(e^{it(u^2-v^2)+ix(v-u)}-1), \tag{147}$$

and

$$\Omega_2^\Lambda = \frac{1}{c^2L^2} \sum_a \sum_{\substack{n \\ \forall k, \lambda_{a,n}\neq\lambda_k \\ |\lambda_{a,n}|<\Lambda}} \left(\frac{2\pi n}{L}\right)^2 e^{it[\lambda_a^2-\lambda_{a,n}^2]+ix'[\lambda_{a,n}-\lambda_a]}$$
$$\times \left[\frac{1}{3}\sum_{\substack{i \\ i\neq a}} 1 - \frac{4}{L'^2}\left\{\sum_{\substack{i,j \\ i\neq j, \, j\neq a \\ |\lambda_j|<\Lambda}} \frac{1}{(\lambda_i-\lambda_j)^2} + \sum_{\substack{j \\ j\neq a}} \frac{1}{(\lambda_a-\lambda_j+\frac{2\pi n}{L'})^2}\right\}\right]. \tag{148}$$

### 5.2.3 Second term $\Sigma_2$

The next contribution is

$$\Sigma_2 = \frac{4}{c^2L^4} \sum_{a\neq b} \sum_{\substack{n \\ \forall k, \lambda_{a,n}\neq\lambda_k \\ |\lambda_{a,n}|<\Lambda}} \sum_{\substack{m \\ \forall i, \lambda_{b,m}\neq\lambda_i \\ \lambda_{b,m}\neq\lambda_{a,n} \\ |\lambda_{b,m}|<\Lambda}} \frac{2(\frac{2\pi n}{L'})^2}{(\lambda_a-\lambda_b)\frac{2\pi m}{L'}} e^{it\left[\lambda_a^2-\lambda_{a,n}^2+\lambda_b^2-\lambda_{b,m}^2\right]+ix'[\lambda_{a,n}-\lambda_a+\lambda_{b,m}-\lambda_b]}. \tag{149}$$

Writing the sum over $m$ again as sums over vacancies minus particles we have

$$\Sigma_2 = \frac{8}{c^2L^4} \sum_{a\neq b} \sum_{\substack{n \\ \forall k, \lambda_{a,n}\neq\lambda_k \\ |\lambda_{a,n}|<\Lambda}} \sum_{\substack{m\neq 0 \\ |\lambda_{b,m}|<\Lambda}} \frac{(\frac{2\pi n}{L'})^2}{(\lambda_a-\lambda_b)\frac{2\pi m}{L'}} e^{it\left[\lambda_a^2-\lambda_{a,n}^2+\lambda_b^2-\lambda_{b,m}^2\right]+ix'[\lambda_{a,n}-\lambda_a+\lambda_{b,m}-\lambda_b]}$$
$$- \frac{8}{c^2L^4} \sum_{a\neq b} \sum_{\substack{n \\ \forall k, \lambda_{a,n}\neq\lambda_k \\ |\lambda_{a,n}|<\Lambda}} \sum_{\substack{i \\ i\neq b \\ |\lambda_i|<\Lambda}} \frac{(\frac{2\pi n}{L'})^2}{(\lambda_a-\lambda_b)(\lambda_i-\lambda_b)} e^{it\left[\lambda_a^2-\lambda_{a,n}^2+\lambda_b^2-\lambda_i^2\right]+ix'[\lambda_{a,n}-\lambda_a+\lambda_i-\lambda_b]}$$
$$- \frac{8}{c^2L^4} \sum_{a\neq b} \sum_{\substack{n \\ \forall k, \lambda_{a,n}\neq\lambda_k \\ |\lambda_{a,n}|<\Lambda}} \frac{(\frac{2\pi n}{L'})^2}{(\lambda_a-\lambda_b)(\frac{2\pi n}{L'}+\lambda_a-\lambda_b)} e^{it\left[\lambda_a^2-2\lambda_{a,n}^2+\lambda_b^2\right]+ix'[2\lambda_{a,n}-\lambda_a-\lambda_b]}. \tag{150}$$

In the first two lines of (150) the sums over $n$ are regular. The first line involves only sums of the form (43), while the second line is of the form (49) and the thermodynamic limit can be worked out using (57). The third term, after splitting the sum over $n$ as sums over vacancies

minus particles is seen to be negligible in $L$. Hence we obtain

$$
\begin{aligned}
\Sigma_2 = & \frac{8}{c^2} \int_{-\infty}^{\infty} d\lambda \rho(\lambda) \int_{-\Lambda}^{\Lambda} d\mu \rho_h(\mu)(\mu-\lambda)^2 B_{x',t}(\lambda) e^{it(\lambda^2-\mu^2)+ix'(\mu-\lambda)} \\
& + \frac{8}{c^2} \int_{-\infty}^{\infty} d\lambda \rho(\lambda) \int_{-\Lambda}^{\Lambda} d\mu \rho_h(\mu)(\mu-\lambda)^2 \left[ \frac{\pi^2}{3}\rho(\lambda)^3 - \gamma_{-2}(\lambda) \right] e^{it(\lambda^2-\mu^2)+ix'(\mu-\lambda)} \\
& + \mathcal{O}(\Lambda^{-1}L^0) + \mathcal{O}(L^{-1}),
\end{aligned} \tag{151}
$$

where $B_{x,t}(\lambda)$ is defined in terms of principal value integrals by

$$
B_{x,t}(\lambda) = \int_{-\infty}^{\infty} du \fint_{-\infty}^{\infty} dv \frac{\rho(u)\rho_h(v)}{(v-u)(\lambda-u)} e^{it(u^2-v^2)+ix(v-u)}. \tag{152}
$$

### 5.2.4  Third term $\Sigma_3$

In this subsection we take the thermodynamic limit of

$$
\Sigma_3 = \frac{4}{c^2 L^4} \sum_{a \neq b} \sum_{\substack{\forall k, \lambda_{a,n} \neq \lambda_k \\ |\lambda_{a,n}| < \Lambda}}^{n} \sum_{\substack{\forall i, \lambda_{b,m} \neq \lambda_i \\ \lambda_{b,m} \neq \lambda_{a,n} \\ |\lambda_{b,m}| < \Lambda}}^{m} 2\frac{n}{m} e^{it\left[\lambda_a^2-\lambda_{a,n}^2+\lambda_b^2-\lambda_{b,m}^2\right]+ix'[\lambda_{a,n}-\lambda_a+\lambda_{b,m}-\lambda_b]}. \tag{153}
$$

Expressing the sum over $m$ as the difference of sums over vacancies and particles $\Sigma_3$ reduces to terms of the form (43) that can be readily expressed as integrals over root densities. We obtain

$$
\Sigma_3 = \frac{8C_{x',t}}{c^2} \int_{-\infty}^{\infty} d\lambda \rho(\lambda) \int_{-\Lambda}^{\Lambda} d\mu \rho_h(\mu)(\mu-\lambda) e^{it(\lambda^2-\mu^2)+ix'(\mu-\lambda)} + \mathcal{O}(\Lambda^{-1}L^0) + \mathcal{O}(L^{-1}), \tag{154}
$$

where we have defined

$$
C_{x,t} = \int_{-\infty}^{\infty} du \fint_{-\infty}^{\infty} dv \frac{\rho(u)\rho_h(v)}{v-u} e^{it(u^2-v^2)+ix(v-u)}. \tag{155}
$$

### 5.2.5  Fourth term $\Sigma_4$

The next contribution is given by

$$
\Sigma_4 = \frac{4}{c^2 L^4} \sum_{a \neq b} \sum_{\substack{\forall k, \lambda_{a,n} \neq \lambda_k \\ |\lambda_{a,n}| < \Lambda}}^{n} \sum_{\substack{\forall i, \lambda_{b,m} \neq \lambda_i \\ \lambda_{b,m} \neq \lambda_{a,n} \\ |\lambda_{b,m}| < \Lambda}}^{m} 2\frac{\lambda_a-\lambda_b}{\frac{2\pi m}{L'}} e^{it\left[\lambda_a^2-\lambda_{a,n}^2+\lambda_b^2-\lambda_{b,m}^2\right]+ix'[\lambda_{a,n}-\lambda_a+\lambda_{b,m}-\lambda_b]}, \tag{156}
$$

and can be treated in complete analogy with $\Sigma_3$. We obtain

$$
\Sigma_4 = \frac{8}{c^2} \int_{-\infty}^{\infty} d\lambda \rho(\lambda) \int_{-\Lambda}^{\Lambda} d\mu \rho_h(\mu)(\lambda C_{x',t} - D_{x',t}) e^{it(\lambda^2-\mu^2)+ix'(\mu-\lambda)} + \mathcal{O}(\Lambda^{-1}L^0) + \mathcal{O}(L^{-1}), \tag{157}
$$

where we have defined

$$
D_{x,t} = \int_{-\infty}^{\infty} du \fint_{-\infty}^{\infty} dv\, u \frac{\rho(u)\rho_h(v)}{v-u} e^{it(u^2-v^2)+ix(v-u)}. \tag{158}
$$

### 5.2.6  Fifth term $\Sigma_5$

The final contribution to $\mathcal{C}_2^\Lambda(x,t)$ is

$$
\Sigma_5 = -\frac{4}{c^2 L^4} \sum_{a\neq b} \sum_{\substack{\forall k,\,\lambda_{a,n}\neq\lambda_k \\ n \\ |\lambda_{a,n}|<\Lambda}} \sum_{\substack{\forall i,\,\lambda_{b,m}\neq\lambda_i \\ m \\ \lambda_{b,m}\neq\lambda_{a,n} \\ |\lambda_{b,m}|<\Lambda}} \frac{(\lambda_a-\lambda_b)^2}{\frac{2\pi m}{L'}(\lambda_a-\lambda_b+\frac{2\pi n}{L'})}
$$
$$
\times\, e^{it\left[\lambda_a^2-\lambda_{a,n}^2+\lambda_b^2-\lambda_{b,m}^2\right]+ix'[\lambda_{a,n}-\lambda_a+\lambda_{b,m}-\lambda_b]}. \tag{159}
$$

In order to take the thermodynamic limit we rewrite this as

$$
\Sigma_5 = -\frac{4}{c^2 L^4} \sum_{a\neq b} \sum_{\substack{\forall k,\,\lambda_{a,n}\neq\lambda_k \\ n \\ |\lambda_{a,n}|<\Lambda}} \sum_{\substack{m\neq 0 \\ |\lambda_{b,m}|<\Lambda}} \frac{(\lambda_a-\lambda_b)^2\, e^{it\left[\lambda_a^2-\lambda_{a,n}^2+\lambda_b^2-\lambda_{b,m}^2\right]+ix'[\lambda_{a,n}-\lambda_a+\lambda_{b,m}-\lambda_b]}}{\frac{2\pi m}{L'}(\lambda_a-\lambda_b+\frac{2\pi n}{L'})}
$$

$$
+\frac{4}{c^2 L^4} \sum_{a\neq b} \sum_{\substack{\lambda_{a,n}\neq\lambda_b \\ n \\ |\lambda_{a,n}|<\Lambda}} \sum_{\substack{i\neq b \\ i \\ |\lambda_i|<\Lambda}} \frac{(\lambda_a-\lambda_b)^2\, e^{it\left[\lambda_a^2-\lambda_{a,n}^2+\lambda_b^2-\lambda_i^2\right]+ix'[\lambda_{a,n}-\lambda_a+\lambda_i-\lambda_b]}}{(\lambda_i-\lambda_b)(\lambda_a-\lambda_b+\frac{2\pi n}{L'})}
$$

$$
-\frac{4}{c^2 L^4} \sum_{a\neq b} \sum_{\substack{k\neq b \\ k \\ |\lambda_k|<\Lambda}} \sum_{\substack{i\neq b \\ i \\ |\lambda_i|<\Lambda}} \frac{(\lambda_a-\lambda_b)^2\, e^{it\left[\lambda_a^2-\lambda_k^2+\lambda_b^2-\lambda_i^2\right]+ix'[\lambda_i+\lambda_k-\lambda_a-\lambda_b]}}{(\lambda_i-\lambda_b)(\lambda_k-\lambda_b)}
$$

$$
+\frac{4}{c^2 L^4} \sum_{a\neq b} \sum_{\substack{\lambda_{a,n}\neq\lambda_b \\ n \\ |\lambda_{a,n}|<\Lambda}} \frac{(\lambda_a-\lambda_b)^2}{(\lambda_a-\lambda_b+\frac{2\pi n}{L'})^2}\, e^{it\left[\lambda_a^2-2\lambda_{a,n}^2+\lambda_b^2\right]+2ix'[2\lambda_{a,n}-\lambda_a-\lambda_b]}
$$

$$
-\frac{4}{c^2 L^4} \sum_{a\neq b} \sum_{\substack{k\neq b \\ k \\ |\lambda_k|<\Lambda}} \frac{(\lambda_a-\lambda_b)^2}{(\lambda_k-\lambda_b)^2}\, e^{it\left[\lambda_a^2-2\lambda_k^2+\lambda_b^2\right]+ix'[2\lambda_k-\lambda_a-\lambda_b]}. \tag{160}
$$

The first two lines are of type (43) while the third and fifth lines are of types (49) and (32) respectively. Finally, in the fourth line we use that $\sum_{n\neq 0}\frac{1}{n^2}=\frac{\pi^2}{3}$ to arrive at

$$
\Sigma_5 = \frac{4}{c^2} \int_{-\infty}^{\infty} d\lambda\rho(\lambda) \int_{-\Lambda}^{\Lambda} d\mu\rho_h(\mu)\Big[(\mu-2\lambda)C_{x',t}+D_{x',t}-(\lambda-\mu)^2 B_{x',t}(\mu)\Big]
$$
$$
\times\, e^{it(\lambda^2-\mu^2)+ix'(\mu-\lambda)}
$$
$$
+\frac{4}{c^2} \int_{-\infty}^{\infty} d\lambda\rho(\lambda) \int_{-\infty}^{\infty} d\mu(\lambda-\mu)^2\left[\frac{\pi^2}{3}\rho(\mu)^3+\frac{\pi^2}{3}\frac{\rho(\mu)}{(2\pi)^2}-2\gamma_{-2}(\mu)\right]e^{it(\lambda^2-\mu^2)+ix'(\mu-\lambda)}
$$
$$
+\mathcal{O}(\Lambda^{-1}L^0)+\mathcal{O}(L^{-1}). \tag{161}
$$

### 5.2.7 Result for the contribution from two particle-hole excitations

Combining the results of sections 5.2.2-5.2.6 we arrive at the following expression for the two particle-hole contribution to the density-density correlation function

$$
\begin{aligned}
C_2^\Lambda(x,t) = \Omega_2^\Lambda &+ \frac{4}{c^2} \int_{-\infty}^\infty d\lambda \rho(\lambda) \int_{-\Lambda}^\Lambda d\mu \rho_h(\mu) \Big[ (\mu-\lambda)^2 [A_{x',t} + 2B_{x',t}(\lambda) - B_{x',t}(\mu)] \\
&\qquad\qquad + (3\mu - 2\lambda) C_{x',t} - D_{x',t} \Big] e^{it(\lambda^2-\mu^2)+ix'(\mu-\lambda)} \\
&+ \frac{8}{c^2} \int_{-\infty}^\infty d\lambda \rho(\lambda) \int_{-\Lambda}^\Lambda d\mu \rho_h(\mu)(\lambda-\mu)^2 \left[ \frac{\pi^2}{3} \rho(\lambda)^3 - \gamma_{-2}(\lambda) \right] e^{it(\lambda^2-\mu^2)+ix'(\mu-\lambda)} \\
&+ \frac{4}{c^2} \int_{-\infty}^\infty d\lambda \rho(\lambda) \int_{-\infty}^\infty d\mu (\lambda-\mu)^2 \left[ \frac{\pi^2}{3} \rho(\mu)^3 + \frac{\rho(\mu)}{12} - 2\gamma_{-2}(\mu) \right] e^{it(\lambda^2-\mu^2)+ix'(\mu-\lambda)} \\
&+ \mathcal{O}(\Lambda^{-1}L^0) + \mathcal{O}(L^{-1}),
\end{aligned}
\tag{162}
$$

where $\Omega_2^\Lambda$ has been defined in (148).

## 5.3 Density-density correlations in arbitrary macro states for all $x$ and $t$ at order $\mathcal{O}(c^{-2})$

### 5.3.1 Compensation of divergent parts

As explained above, the $\mathcal{O}(c^{-2})$ contributions due to one- and two particle-hole excitations are individually divergent in the thermodynamic limit. The divergent parts are given in (138) and (148) respectively. Their difference is

$$
\Omega_1^\Lambda - \Omega_2^\Lambda = \frac{4}{c^2 L^4} \sum_a \sum_{\substack{n \\ \forall k, \lambda_{a,n} \neq \lambda_k \\ |\lambda_{a,n}| < \Lambda}} \sum_{\substack{i,j \\ i \neq j, \ j \neq a \\ |\lambda_j| > \Lambda}} \frac{1}{(\lambda_i - \lambda_j)^2} e^{it[\lambda_a^2 - \lambda_{a,n}^2] + ix'[\lambda_{a,n} - \lambda_a]}.
\tag{163}
$$

Crucially this vanishes for the class of root densities we use in our $\Lambda$-regularization discussed in Section 4.3.3, i.e. $\rho(\lambda) = 0$ for $|\lambda| > \Lambda$. Indeed, the second sum is zero whenever all the roots satisfy $|\lambda_j| < \Lambda$. We conclude that within our regularization scheme all divergences cancel at order $\mathcal{O}(c^{-2})$, but they do so in a non-trivial fashion: divergent contributions from intermediate states with one particle-hole excitation precisely cancel those arising from intermediate states with two particle-hole excitations.

### 5.3.2 Compensation of contributions that depend on the choice of representative state

As we have seen above, the contributions from both one- and two-particle-hole excitations in the thermodynamic limit individually depend on the choice of the representative state through the pair distribution function $\gamma_{-2}(\lambda)$. Importantly, these contributions exactly cancel one another and the full correlation function *does not* depend on the representative state.

### 5.3.3 $\Lambda$-regularized correlation function

Combining the results for the one and two particle-hole excitations we obtain the following result for the dynamical density-density correlator in the $\Lambda$-regularization

$$
\begin{aligned}
\langle \sigma(x,t)\sigma(0,0) \rangle_\Lambda = D^2 &+ \int_{-\infty}^\infty d\lambda \int_{-\Lambda}^\Lambda d\mu\, f_{x',t}(\lambda,\mu) e^{it(\lambda^2-\mu^2)+ix'(\mu-\lambda)} \\
&+ \mathcal{O}(\Lambda^{-1}L^0) + \mathcal{O}(L^{-1}),
\end{aligned}
\tag{164}
$$

where the integrand is given by

$$f_{x,t}(\lambda,\mu) = \chi_{x,t}^{(1)}(\lambda,\mu) + \frac{\chi_{x,t}^{(2)}(\lambda,\mu)}{c^2} + \mathcal{O}(c^{-3}) . \tag{165}$$

Here the contributions due to one and two particle-hole excitations are respectively

$$\chi_{x,t}^{(1)}(\lambda,\mu) = (1 + \tfrac{2D}{c})^2 \rho(\lambda)\rho_h(\mu) \Big[ 1 - \frac{4}{c}(\mu-\lambda)(\tilde{\rho}(\mu) - \tilde{\rho}(\lambda)) + \frac{8}{c^2}(\mu-\lambda)^2(\tilde{\rho}(\mu) - \tilde{\rho}(\lambda))^2$$

$$+ \frac{4\pi^2}{c^2}(\mu-\lambda)^2 \rho(\mu)\rho_h(\mu) \Big],$$

$$\chi_{x,t}^{(2)}(\lambda,\mu) = 4\rho(\lambda)\rho_h(\mu)\Big[(\mu-\lambda)^2[A_{x,t} + 2B_{x,t}(\lambda) - B_{x,t}(\mu)] + (3\mu - 2\lambda)C_{x,t} - D_{x,t}\Big]. \tag{166}$$

The function $\tilde{\rho}(\lambda)$ is defined in (114) and the four functions $A_{x,t}$, $B_{x,t}(\lambda)$, $C_{x,t}$ and $D_{x,t}$ are given in (147), (152), (155) and (158) respectively.

Some comments on the term $\frac{4\pi^2}{c^2}(\mu - \lambda)^2\rho(\lambda)\rho(\mu)\rho_h(\mu)^2$ are in order. This term arises from the sum of the contributions involving the pair distribution function $\gamma_{-2}(\lambda)$ in both $\mathcal{C}_1^\Lambda(x,t)$ and $\mathcal{C}_2^\Lambda(x,t)$. Strictly speaking it therefore involves two particle-hole excitations as well one particle-hole excitations. Since it does not involve double integrals, as is the case for the other contributions from $\mathcal{C}_2^\Lambda(x,t)$, we have chosen to include it entirely in $\chi_{x,t}^{(1)}(\lambda,\mu)$. It can be interpreted as a "dressing" of contributions arising from one particle-hole excitations by two particle-hole excitations.

### 5.3.4 Dynamical correlations

The result (164) gives the thermodynamic limit of the $\Lambda$-regularized correlation function. We now remove the cutoff dependence by taking the limit $\Lambda \to \infty$. The resulting ill-defined integrals (92) are to be understood as distributions following (96). To express the limit $\Lambda \to \infty$ in terms of well-defined integrals we consider the expansion of $f_{x,t}(\lambda,\mu)$ around $\mu \to \infty$

$$f_{x,t}(\lambda,\mu) = \mu^2 \varphi_{x,t}^{(2)}(\lambda) + \mu\varphi_{x,t}^{(1)}(\lambda) + \varphi_{x,t}^{(0)}(\lambda) + o(\mu^0) . \tag{167}$$

Defining

$$\tilde{f}_{x,t}(\lambda,\mu) = \Big[\Big(\tfrac{x}{2t}\Big)^2 + \tfrac{1}{2it} - \mu^2\Big]\varphi_{x,t}^{(2)}(\lambda) + \Big[\tfrac{x}{2t} - \mu\Big]\varphi_{x,t}^{(1)}(\lambda) + f_{x,t}(\lambda,\mu), \tag{168}$$

it follows from (96) that we can express the limit $\Lambda \to \infty$ of (164) as a function of $x$ and $t \neq 0$

$$\langle\sigma(x,t)\sigma(0,0)\rangle = D^2 + \int_{-\infty}^{\infty}\int_{-\infty}^{\infty} \tilde{f}_{x',t}(\lambda,\mu)e^{it(\lambda^2-\mu^2)+ix'(\mu-\lambda)}d\lambda d\mu + \mathcal{O}(L^{-1}). \tag{169}$$

For the energy of a macro state to be well-defined we need $\rho(\mu) = o(\mu^{-2})$ for $\mu \to \infty$. From this we have

$$\varphi_{x,t}^{(2)}(\lambda) = 4\frac{(1 + \tfrac{2D}{c})^3}{2\pi c^2}\rho(\lambda)\Big[2\tilde{\rho}(\lambda)^2 + A_{x',t} + 2B_{x',t}(\lambda)\Big],$$

$$\varphi_{x,t}^{(1)}(\lambda) = 4\frac{(1 + \tfrac{2D}{c})^3}{2\pi c}\rho(\lambda)\Big[\tilde{\rho}(\lambda) - \frac{4\lambda\tilde{\rho}(\lambda)^2 + 4D\tilde{\rho}(\lambda) + 2\lambda A_{x',t} + 4\lambda B_{x',t}(\lambda) - 2C_{x',t}}{c}\Big],$$

$$\varphi_{x,t}^{(0)}(\lambda) = 4\frac{(1 + \tfrac{2D}{c})^3}{2\pi c}\rho(\lambda)\Big[-D - \lambda\tilde{\rho}(\lambda) + \frac{1}{c}\Big[2D^2 + 8D\lambda\tilde{\rho}(\lambda) + 2\lambda^2\tilde{\rho}(\lambda)^2$$

$$+ \lambda^2 A_{x',t} + 2\lambda^2 B_{x',t}(\lambda) - 2D_{x',t} - 4\lambda C_{x',t}\Big]\Big]. \tag{170}$$

Alternatively, one can also write, using (97)

$$\langle \sigma(x,t)\sigma(0,0)\rangle = D^2 + \lim_{\epsilon \to 0^+} \int_{-\infty}^{\infty} \int_{-\infty}^{\infty} f_{x',t}(\lambda,\mu)e^{it\lambda^2 - i(t-i\epsilon)\mu^2 + ix'(\mu-\lambda)} \mathrm{d}\lambda \mathrm{d}\mu + \mathcal{O}(L^{-1}).$$
(171)

### 5.3.5 Static correlations

The result (169) is singular for $t \to 0$ since it behaves as $\frac{1}{\sqrt{t}}e^{-x^2/t}$. However, in a distribution sense we have $\frac{1}{\sqrt{t}}e^{-x^2/t} \to 0$ when $t \to 0$. Defining

$$\tilde{f}_{x,0}(\lambda,\mu) = \lim_{t \to 0}\left[-\mu^2 \varphi_{x,t}^{(2)}(\lambda) - \mu \varphi_{x,t}^{(1)}(\lambda) - \varphi_{x,t}^{(0)}(\lambda) + f_{x,t}(\lambda,\mu)\right],$$
(172)

we have the following representation of the static correlator as a function of $x$

$$\langle \sigma(x,0)\sigma(0,0)\rangle = D^2 + \int_{-\infty}^{\infty} \int_{-\infty}^{\infty} \tilde{f}_{x',0}(\lambda,\mu)e^{ix'(\mu-\lambda)} \mathrm{d}\lambda \mathrm{d}\mu + \mathcal{O}(L^{-1}).$$
(173)

Alternatively, one can also write

$$\langle \sigma(x,0)\sigma(0,0)\rangle = D^2 + \lim_{\epsilon \to 0^+} \int_{-\infty}^{\infty} \int_{-\infty}^{\infty} f_{x',0}(\lambda,\mu)e^{-\epsilon\mu^2 + ix'(\mu-\lambda)} \mathrm{d}\lambda \mathrm{d}\mu + \mathcal{O}(L^{-1}).$$
(174)

## 5.4 Dynamical structure factor in arbitrary macro states for all $\omega$, $q$ at order $c^{-2}$

Given the correlation function (169) for all $x$ and $t$ we can determine the dynamical structure factor (DSF) $S(q,\omega)$ by taking the Fourier transform

$$S(q,\omega) = \int_{-\infty}^{\infty} \int_{-\infty}^{\infty} \left[\lim_{L \to \infty} \langle \sigma(x,t)\sigma(0,0)\rangle\right] e^{i\omega t - iqx} \mathrm{d}x\mathrm{d}t.$$
(175)

It is convenient to decompose $S(q,\omega)$ in terms of the contributions of one and two particle-hole excitations, which we denote by $S^{(1)}(q,\omega)$ and $S^{(2)}(q,\omega)$ respectively:

$$S(q,\omega) = D^2\delta(q)\delta(\omega) + S^{(1)}(q,\omega) + S^{(2)}(q,\omega) + \mathcal{O}(c^{-3}).$$
(176)

In practice we determine the dynamical structure factor by first computing the Fourier transform (175) of the $\Lambda$-regularized correlator (164) and then taking the limit $\Lambda \to \infty$, which turns out to be straightforward.

### 5.4.1 One particle-hole contributions to the dynamical structure factor

The contribution of the one particle-hole excitations to the DSF $S^{(1)}(q,\omega)$ is obtained from the relation

$$\int_{-\infty}^{\infty} \mathrm{d}t e^{i\omega t} \int_{-\infty}^{\infty} \mathrm{d}x e^{-iqx} \int_{-\infty}^{\infty} \mathrm{d}\lambda e^{it\lambda^2 - ix\lambda} \int_{-\Lambda}^{\Lambda} \mathrm{d}\mu e^{-it\mu^2 + ix\mu} f(\lambda,\mu) = $$
$$\frac{2\pi^2}{|q|} f\left(\frac{\omega-q^2}{2q}, \frac{\omega+q^2}{2q}\right) \mathbf{1}_{|\frac{\omega+q^2}{2q}|<\Lambda}.$$
(177)

The $\Lambda \to \infty$ limit of (177) is straightforward and yields at order $\mathcal{O}(c^{-2})$

$$S^{(1)}(q,\omega) = 2\pi^2 \big(1 + \frac{2D}{c}\big)\rho(\tfrac{\omega'-q'^2}{2q'})\rho_h(\tfrac{\omega'+q'^2}{2q'})\Big[\frac{1}{|q'|} - \frac{4\,\mathrm{sgn}(q')}{c}(\tilde{\rho}(\tfrac{\omega'+q'^2}{2q'}) - \tilde{\rho}(\tfrac{\omega'-q'^2}{2q'}))$$
$$+ \frac{8|q'|}{c^2}(\tilde{\rho}(\tfrac{\omega'+q'^2}{2q'}) - \tilde{\rho}(\tfrac{\omega'-q'^2}{2q'}))^2 + \frac{4\pi^2|q'|}{c^2}\rho(\tfrac{\omega'+q'^2}{2q'})\rho_h(\tfrac{\omega'+q'^2}{2q'})\Big], \quad (178)$$

where we have defined

$$q' = \frac{q}{1 + \frac{2D}{c}}, \qquad \omega' = \omega - \frac{4\delta}{c(1 + \frac{2D}{c})}q. \qquad (179)$$

### 5.4.2 Two particle-hole contributions to the dynamical structure factor

The two particle-hole contributions involve the functions $A_{x,t}$, $B_{x,t}(\lambda)$, $C_{x,t}$ and $D_{x,t}$ given in (147), (152), (155) and (158) respectively. Their simple dependence on $x$ and $t$ allows for a straightforward computation of their contribution to the DSF. For example, by first integrating over $x$, then over $v$, then over $t$ and finally over $u$ we find

$$\int_{-\infty}^{\infty} dt \int_{-\infty}^{\infty} dx\, e^{i\omega t - iqx} \int_{-\infty}^{\infty} d\lambda \int_{-\Lambda}^{\Lambda} d\mu\, \rho(\lambda)\rho_h(\mu)(3\mu - 2\lambda)C_{x',t}e^{it(\lambda^2-\mu^2)+ix'(\mu-\lambda)}$$
$$= 2\pi^2 \int_{-\infty}^{\infty} d\lambda \int_{-\Lambda}^{\Lambda} d\mu\, \rho(\lambda)\rho_h(\mu)\rho(\bar{\lambda})\rho_h(\bar{\mu})\frac{3\mu - 2\lambda}{(q'+\lambda-\mu)|q'+\lambda-\mu|}. \qquad (180)$$

Here we have set

$$\bar{\lambda} = \frac{\omega' + \lambda^2 - \mu^2 - (q'+\lambda-\mu)^2}{2(q'+\lambda-\mu)}, \qquad \bar{\mu} = \frac{\omega' + \lambda^2 - \mu^2 + (q'+\lambda-\mu)^2}{2(q'+\lambda-\mu)}. \qquad (181)$$

The limit $\Lambda \to \infty$ of this expression is again routine. It is however not immediately obvious that the double integral over $\lambda$ and $\mu$ in (180) is well-defined, since one of the factors in the integrand exhibits a non-integrable singularity. A closer inspection reveals that this singularity is cancelled by the product of root densities. We will show below by means of a change of variable that the double integral is indeed well-defined.

All other terms involving the functions $B_{x,t}(\lambda)$, $B_{x,t}(\mu)$ and $D_{x,t}$ can be computed analogously. The term involving $A_{x,t}$ however requires a slightly modified approach, since following through the same steps as before would split the $1/v^2$ term into a sum of two quantities that are individually divergent. In order to circumvent this problem we replace the $1/v^2$ by $\frac{1}{v^2+\epsilon^2}$ and send $\epsilon \to 0$ in the final result. We obtain

$$\int_{-\infty}^{\infty} dt \int_{-\infty}^{\infty} dx\, e^{i\omega t - iqx} \int_{-\infty}^{\infty} d\lambda \int_{-\Lambda}^{\Lambda} d\mu\, \rho(\lambda)\rho_h(\mu)(\lambda - \mu)^2 A_{x',t}e^{it(\lambda^2-\mu^2)+ix'(\mu-\lambda)} \qquad (182)$$
$$= 2\pi^2 \int_{-\infty}^{\infty} d\lambda \int_{-\Lambda}^{\Lambda} d\mu\, \frac{1}{|q'+\lambda-\mu|^3}\Big[\rho(\lambda)\rho_h(\mu)\rho(\bar{\lambda})\rho_h(\bar{\mu})(\lambda-\mu)^2$$
$$- |q'(\lambda-\mu)|\rho(\bar{\lambda})\rho_h(\bar{\mu})\rho(\tfrac{\omega'-q'^2}{2q'})\rho_h(\tfrac{\omega+q'^2}{2q'})\Big]. \qquad (183)$$

Putting everything together we obtain the following result for the contribution of two particle-

hole excitations to the DSF

$$S^{(2)}(q,\omega) = \frac{8\pi^2}{c^2} \int_{-\infty}^{\infty} \int_{-\infty}^{\infty} \rho(\lambda)\rho_h(\mu)\rho(\bar{\lambda})\rho_h(\bar{\mu}) \frac{\frac{2(\lambda-\mu)^2}{\lambda-\bar{\lambda}} - \frac{(\lambda-\mu)^2}{\mu-\bar{\lambda}} + 3\mu - 2\lambda - \bar{\lambda}}{(q'+\lambda-\mu)|q'+\lambda-\mu|} d\lambda d\mu$$

$$+ \frac{8\pi^2}{c^2} \int_{-\infty}^{\infty} \int_{-\infty}^{\infty} \frac{1}{|q'+\lambda-\mu|^3} \Big[ \rho(\lambda)\rho_h(\mu)\rho(\bar{\lambda})\rho_h(\bar{\mu})(\lambda-\mu)^2$$

$$- |q'(\lambda-\mu)|\rho(\bar{\lambda})\rho_h(\bar{\mu})\rho(\tfrac{\omega'-q'^2}{2q'})\rho_h(\tfrac{\omega'+q'^2}{2q'}) \Big] d\lambda d\mu$$

$$+ \mathcal{O}(c^{-3}). \tag{184}$$

In order to make the convergence of this integral explicit we perform a change of variables from $\lambda, \mu$ to

$$z = 1 + \frac{\lambda-\mu}{q}, \qquad p = \frac{\omega'-q'(\lambda+\mu)}{2q'z}, \tag{185}$$

and define

$$q_1 = \frac{\omega'-2q'zp-q'^2(1-z)}{2q'}, \qquad q_2 = \frac{\omega'-2q'zp+q'^2(1-z)}{2q'},$$

$$q_3 = \frac{\omega'+2q'p(1-z)-q'^2z}{2q'}, \qquad q_4 = \frac{\omega'+2q'p(1-z)+q'^2z}{2q'}. \tag{186}$$

In terms of the new variables we have

$$S^{(2)}(q,\omega) = \frac{8\pi^2}{c^2} \int_{-\infty}^{\infty} \int_{-\infty}^{\infty} h(q,\omega,z,p) dz dp + \mathcal{O}(c^{-3}),$$

$$h(q,\omega,z,p) = \frac{2}{q'z}\rho(q_1)\rho_h(q_2)\rho(q_3)\rho_h(q_4) \left[ \frac{5q'}{4} - q'z - \frac{p}{2} + \frac{2q'^2(1-z)^2}{(2z-1)q'-2p} - \frac{(1-z)^2q'^2}{q'-2p} \right]$$

$$+ \frac{\rho(q_3)\rho_h(q_4)}{z^2} \Big[ (1-z)^2\rho(q_1)\rho_h(q_2) - |1-z|\rho(\tfrac{\omega'-q'^2}{2q'})\rho_h(\tfrac{\omega'+q'^2}{2q'}) \Big]. \tag{187}$$

The integral over $p$ only has singularities that are integrable in a principal value sense, and after the integral over $p$ has been carried out the integral over $z$ only has a singularity that is integrable in a principal value sense. We conclude that (187) is well defined and can be straightforwardly evaluated numerically.

## 6 Numerical evaluation of the dynamical structure factor

In this section we numerically evaluate the integral representations (178), (187) in order to determine $S(q,\omega)$ for the two examples of root densities introduced in Section 3.4, namely thermal states and the non-equilibrium steady state after a quantum quench from the ground state at $c = 0$.

### 6.1 Zero temperature

We first consider the zero temperature case for density $D = 0.404$ and $c = 3$ in (106). The value $D/c \approx 0.13$ is well within the expected range of validity of the $1/c$-expansion. The same holds true for all other cases considered below. In Figure 3 we present numerical results for the DSF at order $c^{-2}$ as well as for the one particle-hole contribution $S^{(1)}(q,\omega)$. It is well known that at zero temperature the one particle-hole contribution to the DSF is non-zero only in a certain region of the $q,\omega$ plane for kinematic reasons, and exhibits (not necessarily divergent) singularities at the edges of its support [30, 32, 39, 77]. We note that although the

DSF is expected to diverge near the upper threshold, the divergence near the lower thresholds is a consequence of the $1/c$ expansion, that produces logarithms instead of a finite behaviour with a fractional $c$-dependent exponent. Comparing the full result (left panel) to $S^{(1)}(q, \omega)$ (right panel) we observe that the contributions due to two particle-hole excitations significantly modify the numerical values of the DSF within this region. $S^{(2)}(q, \omega)$ is also non-zero outside the region, but this effect is barely visible in the plot.

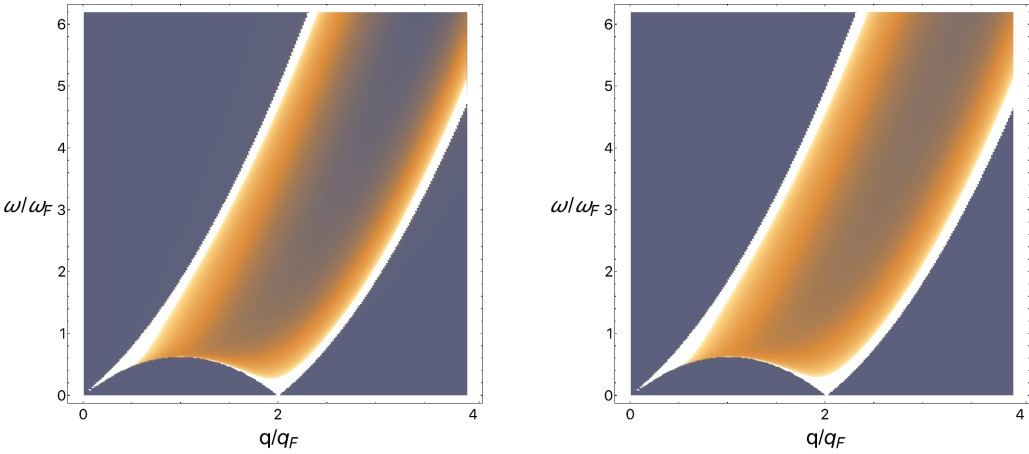

Figure 3: $S(q, \omega)$ (left panel) and $S^{(1)}(q, \omega)$ (right panel) as functions of $q$ and $\omega$ at zero temperature and $D = 0.404$, $c = 3$ in (106). The color scale is the same for both plots.

## 6.2 Finite temperature

We next turn to the DSF at finite temperatures. Figure 4 presents numerical results for the full

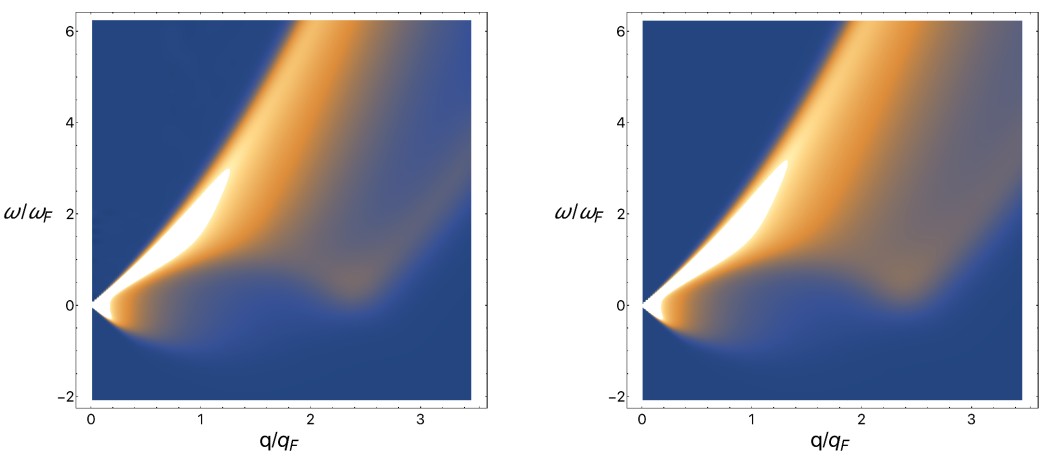

Figure 4: $S(q, \omega)$ (left panel) and $S^{(1)}(q, \omega)$ (right panel) as functions of $q$ and $\omega$ for a thermal state with $\beta = 5$, $D = 0.396$ and $c = 4$. The color scale is the same for both plots.

DSF $S(q, \omega)$ at order $c^{-2}$ for thermal states with $\beta = 5$, $c = 4$ and $D = 0.396$. For comparison we also plot the one particle-hole contribution $S^{(1)}(q, \omega)$. Like in zero temperature case, for these parameter values the one particle-hole contribution already gives a fairly good account of

the full DSF. The two-particle-hole contribution modifies some details that become increasingly significant for $q > 2q_F$. The main difference to the zero temperature case is the emergence of spectral weight at negative frequencies and the "washing out" of the threshold singularities.

In Figure 5 we consider the DSF for a different thermal state characterized by a higher temperature $\beta = 1$ and $D = 0.38$, $c = 4$. The differences between the $S(q, \omega)$ and $S^{(1)}(q, \omega)$ are difficult to discern in these plots. In order to get a more precise notion of the relative

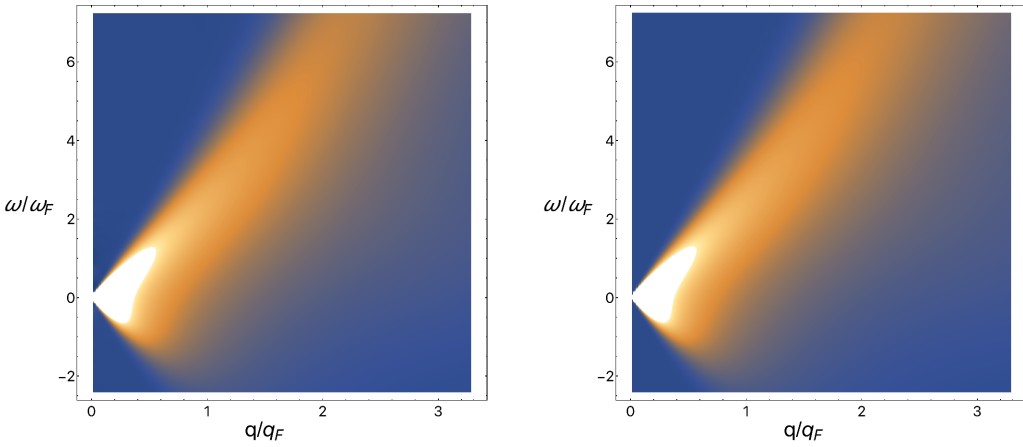

Figure 5: Left panel: DSF $S(q, \omega)$ as a function of $q$ and $\omega$ for a thermal state at inverse temperature $\beta = 1$, $D = 0.38$ and $c = 4$. Right panel: Same for the one particle-hole contribution $S^{(1)}(q, \omega)$. The color scale is the same for both plots.

contributions of $S^{(1)}(q, \omega)$ and $S^{(2)}(q, \omega)$ to the DSF for these values of $D$, $c$ and $\beta$, we show a number of "constant momentum cuts", i.e. plots of $S(q, \omega)$ as a function of $\omega$ for fixed $q$, in Figs 6 and 7. Fig 6 gives representative results at "small" momenta, defined as $q \lesssim q_F$. We see that the contribution from two particle-hole excitations is negligibly small. This is in perfect agreement with observations made in Ref. [72] based on comparisons with numerical computations for a finite number of particles. Our results makes this observation fully quantitative in the thermodynamic limit in the framework of a $1/c$-expansion.

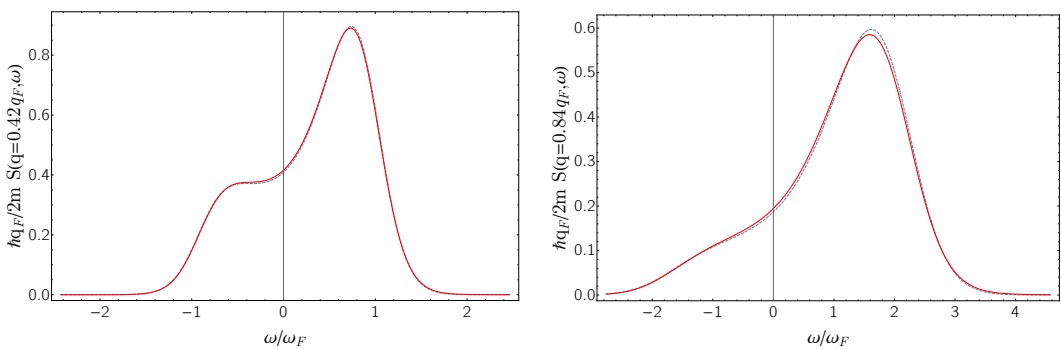

Figure 6: $S(q, \omega)$ (red) and $S^{(1)}(q, \omega)$ (blue, dotted) as functions of $\omega$ for $q = 0.42q_F$ (left panel) and $q = 0.84q_F$ (right panel). The parameters are the same as in Figure 5.

Figure 7 shows how the relative magnitude of $S^{(2)}(q, \omega)$ evolves at larger values of momentum. We see that it grows with $q$ and for the values shown is no longer negligible. We further note that while the DSF is expressed as a spectral sum with only positive terms, the contribution $S^{(2)}(q, \omega)$ can be negative. The explanation of this behaviour is that the contributions of

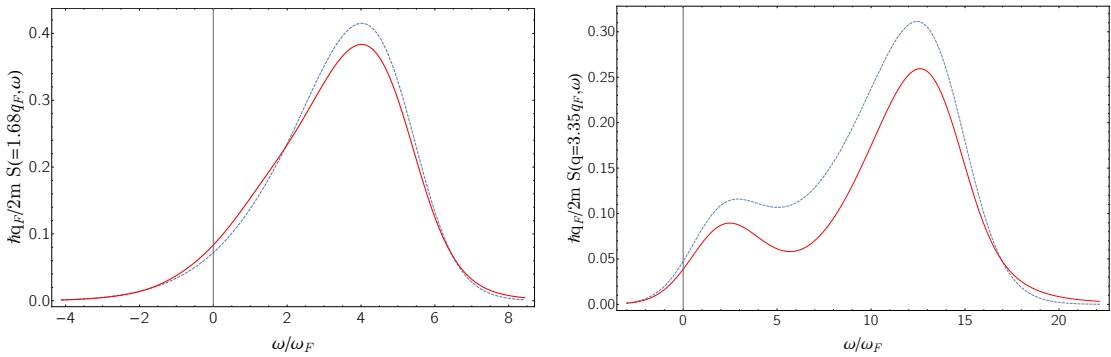

Figure 7: Same as Figure 6 but for $q = 1.68q_F$ (left panel) and $q = 3.35q_F$ (right panel).

one and two particle-hole excitations include terms that arise from cross-cancellations of divergences occurring in their 'bare' spectral sum. Stated differently, each of them incorporates contributions due to one and two particle-hole excitations so as to have well-defined thermodynamic limits. If we consider the spectral sum in a large finite volume, the bare (without cross-cancelling divergences) one and two particle-hole contributions are indeed separately positive in the following sense: The leading $c^0$ term of the bare contribution of one particle-hole excitations is positive, and we see from (148) that the same holds true for the divergent part of the leading $c^{-2}$ term of the two particle-hole excitations. The interpretation of the resulting contributions as one or two-particle-hole excitations is imposed by whether they are expressed as a double integral (one for the particle and one for the hole) or a quadruple integral (two particles and two holes). Finally, we note that the fact that $S^{(2)}(q, \omega)$ can be negative is an inherent feature of the $1/c$ expansion as can be seen by considering the zero temperature limit. Here the successive terms of the $1/c$ expansion of the DSF exhibit a singularity with negative spectral weight, see Section 7.2.2, although at finite $c$ all the higher order terms exponentiate into a positive spectral weight.

## 6.3 DSF in a non-equilibrium steady state

In Figure 8 we show numerical results for $S(q, \omega)$ and $S^{(1)}(q, \omega)$ for the root density given in Section 3.4.2. The latter describes the stationary state reached for the interaction quench of Refs [97, 99, 100], where the system is initialized in the ground state at $c = 0$ and density $D = 1/\pi$ and time-evolved with the Lieb-Liniger Hamiltonian at $c = 4$. We observe that the two particle-hole contributions lead to a slight narrowing of the DSF for $q > q_F$.

# 7 Analysis of the result in limiting cases

In this section we report a detailed analysis of our results for the density-density correlation function (171), (166) and the dynamical structure factor (178), (187). The details of the derivations of the results in this section are reported in Appendix C.

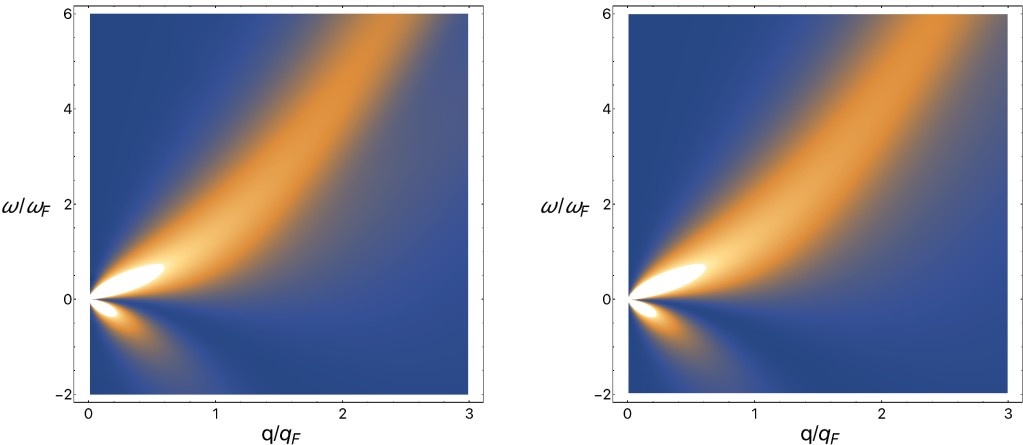

Figure 8: $S(q,\omega)$ (left panel) and one particle-hole contribution $S^{(1)}(q,\omega)$ (right panel) as functions of $q,\omega$ for the steady state root density (62) with $c = 4$ and $D = 1/\pi$. The color scale is the same for both plots.

## 7.1 Density-density correlation function

### 7.1.1 Asymptotics of equal-time correlations at zero temperature

At zero temperature with the root density (106), we obtain the following asymptotic behaviour at large $x$ and at order $c^{-2}$

$$
\langle \sigma(x,0)\sigma(0,0)\rangle = D^2 - \frac{1 + \frac{4D}{c} + \frac{4D^2}{c^2}}{2\pi^2 x^2}
$$
$$
+ A \frac{\cos(2q_F x)}{x^2}\left[ 1 - \left( \frac{8D}{c} + \frac{8D^2}{c^2}\right)\log[2q_F e^{\gamma_E}x] + \frac{32D^2}{c^2}\log^2[2q_F e^{\gamma_E}x]\right] \quad (188)
$$
$$
+ o(x^{-2}), \quad (189)
$$

where

$$
A = \frac{1 + \frac{4D}{c}}{2\pi^2} + \mathcal{O}(c^{-2}), \quad (190)
$$

and $\gamma_E$ is Euler's constant. This expression is the large $x$ behaviour of the $1/c$ expansion of the correlation functions, hence one has $c$ large first and then $x$ large.

Combining CFT/Luttinger liquid theory with exact results provides the following prediction for the correlations at large $x$ at fixed $c$ [26–29,39]

$$
\langle \sigma(x,0)\sigma(0,0)\rangle = D^2 - \frac{K}{2\pi^2 x^2} + A_1\frac{\cos(2q_F x)}{x^{2K}} + \cdots, \quad (191)
$$

with $K$ given in (107), and with $A_1$ a known constant [34,35,40]. If one wishes to compare this expression with (189), one is a priori faced with two problems:

(i) If one expands (191) in powers of $c^{-1}$ one has to take first $x$ large and then $c$ large, which is the reverse of (189). Hence comparing (189) and (191) entails to commute two limits. This commutation is possible if our expansion in $c^{-1}$ (189) is *uniform* in space.

(ii) There could be corrections to (191) that are subleading in $x$ at fixed $c$, but become of the same order as the dominant term once expanded in $c^{-1}$ (i.e. give rise to $\log(x)$ terms). An example would be a term $\propto x^{-4K+2}$. These corrections would be visible in (189), but not in (191).

In the case of density correlations in the Lieb-Liniger model it follows from the exact large $x$ expansion at fixed $c$ [39] that there are no subleading corrections with the property described in (ii). We thus expand (191) in powers of $c^{-1}$. Since $K \to 1$ when $c \to \infty$ the power-law $x^{-2K}$ becomes $x^{-2}$ corrected by logarithms and we find

$$
\begin{aligned}
\langle \sigma(x,0)\sigma(0,0)\rangle = D^2 &- \frac{1 + \frac{4D}{c} + \frac{4D^2}{c^2}}{2\pi^2 x^2} \\
&+ A_1 \frac{\cos(2q_F x)}{x^2}\left[1 - \left(\frac{8D}{c} + \frac{8D^2}{c^2}\right)\log x + \frac{32D^2}{c^2}\log^2 x\right] + \mathcal{O}(c^{-3}).
\end{aligned} \tag{192}
$$

The coefficient $A_1$ depends on $c$ but as its representation is rather complicated [34, 35] (with approximations in [115, 116]) we left calculating its $c^{-1}$ expansion for future work. We see that it agrees with (189) if we identify

$$
A_1 = A(2q_F e^{\gamma_E})^{2-2K} + \mathcal{O}(c^{-2}). \tag{193}
$$

In particular the critical exponents are reproduced at order $c^{-2}$. This both provides a check of our formula, and shows that our $1/c$ expansion is *uniform* in space.

### 7.1.2  Dynamical correlations asymptotics at zero temperature

At zero temperature (with the root density (106)) we can evaluate the asymptotic behaviour of the dynamical correlation function at large $x, t$ at fixed

$$
\alpha = \frac{x}{2t}. \tag{194}
$$

It is convenient to define

$$
\alpha' = \frac{x'}{2t} = \left(1 + \frac{2D}{c}\right)\alpha, \tag{195}
$$

and set

$$
s = \begin{cases} 1 & \text{if } |\alpha| > q_F \\ -1 & \text{if } |\alpha| < q_F \end{cases}. \tag{196}
$$

We obtain

$$
\langle \sigma(x,t)\sigma(0,0)\rangle = D^2 + \sum_{\sigma=\pm} B_\sigma \frac{e^{ist(Q+\sigma\alpha')^2}}{|t|^{3/2}}\left[1 - \nu_\sigma \log(i\varpi_\sigma t) + \frac{\nu_\sigma^2}{2}\log^2(i\varpi_\sigma t)\right] \tag{197}
$$

$$
+ o(t^{-3/2}), \tag{198}
$$

with

$$
\begin{aligned}
B_\sigma &= \frac{\operatorname{sgn}(t)e^{-s\frac{i\pi}{4}\operatorname{sgn}(t)}\left(1 + \frac{2D}{c}\right)^4}{8i\pi^{\frac{3}{2}}(Q+\sigma\alpha')} + \mathcal{O}(c^{-2}), \\
\nu_\sigma &= \left(1 + \frac{2Q}{\pi c}\right)\frac{2(Q+\sigma\alpha')}{\pi c} + \frac{2(Q+\sigma\alpha')^2}{\pi^2 c^2} + \mathcal{O}(c^{-3}), \\
\varpi_\sigma &= s\sigma 4Q\frac{(Q+\sigma\alpha')^2}{|Q-\sigma\alpha'|}e^{\gamma_E}.
\end{aligned} \tag{199}
$$

Ref. [39] derived the full asymptotic expansion at large $x, t$ for any value of $c$ at zero temperature. The $c^{-1}$ expansion of this result at order $c^{-2}$ (without expanding the prefactors) is in agreement with (197). In particular the critical exponents $\nu_\pm$ are reproduced at order $c^{-2}$. This both provides a check of our calculation and shows that our $1/c$ expansion is uniform in time as well, since the large $x, t$ and large $c$ limits commute.

### 7.1.3 Asymptotics of dynamical correlations for a generic root density and Generalized Hydrodynamics

For a generic continuous root density $\rho$ in the large $x, t$ regime at fixed $\alpha$ (194) we obtain the following asymptotic behaviour

$$
\begin{aligned}
\langle \sigma(x,t)\sigma(0,0)\rangle =& D^2 + \frac{\pi(1+\frac{2D}{c})^2 \rho(\alpha')\rho_h(\alpha')}{|t|} + \frac{i\pi(1+\frac{2D}{c})^2 \left[\rho''(\alpha')\rho_h(\alpha') - \rho(\alpha')\rho_h''(\alpha')\right]}{4t|t|} \\
& + \frac{\pi^2}{t^2 c^2}\Bigg[12(\rho\rho_h)^2(\alpha') + 8(\rho\rho_h)'(\alpha')\int_{-\infty}^{\infty} \mathrm{sgn}\,(\alpha'-\zeta)\rho(\zeta)\rho_h(\zeta)\mathrm{d}\zeta \\
& \qquad + 2(\rho\rho_h)''(\alpha')\int_{-\infty}^{\infty}|\alpha'-\zeta|\rho(\zeta)\rho_h(\zeta)\mathrm{d}\zeta\Bigg] + o(t^{-2}),
\end{aligned}
\tag{200}
$$

where we recall the definition of $\alpha'$ in (195). The first line arises from one particle-hole contributions, while the second and third lines are two particle-hole contributions. If the root density is not continuous the leading term in $1/t$ is still correct, but the higher order corrections may change.

GHD [85, 86] makes predictions for the coefficient of the $1/t$ term in the density-density correlator for any value of $c$ [89, 90]. For the sake of completeness we summarize the $1/c$-expansion of the GHD results in Appendix C.1.4. The leading term proportional to $1/|t|$ of (200) is in perfect agreement with the order $c^{-2}$ expansion of the GHD results. To the best of our knowledge this constitutes the most non-trivial check to date of GHD predictions in an interacting integrable model.

Importantly, we can assess the accuracy of the GHD approximation outside the asymptotic large space and time regime by comparing it to the full correlations at order $c^{-2}$. In Figure 9 we show our results for the real part and the modulus of $\langle \sigma(x,t)\sigma(0,0)\rangle$ for two thermal states at $c = 4$ together with the GHD approximation. We see that at high temperature $\beta = 1$ the GHD approximation is surprisingly good even at short times. At lower temperatures $\beta = 3$ the correlation is still very well approximated by GHD, but is seen to display damped oscillations in the absolute value that arise from the imaginary part of the correlations that decay as $t^{-2}$ and is not accounted for by GHD.

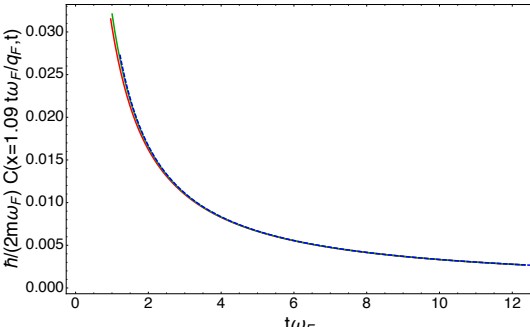
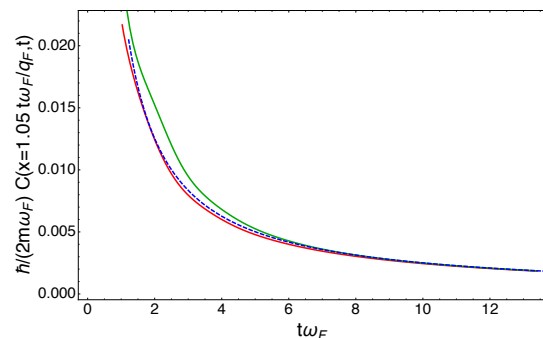

Figure 9: Correlation function $C(x,t) \equiv \langle \sigma(x,t)\sigma(0,0)\rangle$ in a thermal state for $x = 2\alpha t$ as a function of $t$ at $c = 4$, for $\beta = 1$ and $D = 0.38$ (left) and $\beta = 3$ and $D = 0.386$ (right). The three curves depict the real part (red), the modulus (green) and the GHD approximation (dashed blue).

In Figure 10 we present the analogous comparison for a non-equilibrium steady state with root density (62). This root density is "less regular" than thermal densities in the sense that it

has a narrower peak at zero. As a consequence, we expect higher Fourier-like corrections to the oscillatory integral, whereas GHD describes saddle-point-like corrections only. We indeed observe a more pronounced discrepancy for short or intermediate times, but the agreement at later times is still excellent, and globally remains very good.

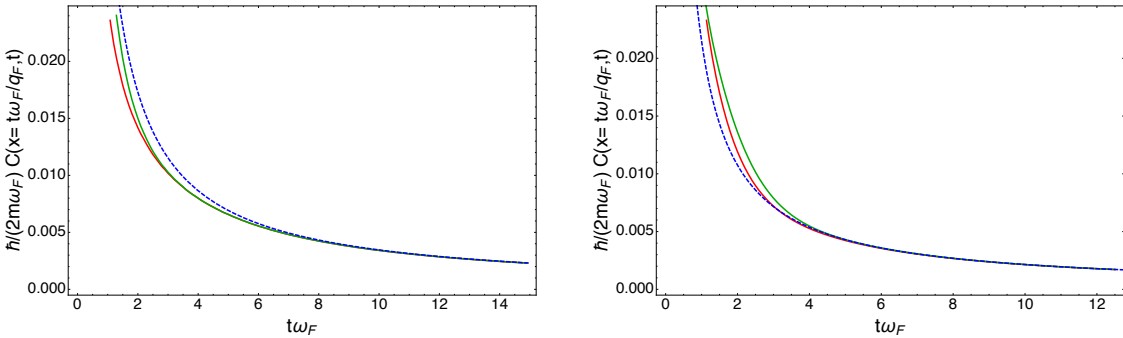

Figure 10: Correlation function $C(x,t) \equiv \langle \sigma(x,t)\sigma(0,0)\rangle$ in the non-equilibrium steady state (62) for $x = 2\alpha t$ as a function of $t$ at $c = 4$, for $D = 1/\pi$ (left) and $D = 1/(2\pi)$ (right). The three curves depict the real part (red), the modulus (green) and the GHD approximation (dashed blue).

## 7.2 Dynamical structure factor

### 7.2.1 A simplified expression at zero temperature

Equations (178), (187) for the DSF can be simplified at zero temperature. The one particle-hole contribution can be written as

$$S^{(1)}(q,\omega) = \frac{\left(1 + \frac{2D}{c}\right)^4}{2|q|}\left[1 - \frac{2q}{\pi c}\log\left|\frac{\omega^2 - \omega_+^2}{\omega^2 - \omega_-^2}\right| + \frac{2q^2}{\pi^2 c^2}\log^2\left|\frac{\omega^2 - \omega_+^2}{\omega^2 - \omega_-^2}\right|\right]\mathbf{1}_{\omega_- < \omega < \omega_+}$$
$$+ \mathcal{O}(c^{-3}), \tag{201}$$

where we have defined

$$\omega_\pm(q) = \left|q'^2 \pm 2|q'|Q\right|. \tag{202}$$

The contribution from two particle-hole excitations can be simplified by carrying out the integral over $p$ in (187)

$$S^{(2)}(q,\omega) = \frac{1}{4\pi^2 q'c^2}\int_{-\infty}^{\infty}\left[\frac{1}{z}\left[(5-4z)q'(Z_+ - Z_-) + Z_-^2 - Z_+^2 + 2q'(1-z)^2\frac{Z_+ - Z_-}{z}\right.\right.$$
$$\left.+ 2q'^2(1-z)^2\log\left|\frac{q' - 2Z_+}{q' - 2Z_-}\right| - 4q'^2(1-z)^2\log\left|\frac{(2z-1)q' - 2Z_+}{(2z-1)q' - 2Z_-}\right|\right]\mathbf{1}_{Z_- < Z_+}$$
$$\left.- \frac{2q'\min(|q'z|, 2Q)}{z^2}\mathbf{1}_{\omega_-(q) < \omega < \omega_+(q)}\right]dz. \tag{203}$$

Here we have defined

$$Z_+(z) = \begin{cases} \min\left[\frac{\omega + 2q'Q + q'^2(z-1)}{2q'z}, \frac{\omega - |2q'Q - q'^2z|}{2q'(z-1)}\right] & \text{if } z \geq 1 \\ \min\left[\frac{\omega - |2q'Q - q'^2(1-z)|}{2q'z}, \frac{-\omega + 2q'Q + q'^2z}{2q'(1-z)}\right] & \text{if } 0 \leq z \leq 1 \\ \min\left[\frac{\omega - 2q'Q - q'^2(1-z)}{2q'z}, \frac{-\omega - |2q'Q + q'^2z|}{2q'(1-z)}\right] & \text{if } z \leq 0, \end{cases} \tag{204}$$

and

$$Z_-(z) = \begin{cases} \max\left[\frac{\omega + |2q'Q - q'^2(z-1)|}{2q'z}, \frac{\omega - 2q'Q - q'^2 z}{2q'(z-1)}\right] & \text{if } z \geq 1 \\ \max\left[\frac{\omega - 2q'Q - q'^2(1-z)}{2q'z}, \frac{-\omega + |-2q'Q + q'^2 z|}{2q'(1-z)}\right] & \text{if } 0 \leq z \leq 1 \\ \max\left[\frac{\omega - |2q'Q - q'^2(1-z)|}{2q'z}, \frac{-\omega - 2q'Q + q'^2 z}{2q'(1-z)}\right] & \text{if } z \leq 0. \end{cases} \tag{205}$$

### 7.2.2 Behaviour near the thresholds at zero temperature

At zero temperature, the DSF exhibits divergences at certain threshold energies $\omega_{\text{th}}(q)$. In our case, at order $\mathcal{O}(c^{-2})$, the two thresholds occur at $\omega_\pm(q)$ defined in (202). For $q < 2Q$ we find the following singular behaviour of the one particle-hole DSF near them

$$S^{(1)}(q, \omega_+ + \delta\omega) = \frac{\left(1 + \frac{2D}{c}\right)^2}{2|q|}\left(1 - \frac{2q}{\pi c}\log\left|\delta\omega \frac{2\omega_+}{\omega_+^2 - \omega_-^2}\right| + \frac{2q^2}{\pi^2 c^2}\log^2\left|\delta\omega \frac{2\omega_+}{\omega_+^2 - \omega_-^2}\right|\right)\mathbf{1}_{\delta\omega<0}$$

$$S^{(1)}(q, \omega_- + \delta\omega) = \frac{\left(1 + \frac{2D}{c}\right)^2}{2|q|}\left(1 + \frac{2q}{\pi c}\log\left|\delta\omega \frac{2\omega_-}{\omega_+^2 - \omega_-^2}\right| + \frac{2q^2}{\pi^2 c^2}\log^2\left|\delta\omega \frac{2\omega_-}{\omega_+^2 - \omega_-^2}\right|\right)\mathbf{1}_{\delta\omega>0}. \tag{206}$$

The analogous results for the two particle-hole contribution are

$$S^{(2)}(q, \omega_+ + \delta\omega) = \left(\frac{q}{2\pi^2 c^2}\log|\delta\omega|\right)\mathbf{1}_{\delta\omega<0} - \left(\frac{q}{2\pi^2 c^2}\log|\delta\omega|\right)\mathbf{1}_{\delta\omega>0},$$

$$S^{(2)}(q, \omega_- + \delta\omega) = \left(\frac{q}{\pi^2 c^2}\log|\delta\omega|\right)\mathbf{1}_{\delta\omega>0}. \tag{207}$$

These limiting behaviours are obtained at large $c$ first, and then $\omega$ close to $\omega_\pm$.

At fixed $c$ non-linear Luttinger liquid theory predicts the exponent of the power-law divergence near these thresholds [30–34, 39]

$$S(q, \omega_+ + \delta\omega) = C_0|\delta\omega|^{\mu_+}\mathbf{1}_{\delta\omega>0} + C_1|\delta\omega|^{\mu_+}\mathbf{1}_{\delta\omega<0} + C_2 + \ldots$$

$$S(q, \omega_- + \delta\omega) = C_3(\delta\omega)^{\mu_-}\mathbf{1}_{\delta\omega>0} + C_4\mathbf{1}_{\delta\omega>0} + \ldots. \tag{208}$$

Here $C_{0,1,2,3,4}$ are $c$-dependent constants and the exponents $\mu_\pm$ have simple expressions that depend on $c$. Results for the non-universal prefactors $C_{0,1,2,3,4}$ at finite $c$ are available in the literature [34]. The dots encompass less singular pieces $|\delta\omega|^\mu$ with a $c$-dependent exponent $\mu > \mu_\pm$, and regular pieces $C_5\delta\omega + O((\delta\omega)^2)$. In the framework of the $1/c$ expansion these power-laws give rise to logarithms

$$|\delta\omega|^{\mu_+} = 1 - \frac{2q}{\pi c}\log|\delta\omega| + \frac{2q^2}{\pi^2 c^2}\log^2|\delta\omega| + \frac{2q^2}{\pi^2 c^2}\log|\delta\omega| + \mathcal{O}(c^{-3})$$

$$|\delta\omega|^{\mu_-} = 1 + \frac{2q}{\pi c}\log|\delta\omega| + \frac{2q^2}{\pi^2 c^2}\log^2|\delta\omega| + \frac{2q^2}{\pi^2 c^2}\log|\delta\omega| + \mathcal{O}(c^{-3}). \tag{209}$$

These expansions are valid if we take $\omega$ close to $\omega_\pm$ first and then consider the large-$c$ limit, and in order to compare with our result we have to commute these two limits. Importantly, the less singular pieces that are subleading in $\delta\omega$ can also produce logarithms if their exponent goes to 0 when $c \to \infty$. However, it follows from our asymptotic analysis in real space that there are no such terms. Comparing (208) with (206) and (207) we find that our result is in

agreement with the non-linear Luttinger liquid predictions if we identify

$$
\begin{aligned}
C_0 &= \frac{1}{4\pi c} + \mathcal{O}(c^{-2}) \\
C_1 &= \frac{(1 + \frac{2D}{c})^2}{2|q|} \left( \frac{2\omega_+}{\omega_+^2 - \omega_-^2} \right)^{\mu_+} + \frac{1}{4\pi c} + \mathcal{O}(c^{-2}) \\
C_2 &= -\frac{1}{4\pi c} + \mathcal{O}(c^{-2}) \\
C_3 &= \frac{(1 + \frac{2D}{c})^2}{2|q|} \left( \frac{2\omega_-}{\omega_+^2 - \omega_-^2} \right)^{\mu_-} + \mathcal{O}(c^{-2}) \\
C_4 &= 0 + \mathcal{O}(c^{-2}).
\end{aligned}
\tag{210}
$$

In particular we obtain the correct exponents at order $c^{-2}$. This provides a check of our result for the DSF and shows that it is uniform in $q$ and $\omega$.

### 7.2.3 Sum rule at zero temperature

The f-sum rule for the dynamical structure factor in equilibrium states reads [113]

$$
\int_{-\infty}^{\infty} S(q,\omega)\omega \mathrm{d}\omega = 2\pi D q^2 \,.
\tag{211}
$$

In our calculation, this sum rule has to be perfectly satisfied at order $c^{-2}$. It is a stringent test of validity of our formula since it has to be satisfied for all $q$ and encompasses every single piece of the DSF. At zero temperature we obtain from Equation (201) that

$$
\int_{-\infty}^{\infty} S^{(1)}(q,\omega)\omega \mathrm{d}\omega = 2\pi D q^2 + \frac{4}{3c^2} q^4 Q + \mathcal{O}(c^{-3}).
\tag{212}
$$

This means that the two-particle-hole DSF (203) $\bar{S}^{(2)}(q,\omega) \equiv c^2 S^{(2)}(q,\omega)$ evaluated at $c = \infty$, must satisfy

$$
\int_{-\infty}^{\infty} \bar{S}^{(2)}(q,\omega)\omega \mathrm{d}\omega = -\frac{4}{3} q^4 Q.
\tag{213}
$$

We computed this integral numerically from (203) for several values of $q$. We find that (213) is indeed satisfied within the numerical accuracy of our calculation. The relative deviations of our results from (213) are around $10^{-4}$, which is quite satisfactory.

### 7.2.4 Detailed balance for thermal states

The dynamical structure factor of a thermal state at inverse temperature $\beta$ should satisfy the detailed balance relation for all values of $q, \omega$

$$
S(q, -\omega) = e^{-\beta \omega} S(q, \omega).
\tag{214}
$$

In our calculation the detailed balance relation for $S(q, \omega)$ should be perfectly satisfied at order $c^{-2}$. We note it is a very stringent test of validity of our formulas for $S(q, \omega)$, given that a thermal state at finite temperature corresponds to a generic root density with a complicated $c$ dependence, while for an arbitrary root density there is no particular relation between $S(q, -\omega)$ and $S(q, \omega)$.

In order to check that our formulas for the DSF satisfy detailed balance at order $c^{-2}$, we need to evaluate (214) with $\rho(\lambda)$ given at order $c^{-2}$ in (103). We found convenient to define

$$
\tilde{S}^{(1)}(q,\omega) = S^{(1)}(q,\omega) - \frac{8\pi^4 |q'|}{c^2} \rho(\tfrac{\omega'-q'^2}{2q'}) \rho(\tfrac{\omega'+q'^2}{2q'}) \rho_h(\tfrac{\omega'+q'^2}{2q'})^2,
\tag{215}
$$

i.e. the one-particle-hole DSF without the dressed piece coming from two particle-hole excitations. We recall that $\omega'$ and $q'$ were previously defined in (179). It is straightforward to check numerically that $\tilde{S}^{(1)}(q, \omega)$ satisfies detailed balance at order $c^{-2}$

$$\tilde{S}^{(1)}(q, -\omega) = e^{-\beta\omega}\tilde{S}^{(1)}(q, \omega) + \mathcal{O}(c^{-3}). \tag{216}$$

Hence the following quantity

$$\tilde{S}^{(2)}(q, \omega) = c^2 S^{(2)}(q, \omega) + 8\pi^4 |q'| \rho(\tfrac{\omega' - q'^2}{2q'}) \rho(\tfrac{\omega' + q'^2}{2q'}) \rho_h(\tfrac{\omega' + q'^2}{2q'})^2, \tag{217}$$

evaluated at $c = \infty$ should also independently satisfy detailed balance

$$\tilde{S}^{(2)}(q, -\omega) = e^{-\beta\omega}\tilde{S}^{(2)}(q, \omega). \tag{218}$$

We find that this indeed holds within the accuracy of our numerical computation, i.e. within a relative error of $10^{-5}$. This is quite satisfactory.

### 7.2.5 Behaviour at small $q, \omega$

At small $q, \omega$ with fixed

$$\gamma = \frac{\omega}{2q}, \tag{219}$$

we find the following behaviour of the DSF

$$
\begin{aligned}
S(q, \omega) = {}& \frac{2\pi^2\left(1 + \frac{2D}{c}\right)}{|q'|} \rho\left(\gamma' - \frac{q'}{2}\right) \rho_h\left(\gamma' + \frac{q'}{2}\right) \\
& + \frac{8\pi^2}{c^2} \int_{-\infty}^{\infty} \int_{-\infty}^{\infty} \frac{\rho(u)\rho_h(u)}{(\gamma' - \lambda)^2} \Big[ \operatorname{sgn}(u - \lambda)(2\gamma' + u - 3\lambda)\rho(\lambda)\rho_h(\lambda) \\
& \hspace{5cm} - |\gamma' - u|\rho(\gamma')\rho_h(\gamma') \Big] \mathrm{d}\lambda \mathrm{d}u
\end{aligned}
\tag{220}
$$

$$+ o(q^0). \tag{221}$$

We have set

$$\gamma' = \frac{\omega'}{2q'}. \tag{222}$$

Here the term proportional to $1/|q|$ arises only from the one particle-hole contribution, while the constant term is due to two particle-hole excitations. This result can again be compared to GHD predictions, which at order $c^{-2}$ give [85, 86, 90]

$$S_{\mathrm{GHD}}(q, \omega) = \frac{2\pi^2\left(1 + \frac{2D}{c}\right)^2}{|q|} \rho\Big(\gamma(1 + \frac{2D}{c})\Big) \rho_h\Big(\gamma(1 + \frac{2D}{c})\Big). \tag{223}$$

This is indeed in agreement with the leading term in (221) at small $q$. It would be interesting to see whether the subleading terms in (221) can be obtained by considering corrections to GHD following Ref. [114].

### 7.2.6 High frequency tail

Finally we consider the large-$\omega$ behaviour of the DSF $S(q, \omega)$ (at fixed $q$) in an arbitrary eigenstate $|\boldsymbol{\lambda}\rangle$ with a root density $\rho(\lambda)$ that decays faster than any power law $|\lambda|^{-n}$ at infinity. For such states we find

$$S(q, \omega) = \frac{32\sqrt{2}q'^4}{c^2\omega'^{7/2}}(\varepsilon D - \delta^2) + \mathcal{O}(\omega'^{-9/2}), \tag{224}$$

where $\varepsilon = \int u^2 \rho(u) \mathrm{d}u$ is the energy of the state and $\delta$ its momentum defined in (89). The result (224) arises entirely from the two particle-hole contribution since the one particle-hole contribution decays faster than any power in $\omega$ for $\omega \to \infty$. The corrections to this leading behaviour can all be computed and expressed as a series in $\omega'^{-1/2}$. For example the next term is

$$\frac{4\sqrt{2}q'^4}{c^2\omega'^{9/2}} \int_{-\infty}^{\infty} \int_{-\infty}^{\infty} (u-v)^2 [(u-v)^2 + 14q'(u-v) + 15q'^2 + 28q'v] \rho(u)\rho(v) \mathrm{d}u \mathrm{d}v$$
$$+ \mathcal{O}(\omega'^{-11/2}). \tag{225}$$

For eigenstates $|\boldsymbol{\lambda}\rangle$ corresponding to root densities that instead decay like a power law at infinity it is straightforward to see that the one particle-hole contribution to the DSF decays at large $\omega$ with the same power-law. For such root densities the large-$\omega$ expansion of the two particle-hole contribution breaks down at some order because the coefficients would diverge.

It has been shown some time ago that the large $\omega$ behaviour of the DSF $S(q,\omega)$ in equilibrium states is universal, with a decay $\propto q^4 \omega^{-7/2}$ for quantum fluids with short-range interactions [117–119]. This behaviour was also observed to be in good agreement with scattering experiments [118]. Our result (224) is in perfect agreement with these findings, which again confirms that our $1/c$ expansion is indeed uniform in $q, \omega$.

# 8 Conclusions

In this work we have introduced and developed an *ab initio* expansion of dynamical density-density correlation functions in the Lieb-Liniger model that can be performed within any energy eigenstate. It is a combined expansion in $1/c$ and in the number of particle-hole excitations taken into account in the spectral representation of the dynamical correlation function. The expansion has a well-defined thermodynamic limit and is uniform in all $x$ and $t$, or equivalently all $\omega$ and $q$. We have obtained fully explicit and readily usable expressions for both the correlator and the dynamical structure factor at order $\mathcal{O}(c^{-2})$ which take into account all one- and two particle-hole excitations, Equations (171), (166), (178) and (187).

The main obstacle we faced in deriving these results occurs at order $\mathcal{O}(c^{-2})$. Indeed, the leading $\mathcal{O}(c^0)$ term of the expansion is simply the result for impenetrable bosons, which can be straightforwardly obtained using the mapping to free fermions [3, 120]. In terms of the form factor expansion the only non-zero form factors are those involving a single particle-hole excitation, and they are all equal. The $\mathcal{O}(c^{-1})$ term is almost as simple since its form factor expansion is identical to the impenetrable limit case albeit with a root density dependent numerical modification of the form factors. In contrast the $\mathcal{O}(c^{-2})$ contribution comes with a number of complications.

As is well-known the form factor expansion generally exhibits non-integrable singularities whenever two rapidities coincide. In the framework of the $1/c$ expansion these first arise at order $\mathcal{O}(c^{-2})$ for contributions involving both one- and two particle-hole excitations. The presence of such singularities precludes directly taking the thermodynamic limit and expressing the spectral sum as integrals over root densities in a simple way. Indeed, we find that the contributions from both one- and two particle-hole excitations are individually divergent in the thermodynamic limit, but their sum is not. Even after compensating the divergent parts they individually depend on the particular choice of representative state and cannot be expressed in terms of the root densities. But remarkably, and reassuringly, their sum – and thus the correlation function – is representative-state-independent, i.e. depends only on the root

density. These cancellations eventually leave a piece that can be interpreted as a dressing of the contribution due to one particle-hole excitations by two particle-hole excitations. Although this vanishes for the zero-temperature ground state as well as for any zero-entropy states it is non-zero in general and is crucial for detailed balance to be satisfied in thermal states. Such a fine-tuned "regularisation" of the divergences could only be achieved with a careful treatment of the thermodynamic limit of the exact spectral sum in a finite volume. Anticipating that for other quantities the representative-state-dependent parts may not always compensate one another we derived a formula for their average over all representative states for a given root density.

We have verified that our results are in full accord with known results including CFT and (non-linear) Luttinger liquid theory predictions for zero-temperature critical exponents, thresholds singularities, sum rules, detailed balance relations and high frequency behaviour. We have also recovered the order $\mathcal{O}(c^{-2})$ GHD predictions for Euler scale density correlations in finite entropy states. This constitutes the most non-trivial verification of GHD in an interacting integrable model. We have also determined corrections to GHD and compared the GHD result to the full correlator at order $\mathcal{O}(c^{-2})$ outside the asymptotic regime. We found that GHD provides a rather good description of the correlator even at short times and distances.

The framework developed in this work is not restricted to density correlations in the Lieb-Liniger model but is expected to apply to any *local* operator in any integrable model that has a well-behaved expansion around a strong coupling limit. One example is the large anisotropy regime of the spin-1/2 Heisenberg XXZ chain [121,122]. A significant complication that occurs in that case is the presence of string solutions to the Bethe Ansatz equations. The restriction to local operators is crucial as the spectral representation of two-point functions of semi-local operators such as the field $\psi(x)$ are dominated by a completely different set of excited states [78] and does not allow for an expansion in the number of particle-hole-excitations.

Our work opens up several interesting lines of further enquiry. First, our analysis should be extended to higher orders in the expansion. The $\mathcal{O}(c^{-3})$ term still involves at most two particle-hole excitations, but the expansions of the Bethe equations and the determinant in the expression for the form factors become more involved. Second, the repulsive Lieb-Liniger model is particularly simple in that the Bethe equations have only real roots. It would be very interesting to extend our analysis to a model with complex roots, e.g. the spin-1/2 Heisenberg XXZ chain. Third, our framework is readily generalized to quench dynamics [123] by combining it with the quench action approach [91, 92]. Here the novel feature is that the spectral sum involves "overlaps" that multiply the form factors. Finally, it would be interesting to recover results obtained from the $1/c$-expansion considering corrections to GHD as well as the thermodynamic bootstrap program [124].

## Acknowledgements

We are grateful to Jean-Sébastien Caux, Jacopo de Nardis and Karol Kozlowski for very helpful discussions and comments. This work was supported by the EPSRC under grant EP/S020527/1.

# A  Double principal values

## A.1  Proof of Equation (53)

We start by recalling that a single principal value can be expressed as a regular integral

$$\fint \frac{F(\lambda,\mu)}{\lambda-\mu}d\lambda = \frac{1}{2}\int \frac{F(\lambda,\mu)-F(-\lambda+2\mu,\mu)}{\lambda-\mu}d\lambda. \tag{226}$$

Hence successive principal value triple integrals can be written as

$$\fint \frac{F(\lambda,\mu,\nu)}{(\lambda-\mu)(\mu-\nu)}d\lambda d\mu d\nu =$$
$$\frac{1}{4}\iiint \left[\frac{F(\lambda,\mu,\nu)-F(2\mu-\lambda,\mu,\nu)}{(\lambda-\mu)(\mu-\nu)}-\frac{F(\lambda,\mu,2\mu-\nu)-F(2\mu-\lambda,\mu,2\mu-\nu)}{(\lambda-\mu)(\mu-\nu)}\right]d\lambda d\mu d\nu. \tag{227}$$

If $G(\lambda,\mu,\nu)$ is a function without singularities, then we have

$$\iiint G(\lambda,\mu,\nu)d\lambda d\mu d\nu = \lim_{L\to\infty}\frac{1}{L^3}\sum_{i,j,k}G(x_i,x_j,x_k), \tag{228}$$

where

$$x_i = \frac{i}{L}, \tag{229}$$

and $i,j,k$ range e.g. between $-L^2$ and $L^2$. In (228) one has the freedom to exclude some values, e.g. consider $i\neq j$, since this only amounts to subleading corrections in $L$ that vanish when taking the limit. The integrand of (227) is of this type. Hence one can write

$$\fint \frac{F(\lambda,\mu,\nu)}{(\lambda-\mu)(\mu-\nu)}d\lambda d\mu d\nu = \frac{1}{4}\lim_{L\to\infty}\frac{1}{L^3}\sum_{\substack{i,j,k\\i\neq j\\j\neq k}}\Big[\frac{F(x_i,x_j,x_k)-F(-x_i+2x_j,x_j,x_k)}{(x_i-x_j)(x_j-x_k)}$$
$$-\frac{F(x_i,x_j,2x_j-x_k)-F(2x_j-x_i,x_j,2x_j-x_k)}{(x_i-x_j)(x_j-x_k)}\Big]. \tag{230}$$

Separating the four sums and changing variables so that the argument of $f$ is always $x_i,x_j,x_k$ leads to

$$\fint \frac{F(\lambda,\mu,\nu)}{(\lambda-\mu)(\mu-\nu)}d\lambda d\mu d\nu = \lim_{L\to\infty}\frac{1}{L^3}\sum_{\substack{i,j,k\\i\neq j\\j\neq k}}\frac{F(x_i,x_j,x_k)}{(x_i-x_j)(x_j-x_k)}. \tag{231}$$

Finally we turn this into a simultaneous principal value integral by adding the condition $i\neq k$

$$\fint \frac{F(\lambda,\mu,\nu)}{(\lambda-\mu)(\mu-\nu)}d\lambda d\mu d\nu = \fint \frac{F(\lambda,\mu,\nu)}{(\lambda-\mu)(\mu-\nu)}d\lambda d\mu d\nu - \lim_{L\to\infty}\frac{1}{L^3}\sum_{\substack{i,j\\i\neq j}}\frac{F(x_i,x_j,x_i)}{(x_i-x_j)^2}. \tag{232}$$

Using $\sum_{i\neq 0}\frac{1}{i^2}=\frac{\pi^2}{3}$, we have

$$\lim_{L\to\infty}\frac{1}{L^3}\sum_{\substack{i,j\\i\neq j}}\frac{F(x_i,x_j,x_i)}{(x_i-x_j)^2} = \frac{\pi^2}{3}\int F(x,x,x)dx, \tag{233}$$

and obtain Equation (53).

## A.2 Proof of Equation (56)

We note that formulae (55) are direct consequences of Equation (56), as can be seen by interchanging the dummy variables.

We start with representation (227). As the integrand is regular one can impose that $|\lambda - \mu| > \epsilon$ and $|\mu - \nu| > \epsilon'$ with an error $\mathcal{O}(\epsilon) + \mathcal{O}(\epsilon')$. This allows one to separate the integral into four pieces and make appropriate changes of variables so that the argument of $F$ is always $\lambda', \mu', \nu'$. One sees that in the four cases one has $|\lambda' - \mu'| > \epsilon$ and $|\mu' - \nu'| > \epsilon'$. Hence

$$\int \frac{F(\lambda, \mu, \nu)}{(\lambda - \mu)(\mu - \nu)} d\lambda d\mu d\nu = \int_{\substack{|\lambda' - \mu'| > \epsilon \\ |\nu' - \mu'| > \epsilon'}} \frac{F(\lambda', \mu', \nu')}{(\lambda' - \mu')(\mu' - \nu')} d\lambda' d\mu' d\nu' + \mathcal{O}(\epsilon) + \mathcal{O}(\epsilon'), \quad (234)$$

which is precisely (56).

# B Proof of Equation (41)

## B.1 Reduction to a combinatorial problem

For a given solution to the Bethe equations $\{\lambda_i\}_i \in \mathfrak{S}_L$ we define the set of pairs of rapidities that belong to the same bin

$$B = \left\{ (\lambda_i, \lambda_j) \,\middle|\, i \neq j \,, \, \exists k \in \{1, ..., n_L\}, \, \lambda_i, \lambda_j \in [x_{L,k}, x_{L,k+1}] \right\}. \quad (235)$$

We have

$$\frac{1}{L^3} \sum_{i \neq j} \frac{f(\lambda_i, \lambda_j)}{(\lambda_i - \lambda_j)^2} = \frac{1}{L^3} \sum_{(\lambda_i, \lambda_j) \in B} \frac{f(\lambda_i, \lambda_j)}{(\lambda_i - \lambda_j)^2} + \frac{1}{L^3} \sum_{(\lambda_i, \lambda_j) \notin B} \frac{f(\lambda_i, \lambda_j)}{(\lambda_i - \lambda_j)^2}. \quad (236)$$

Let us show that when the pairs of rapidities are not in $B$, the sum is negligible. We observe that

$$\frac{1}{L^3} \sum_{\substack{\lambda_i \in [x_{L,k}, x_{L,k+1}] \\ \lambda_j \in [x_{L,p}, x_{L,p+1}]}} \frac{|f(\lambda_i, \lambda_j)|}{(\lambda_i - \lambda_j)^2} \leq \frac{(L\epsilon_L)^2}{(|k - p| - 1)^2 L^3} \frac{\max_{\lambda, \lambda'} |f(\lambda, \lambda')|}{C_0^2 \epsilon_L^2}, \quad (237)$$

provided that $|k - p| > 1$, i.e. if the bins to which $\lambda_i$ and $\lambda_j$ belong are not adjacent. Here $C_0 = \min_y [\rho(y)]^{-1}$ is a constant independent of the representative state and of the bins. Indeed, in this case we have $|\lambda_i - \lambda_j| > (|k - p| - 1) C_0 \epsilon_L$ and there are $(L\epsilon_L)^2$ pairs of roots. Since there are $D/\epsilon_L$ bins, by summing over $p$ and $k$ these contributions are $\mathcal{O}(\frac{1}{L\epsilon_L})$, and since $L\epsilon_L \to \infty$, they are negligible in the thermodynamic limit.

If the bins are adjacent we have

$$\frac{1}{L^3} \sum_{\substack{\lambda_i \in [x_{L,k}, x_{L,k+1}] \\ \lambda_j \in [x_{L,k+1}, x_{L,k+2}]}} \frac{|f(\lambda_i, \lambda_j)|}{(\lambda_i - \lambda_j)^2} \leq \frac{C_1}{L^3} \sum_{\substack{0 \leq n,m \leq L\epsilon_L \\ n+m \neq 0}} \frac{1}{(n+m)^2/L^2} = \mathcal{O}(\tfrac{\log(L\epsilon_L)}{L}), \quad (238)$$

with $C_1$ another constant independent of the representative state and of the bins. Since there are $D/\epsilon_L$ bins and $L\epsilon_L \to \infty$, these contributions are also negligible in the thermodynamic limit. Hence we have

$$\frac{1}{L^3} \sum_{i \neq j} \frac{f(\lambda_i, \lambda_j)}{(\lambda_i - \lambda_j)^2} = \frac{1}{L^3} \sum_{(\lambda_i, \lambda_j) \in B} \frac{f(\lambda_i, \lambda_j)}{(\lambda_i - \lambda_j)^2} + o(L^0), \quad (239)$$

with the $o(L^0)$ being independent of the representative state. Hence we also have

$$\lim_{L\to\infty}\frac{1}{|\mathfrak{S}_L|}\sum_{\{\lambda_i\}_i\in\mathfrak{S}_L}\frac{1}{L^3}\sum_{i\neq j}\frac{f(\lambda_i,\lambda_j)}{(\lambda_i-\lambda_j)^2}=\lim_{L\to\infty}\frac{1}{|\mathfrak{S}_L|}\sum_{\{\lambda_i\}_i\in\mathfrak{S}_L}\frac{1}{L^3}\sum_{(\lambda_i,\lambda_j)\in B}\frac{f(\lambda_i,\lambda_j)}{(\lambda_i-\lambda_j)^2}.\tag{240}$$

Writing

$$\sum_{(\lambda_i,\lambda_j)\in B}\frac{f(\lambda_i,\lambda_j)}{(\lambda_i-\lambda_j)^2}=\sum_{k=1}^{n_L}\sum_{\substack{i\neq j\\\lambda_i,\lambda_j\in[x_{L,k},x_{L,k+1}]}}\frac{f(\lambda_i,\lambda_j)}{(\lambda_i-\lambda_j)^2},\tag{241}$$

we have

$$\frac{1}{|\mathfrak{S}_L|}\sum_{\{\lambda_i\}_i\in\mathfrak{S}_L}\frac{1}{L^3}\sum_{(\lambda_i,\lambda_j)\in B}\frac{f(\lambda_i,\lambda_j)}{(\lambda_i-\lambda_j)^2}=\sum_{k=1}^{n_L}\frac{1}{|\mathfrak{S}_L|}\sum_{\{\lambda_i\}_i\in\mathfrak{S}_L}\frac{1}{L^3}\sum_{\substack{i\neq j\\\lambda_i,\lambda_j\in[x_{L,k},x_{L,k+1}]}}\frac{f(\lambda_i,\lambda_j)}{(\lambda_i-\lambda_j)^2}.\tag{242}$$

To go further and decouple the average over the representative states one needs to ensure that the modification of rapidities in one bin does not notably affect the distance between rapidities in another bin. From the Bethe equations, a modification of order $\epsilon_L$ of $DL$ rapidities modifies the distance between two rapidities $i,j$ in the same bin by an order $\frac{1}{L}DL\epsilon_L(\lambda_i-\lambda_j)$, which is indeed subleading compared to $\lambda_i-\lambda_j$. Hence one can assume at leading order in $L$ that the rapidities decouple, and given a bin $k$, sum over the rapidities of the other bins without modifying the values of the rapidities inside bin $k$. Thus one can write

$$\frac{1}{|\mathfrak{S}_L|}\sum_{\{\lambda_i\}_i\in\mathfrak{S}_L}\frac{1}{L^3}\sum_{\substack{i\neq j\\\lambda_i,\lambda_j\in[x_{L,k},x_{L,k+1}]}}\frac{f(\lambda_i,\lambda_j)}{(\lambda_i-\lambda_j)^2}$$
$$=\frac{1}{\binom{K_{L,k}}{\lfloor L\epsilon_L\rfloor}}\sum_{\{\lambda_i\}_i\in\mathfrak{S}_L^k}\frac{1}{L^3}\sum_{\substack{i\neq j\\\lambda_i,\lambda_j\in[x_{L,k},x_{L,k+1}]}}\frac{f(\lambda_i,\lambda_j)}{(\lambda_i-\lambda_j)^2}+o(L^0),\tag{243}$$

with $\mathfrak{S}_L^k\subset\mathfrak{S}_L$ the subset of $\mathfrak{S}_L$ containing states whose rapidities outside the bin $[x_{L,k},x_{L,k+1}]$ are fixed to those of an arbitrary representative state, and

$$K_{L,i}=\lfloor L(x_{L,i+1}-x_{L,i})(\rho(x_{L,i})+\rho_h(x_{L,i}))\rfloor\tag{244}$$

is the number of vacancies in $[x_{L,i},x_{L,i+1}]$. Since around $\lambda$ two consecutive vacancies are separated by $\frac{1}{L(\rho(\lambda)+\rho_h(\lambda))}$ at leading order in $L$, we can write $\lambda_i-\lambda_j$ as an integer times $\frac{1}{L(\rho(x_{L,k})+\rho_h(x_{L,k}))}$, for $\lambda_i$ and $\lambda_j$ in the same bin $[x_{L,k},x_{L,k+1}]$. This yields

$$\frac{1}{|\mathfrak{S}_L|}\sum_{\{\lambda_i\}_i\in\mathfrak{S}_L}\frac{1}{L^3}\sum_{(\lambda_i,\lambda_j)\in B}\frac{f(\lambda_i,\lambda_j)}{(\lambda_i-\lambda_j)^2}=$$
$$\frac{1}{L}\sum_{k=1}^{n_L}\frac{f(x_{L,k},x_{L,k})}{\binom{K_{L,k}}{\lfloor L\epsilon_L\rfloor}}(\rho(x_{L,k})+\rho_h(x_{L,k}))^2\sum_{\substack{I\subset\{1,\dots,K_{L,k}\}\\|I|=\lfloor L\epsilon_L\rfloor}}\sum_{\substack{i,j\in I\\i\neq j}}\frac{1}{(i-j)^2}$$
$$+o(L^0).\tag{245}$$

This reduces the problem to evaluating the large $K,M$ limit at fixed $K/M$ of the following combinatorial quantity

$$\mathcal{C}_{M,K}=\sum_{\substack{I\subset\{1,\dots,K\}\\|I|=M}}\sum_{\substack{i,j\in I\\i\neq j}}\frac{1}{(i-j)^2}.\tag{246}$$

## B.2 The generating functions

To simplify the expression of $\mathcal{C}_{M,K}$, we would like to recast the sum over pairs of integers into a sum over (next-nearest-)neighbouring integers. We exactly rewrite $\mathcal{C}_{M,K}$ in the form

$$\mathcal{C}_{M,K} = 2 \sum_{m=1}^{M} \sum_{j=1}^{m} \sum_{\substack{a_1 < \ldots < a_M \\ \in \{1,\ldots,K\}}} \sum_{\substack{i \geq 0 \\ j+(i+1)m \leq M}} \frac{1}{(a_{j+im} - a_{j+(i+1)m})^2}. \tag{247}$$

Introducing

$$C_{M,K}^{[m]} = \sum_{\substack{a_1 < \ldots < a_M \\ \in \{1,\ldots,K\}}} \sum_{\substack{i \geq 0 \\ (i+1)m \leq M}} \frac{1}{(a_{im} - a_{(i+1)m})^2} \tag{248}$$

with $a_0 = 0$, we have

$$\mathcal{C}_{M,K} = 2 \sum_{m=1}^{M} \sum_{j=1}^{m} \sum_{a=j}^{K} \binom{a-1}{j-1} C_{M-j,K-a}^{[m]}. \tag{249}$$

Let us now determine the asymptotic behaviour of $C_{M,K}^{[m]}$ for large $K, M$ at fixed $K/M$. Summing separately over $a_1, \ldots, a_m$, one obtains the following recurrence relation

$$C_{M,K}^{[m]} = \sum_{a_m=m}^{K-M+m} \binom{a_m-1}{m-1} \left[ \frac{\binom{K-a_m}{M-m}}{a_m^2} + C_{M-m,K-a_m}^{[m]} \right], \tag{250}$$

where we use conventions such that $C_{M,K}^{[m]} = 0$ if $K < M$ or $M < m$. Indeed, the factor $\binom{a_m-1}{m-1}$ counts the number of possibilities for the first $m-1$ particles between 1 and $a_m - 1$, while the factor $\binom{K-a_m}{M-m}$ counts the number of times this $1/a_m^2$ term will appear in all the subsequent configurations for $a_{m+1}, \ldots, a_M$. Introducing the generating functions

$$C^{[m]}(x,y) = \sum_{M,K \geq 0} C_{M,K}^{[m]} x^M y^K, \qquad S^{[m]}(x,y) = \sum_{M,K \geq 0} \sum_{a=m}^{K-M+m} \binom{a-1}{m-1} \frac{\binom{K-a}{M-m}}{a^2} x^M y^K, \tag{251}$$

this recurrence relation implies that

$$C^{[m]}(x,y) = S^{[m]}(x,y) + \frac{x^m y^m}{(1-y)^m} C^{[m]}(x,y). \tag{252}$$

Expressing $S^{[m]}(x,y)$ as

$$S^{[m]}(x,y) = \frac{x^m}{1-y(1+x)} \sum_{a \geq 1} \frac{\binom{a-1}{m-1}}{a^2} y^a, \tag{253}$$

we obtain the following generating function

$$C^{[m]}(x,y) = \frac{x^m (1-y)^m}{(1-y(1+x))^2 \sum_{k=0}^{m-1} (xy)^k (1-y)^{m-1-k}} \sum_{a \geq 1} \frac{\binom{a-1}{m-1}}{a^2} y^a. \tag{254}$$

## B.3 Asymptotics of the coefficients

We now use Ref. [84] which shows how to determine the asymptotic behaviour of combinatorial coefficients from the analytic behaviour of their generating function[2]. One obtains

$$C_{M,K}^{[m]} = M \binom{K}{M} \left( \frac{M/K}{1-M/K} \right)^m \frac{1}{m} \sum_{a \geq 1} \frac{(1-M/K)^a}{a^2} \binom{a-1}{m-1} + \mathcal{O}\left( \binom{K}{M} \right). \tag{257}$$

---

[2]Specifically, in order to have only a simple pole in the generating function as in [84], we define

$$\bar{C}^{[m]}(x,y) = \int_0^x C^{[m]}(u,y) \mathrm{d}u. \tag{255}$$

This implies that

$$C_{M-j,K-a}^{[m]} = (\tfrac{M}{K})^j(1-\tfrac{M}{K})^{a-j}C_{M,K}^{[m]} + \mathcal{O}(\binom{K}{M}),\tag{258}$$

and substituting this into (249) we obtain at leading order

$$\mathcal{C}_{M,K} = \left[2\sum_{m=1}^{\infty}mC_{M,K}^{[m]}\right](1+\mathcal{O}(M^{-1})).\tag{259}$$

Using the asymptotics (257) one then finds in the limit $M, K \to \infty$ at fixed $K/M$

$$\mathcal{C}_{M,K} = \frac{\pi^2}{3}M\frac{M}{K}\binom{K}{M} + \mathcal{O}(\binom{K}{M}).\tag{260}$$

## B.4   Conclusion

Coming back to (245), we have when $L \to \infty$

$$M \sim L\epsilon_L, \qquad K \sim L\epsilon_L \frac{\rho(x_{L,k}) + \rho_h(x_{L,k})}{\rho(x_{L,k})},\tag{261}$$

which yields

$$\frac{1}{|\mathfrak{S}_L|}\sum_{\{\lambda_i\}_i\in\mathfrak{S}_L}\frac{1}{L^3}\sum_{(\lambda_i,\lambda_j)\in B}\frac{f(\lambda_i,\lambda_j)}{(\lambda_i-\lambda_j)^2}$$
$$= \epsilon_L\frac{\pi^2}{3}\sum_{k=1}^{n_L}f(x_{L,k},x_{L,k})(\rho(x_{L,k})+\rho_h(x_{L,k}))\rho(x_{k,L}) + o(L^0).\tag{262}$$

In the limit $L \to \infty$ we then arrive at the result (41)

$$\lim_{L\to\infty}\frac{1}{|\mathfrak{S}_L|}\sum_{\{\lambda_i\}_i\in\mathfrak{S}_L}\frac{1}{L^3}\sum_{i\neq j}\frac{f(\lambda_i,\lambda_j)}{(\lambda_i-\lambda_j)^2} = \frac{\pi^2}{3}\int_{-\infty}^{\infty}f(\lambda,\lambda)(\rho(\lambda)+\rho_h(\lambda))\rho(\lambda)^2\mathrm{d}\lambda.\tag{263}$$

# C   Derivations of the results presented in Section 7

## C.1   Correlation functions

### C.1.1   Asymptotics of static correlators at zero-temperature

The study of the asymptotic behaviour of (173) at large $x$ at zero temperature reduces to the asymptotics of the Fourier transform

$$\hat{f}(x) = \int_{-\infty}^{\infty}f(u)e^{-ixu}\mathrm{d}u,\tag{264}$$

---

We then integrate by parts

$$\bar{C}^{[m]}(x,y) = \frac{x^m(1-y)^m}{(1-y(1+x))\sum_{k=0}^{m-1}(xy)^k(1-y)^{m-1-k}}\sum_{a\geq 1}\frac{\binom{a-1}{m-1}}{a^2}y^{a-1}$$
$$-\int_0^x\mathrm{d}u\frac{1}{1-y(1+u)}\frac{d}{du}\frac{u^m(1-y)^m}{\sum_{k=0}^{m-1}(uy)^k(1-y)^{m-1-k}}\sum_{a\geq 1}\frac{\binom{a-1}{m-1}}{a^2}y^{a-1}\tag{256}$$

and use Theorem 1.3 and Corrolary 3.21 of [84] on the first term, where in their notations $x = \frac{M/K}{1-M/K}$ and $y = 1-M/K$. The second term gives negligible contributions because the $x$ integral will give rise to a multiplicative factor $M^{-1}$.

of a given function $f(u)$. These asymptotics depend on the regularity of the integrand, hence at leading order on points of non-analyticity of $f$ on the real axis. We have the following behaviours.

- If $f(u)$ has a discontinuity $\Delta = f(u_0^+) - f(u_0^-)$ at $u_0$ and is otherwise regular, then for $x \to \infty$

$$\hat{f}(x) = \Delta \frac{e^{-ixu_0}}{ix} + \mathcal{O}(x^{-2}). \tag{265}$$

This is straightforwardly obtained with an integration by part.

- If $f(u) \sim \Delta \log^n |u - u_0|$ for $u > u_0$ and is regular and bounded for $u < u_0$, then

$$\hat{f}(x) = \begin{cases} -\Delta \frac{e^{-ixu_0}}{ix}(\log|x| + p_1) + o(\frac{1}{x}) & \text{if } n = 1 \\ \Delta \frac{e^{-ixu_0}}{ix}(\log^2|x| + 2p_1 \log|x| + p_2) + o(\frac{1}{x}) & \text{if } n = 2. \end{cases} \tag{266}$$

Here the constants $p_{1,2}$ are given by

$$\begin{aligned} p_1 &= \gamma_E + \frac{i\pi}{2}\operatorname{sgn}(x) \\ p_2 &= \gamma_E^2 + i\pi\gamma_E\operatorname{sgn}(x) - \frac{\pi^2}{12}, \end{aligned} \tag{267}$$

where $\gamma_E$ is Euler's gamma constant. If $f(u) \sim \Delta \log^n |u - u_0|$ for $u < u_0$ and is regular and bounded for $u > u_0$, then the result is multiplied by $-1$ and $p_1, p_2$ are changed to their complex conjugates $p_1^*, p_2^*$.

These relations are obtained from the relation

$$\int_0^\infty e^{-ixu}u^\alpha \mathrm{d}u = \Gamma(1+\alpha)[i(x-i0)]^{-1-\alpha}, \tag{268}$$

expanded around $\alpha = 0$.

- If $f(u) \sim \Delta \log^n |u - u_0|$ for both $u < u_0$ and $u > u_0$, then we have

$$\hat{f}(x) = \begin{cases} -\pi\Delta \frac{e^{-ixu_0}}{x} + o(\frac{1}{x}) & \text{if } n = 1 \\ 2\pi\Delta \frac{e^{-ixu_0}}{x}(\log|x| + \gamma_E) + o(\frac{1}{x}) & \text{if } n = 2. \end{cases} \tag{269}$$

These equations directly follow from the previous results.

In order to determine the large $x$ behaviour of the correlation functions, we also need zero-temperature result for $\tilde{\rho}(x)$ defined in (114)

$$\tilde{\rho}(\lambda) = \frac{1 + 2D/c}{2\pi} \log\left|\frac{\lambda + Q}{\lambda - Q}\right|, \tag{270}$$

and the large $x$ behaviour of the functions $A_{x,0}$, $C_{x,0}$ and $D_{x,0}$ defined in (147), (155) and (158) respectively

$$\begin{aligned} A_{x,0} &= -\frac{\log|x|}{2\pi^2} + \mathcal{O}(x^0) \\ C_{x,0} &= o(x^{-1}) \\ D_{x,0} &= o(x^{-1}). \end{aligned} \tag{271}$$

The asymptotics of $C_{x,0}$ and $D_{x,0}$ follow from (300) and (301) for a generic root density. As for $A_{x,0}$, integrating (147) by parts we obtain for a generic root density

$$A_{x,0} = \int \frac{\rho(u)\rho'_h(v)}{v-u}(e^{ix(v-u)}-1)\mathrm{d}u\mathrm{d}v + ixC_{x,0}. \tag{272}$$

Specializing to zero temperature at leading order in $c^{-1}$, it yields

$$A_{x,0} = \frac{1}{4\pi^2}\int_{-Q}^{Q}\frac{e^{ix(Q-u)}-1}{Q-u}\mathrm{d}u - \frac{1}{4\pi^2}\int_{-Q}^{Q}\frac{e^{ix(-Q-u)}-1}{-Q-u}\mathrm{d}u + ixC_{x,0}, \tag{273}$$

that is

$$\begin{aligned} A_{x,0} &= \frac{1}{4\pi^2}\int_{0}^{2Qx}\frac{e^{iu}+e^{-iu}-2}{u}\mathrm{d}u + ixC_{x,0} \\ &= -\frac{\log|x|}{2\pi^2} + \mathcal{O}(x^0). \end{aligned} \tag{274}$$

As for the $B_{x,0}(\lambda)$ and $B_{x,0}(\mu)$ terms, they require a special treatment since they cannot be decoupled from the $\lambda, \mu$ integrals. The $B_{x,0}(\lambda)$ term involves the following functions

$$f_n(x) = \int_{-\infty}^{\infty}\rho(\lambda)\lambda^n B_{x,0}(\lambda)e^{-i\lambda x}\mathrm{d}\lambda, \tag{275}$$

for $n=0,1,2$, whose we wish to determine the asymptotic behaviour at large $x$, by computing its Fourier transform $\hat{f}_n(q) = \int_{-\infty}^{\infty}e^{-iqx}f(x)\mathrm{d}x$. We have (at leading order in $c^{-1}$)

$$\hat{f}_n(q) = 2\pi\int_{-\infty}^{\infty}\int_{-\infty}^{\infty}\rho(\lambda)\rho(u)\lambda^n\frac{\frac{1}{2\pi}-\rho(u+\lambda+q)}{(\lambda+q)(\lambda-u)}\mathrm{d}u\mathrm{d}\lambda. \tag{276}$$

Specializing this relation to the ground state root density we obtain

$$\hat{f}_n(q) = \frac{1}{4\pi^2}\int_{-Q}^{Q}\frac{\lambda^n}{|\lambda+q|}\log\left|\frac{\lambda+\min(Q,-Q+|\lambda+q|)\,\mathrm{sgn}(\lambda+q)}{\lambda-Q\,\mathrm{sgn}(\lambda+q)}\right|\mathrm{d}\lambda. \tag{277}$$

We note that the non-integrable divergence near $\lambda = q$ is compensated by the argument of the log going to 1 in this limit. In the vicinity of $q = Q$ we have for $\eta > 0$

$$\begin{aligned} \hat{f}_n(Q+\eta) &= \frac{1}{4\pi^2}\int_{-Q}^{Q}\frac{\lambda^n}{\lambda+Q}\log\left|\frac{2\lambda}{\lambda-Q}\right|\mathrm{d}\lambda + o(\eta^0) \\ \hat{f}_n(Q-\eta) &= \frac{1}{4\pi^2}\int_{-Q}^{Q}\frac{\lambda^n}{\lambda+Q}\log\left|\frac{2\lambda}{\lambda-Q}\right|\mathrm{d}\lambda + \frac{(-Q)^n}{4\pi^2}\int_{0}^{1}\frac{1}{v-1}\log\left|\frac{v}{2v-1}\right|\mathrm{d}v + o(\eta^0), \end{aligned} \tag{278}$$

where the last integral is $\int_{0}^{1}\frac{1}{v-1}\log\left|\frac{v}{2v-1}\right|\mathrm{d}v = -\frac{\pi^2}{12}$, so that $\hat{f}_n$ has a discontinuity at $Q$ of

$$\lim_{\eta\to0}\left[\hat{f}_n(Q+\eta)-\hat{f}_n(Q-\eta)\right] = \frac{(-Q)^n}{48}. \tag{279}$$

Similarly we find

$$\lim_{\eta\to0}\left[\hat{f}_n(-Q+\eta)-\hat{f}_n(-Q-\eta)\right] = -\frac{Q^n}{48}, \tag{280}$$

and $\hat{f}_n(q)$ does not have discontinuities elsewhere. This implies that

$$
\begin{aligned}
f_0(x) &= -\frac{1}{48\pi}\frac{\sin(Qx)}{x} + o(x^{-1}) \\
f_1(x) &= -\frac{iQ}{48\pi}\frac{\cos(Qx)}{x} + o(x^{-1}) \\
f_2(x) &= -\frac{Q^2}{48\pi}\frac{\sin(Qx)}{x} + o(x^{-1}).
\end{aligned}
\tag{281}
$$

Then one builds the full $B_{x,0}(\lambda)$ term from the functions (275), in particular by taking into account the remaining oscillatory $\mu$ integral. One obtains that the $B_{x,0}(\lambda)$ term gives contributions that decay as $\cos(2q_F x)/x^2$ and of order $c^{-2}$, which are encapsulated in the result in the $\mathcal{O}(c^{-2})$ term of the $A$ in (190). A similar analysis shows that the $B_{x,0}(\mu)$ term also gives contributions that decay as $\cos(2q_F x)/x^2$.

From these various relations, it is straightforward albeit tedious to determine the asymptotics of the static correlation functions. Putting everything together we find that $\chi_{x,0}^{(1,2)}(\lambda,\mu)$ given in (166) contributes to the large-$x$ behaviour of the density-density correlator as follows:

- $\mathcal{O}(c^0)$ contribution of $\chi_{x,0}^{(1)}(\lambda,\mu)$

$$
-\frac{1+\frac{4D}{c}+\frac{4D^2}{c^2}}{2\pi^2 x^2}(1-\cos(2q_F x)),
\tag{282}
$$

- $\mathcal{O}(c^{-1})$ contribution of $\chi_{x,0}^{(1)}(\lambda,\mu)$

$$
-\frac{4D(1+\frac{4D}{c})}{c\pi^2}\frac{\cos(2q_F x)}{x^2}\log|2q_F e^{\gamma_E}x| + o(x^{-2}),
\tag{283}
$$

- $\mathcal{O}(c^{-2})$ contribution of $\chi_{x,0}^{(1)}(\lambda,\mu)$

$$
\frac{16D^2}{\pi^2 c^2}\frac{\cos(2q_F x)}{x^2}\log^2|2q_F e^{\gamma_E}x| + \mathcal{O}(\frac{\cos 2q_F x}{x^2}),
\tag{284}
$$

- Contribution of $\chi_{x,0}^{(2)}(\lambda,\mu)$

$$
-\frac{4D^2}{\pi^2 c^2}\frac{\cos(2q_F x)}{x^2}\log|2q_F e^{\gamma_E}x| + \mathcal{O}(\frac{\cos 2q_F x}{x^2}).
\tag{285}
$$

This establishes (189).

### C.1.2 Asymptotics of dynamical correlations zero temperature

The study of the asymptotic behaviour of (164) at large $x,t$ at fixed $\alpha = \frac{x}{2t}$ at zero temperature reduces to the study of an oscillatory integral of the type

$$
I(x,t) = \int_{-\infty}^{\infty} f(u)e^{itu^2-ix'u}\mathrm{d}u.
\tag{286}
$$

In this regime, the integral is dominated by the point where the phase has an extremum as a function of $u$, which is $\alpha'$ defined in (195). If $f$ is regular and $\alpha'$ in the support of $f$, then we have

$$
I(x,t) = \frac{\sqrt{\pi}e^{i\operatorname{sgn}(t)\pi/4}e^{-i\alpha'^2 t}}{\sqrt{|t|}}\left(f(\alpha') + \frac{i}{4t}f''(\alpha') - \frac{1}{32t^2}f''''(\alpha')\right) + \mathcal{O}(|t|^{-7/2}).
\tag{287}
$$

If $f$ has singular points one has to combine (287) with the results of Section C.1.1.

The correlation function (164) is expressed as a double integral, one over $\lambda$ with a factor $\rho(\lambda)$ and one over $\mu$ with a factor $\rho_h(\mu)$. Because of the very particular structure of $\rho(\lambda)$ at zero temperature (106), the saddle point $\alpha$ necessarily lies within the support of either $\rho(\lambda)$ or $\rho_h(\mu)$, but not both. Hence if $|\alpha| > q_F$, the $\lambda$ integral is dominated by boundary effects as in the static case, while the $\mu$ integral is dominated by the saddle point. If $|\alpha| < q_F$, the converse holds true.

Let us detail the case $|\alpha| > q_F$ (the case $|\alpha| < q_F$ is very similar). We perform a change of variables $\lambda \to \lambda + \alpha'$ and $\mu \to \mu + \alpha'$ in (164) in order to move the saddle point to 0, which results in shifting the arguments of the root densities by $\alpha'$. The $\mu$ integral is then simply evaluated at $\mu = 0$, while the $\lambda$ integral is dominated by the vicinities of the points $Q - \alpha'$ and $-Q - \alpha'$. Using the results for (264) with $x = -2t(Q - \alpha')$ we obtain the leading contribution from $Q - \alpha'$ to the integral over $\chi_{x,t}^{(1)}(\lambda, \mu)$, with $\pm = \text{sgn}(t)$

$$\frac{e^{-i\,\text{sgn}(t)\frac{\pi}{4}}\left(1+\frac{2D}{c}\right)^4}{4\pi^{\frac{3}{2}}|t|^{\frac{1}{2}}} \frac{e^{it(Q-\alpha')^2}}{2it(Q-\alpha')}\left[1 - \frac{4}{c}\frac{1+\frac{2D}{c}}{2\pi}(Q-\alpha')\left[\log\left|4Qt\frac{(Q-\alpha')^2}{Q+\alpha'}\right| + \gamma_E \mp i\frac{\pi}{2}\right]\right.$$
$$\left. + \frac{8}{c^2}\left(\frac{1+\frac{2D}{c}}{2\pi}\right)^2 (Q-\alpha')^2\left[\log\left|4Qt\frac{(Q-\alpha')^2}{Q+\alpha'}\right| + \gamma_E \mp i\frac{\pi}{2}\right]^2 + \mathcal{O}(t^0 c^{-2})\right]. \quad (288)$$

The leading contribution from $-Q - \alpha'$ to the integral over $\chi_{x,t}^{(1)}(\lambda, \mu)$ is obtained analogously

$$\frac{e^{-i\,\text{sgn}(t)\frac{\pi}{4}}\left(1+\frac{2D}{c}\right)^4}{4\pi^{\frac{3}{2}}|t|^{\frac{1}{2}}} \frac{e^{it(Q+\alpha')^2}}{2it(Q+\alpha')}\left[1 - \frac{4}{c}\frac{1+\frac{2D}{c}}{2\pi}(Q+\alpha')\left[\log\left|4Qt\frac{(Q+\alpha')^2}{Q-\alpha'}\right| + \gamma_E \pm i\frac{\pi}{2}\right]\right.$$
$$\left. + \frac{8}{c^2}\left(\frac{1+\frac{2D}{c}}{2\pi}\right)^2 (Q+\alpha')^2\left[\log\left|4Qt\frac{(Q+\alpha')^2}{Q-\alpha'}\right| + \gamma_E \pm i\frac{\pi}{2}\right]^2 + \mathcal{O}(t^0 c^{-2})\right]. \quad (289)$$

In order to determine the asymptotic behaviour of the two particle-hole contribution $\chi_{x,t}^{(2)}(\lambda, \mu)$ we require the asymptotic behaviours of the functions $A_{2\alpha't,t}$, $C_{2\alpha't,t}$ and $D_{2\alpha't,t}$ defined in (147), (155) and (158) respectively. We find

$$A_{2\alpha't,t} = -\frac{\log|t|}{2\pi^2} + \mathcal{O}(t^0),$$
$$C_{2\alpha't,t} = o(t^{-1}),$$
$$D_{2\alpha't,t} = o(t^{-1}). \quad (290)$$

The results for $C_{2\alpha't,t}$ and $D_{2\alpha't,t}$ again follow from (300) and (301) for a generic root density. As for $A_{2\alpha't,t}$, we integrate by parts to express it in the form

$$A_{2\alpha't,t} = \int \frac{\rho(u+\alpha')\rho_h'(v+\alpha')}{v-u}(e^{it(u^2-v^2)}-1)\mathrm{d}u\mathrm{d}v$$
$$- 2it\iint \rho(u+\alpha')\rho_h(v+\alpha')e^{it(u^2-v^2)}\mathrm{d}u\mathrm{d}v - 2itD_{2\alpha't,t}. \quad (291)$$

A saddle point approximation on the second double integral shows that the second line is $\mathcal{O}(t^0)$. Specializing to zero temperature we then have at leading order in $c^{-1}$

$$A_{2\alpha't,t} = \frac{1}{4\pi^2}\int_0^{2Q}\frac{\mathrm{d}u}{u}[e^{it(u^2-2u(Q-\alpha))}-1]$$
$$+ \frac{1}{4\pi^2}\int_0^{2Q}\frac{\mathrm{d}u}{u}[e^{it(u^2-2u(Q+\alpha))}-1] + \mathcal{O}(t^0), \quad (292)$$

which can be further simplified

$$A_{2\alpha't,t} = \frac{1}{4\pi^2}\int_0^{2Qt}\frac{du}{u}[e^{i(u^2/t-2u(Q-\alpha))}-1] + \frac{1}{4\pi^2}\int_0^{2Qt}\frac{du}{u}[e^{i(u^2/t-2u(Q+\alpha))}-1] + \mathcal{O}(t^0)$$

$$= -\frac{\log|t|}{2\pi^2} + \mathcal{O}(t^0). \tag{293}$$

Finally there are the contributions of the $B_{x,t}(\lambda)$ and $B_{x,t}(\mu)$ terms. One can perform an analysis similar to the static case and obtain that they are both $\mathcal{O}(t^{-3/2})$. Their contributions are encapsulated in the result in the $\mathcal{O}(c^{-2})$ term of the $B_{\pm}$ in (197). Putting everything together we obtain the leading contribution from the vicinity of $Q - \alpha'$ to the double integral over $\chi_{x,t}^{(2)}(\lambda,\mu)$

$$\frac{4}{c^2}\frac{e^{-\mathrm{sgn}(t)i\pi/4}\left(1+\frac{2D}{c}\right)^4}{4\pi^{3/2}|t|^{1/2}}\frac{e^{it(Q-\alpha')^2}}{2it(Q-\alpha')}(Q-\alpha)^2\left[-\frac{\log|t|}{2\pi^2}+\mathcal{O}(t^0)\right]. \tag{294}$$

The analogous result for the contribution from the vicinity of $-Q - \alpha'$ is

$$\frac{4}{c^2}\frac{e^{-\mathrm{sgn}(t)i\pi/4}\left(1+\frac{2D}{c}\right)^4}{4\pi^{3/2}|t|^{1/2}}\frac{e^{it(Q+\alpha')^2}}{2it(Q+\alpha')}(Q+\alpha)^2\left[-\frac{\log|t|}{2\pi^2}+\mathcal{O}(t^0)\right]. \tag{295}$$

### C.1.3   Euler scale asymptotic behaviour

In this section we will assume $\rho$ to be continuous. If it is not continuous the leading $1/t$ behaviour is unchanged, but the $1/t^2$ corrections might differ.

For a generic continuous root density at large $x, t$ and fixed $\alpha = \frac{x}{2t}$, the two integrals over $\lambda$ and $\mu$ in the correlation function (164) are both dominated by the saddle point at $\alpha'$. Applying (287) to the one particle-hole contribution gives

$$\frac{\pi\left(1+\frac{2D}{c}\right)^2\rho(\alpha')\rho_h(\alpha')}{|t|} + \frac{i\pi\left(1+\frac{2D}{c}\right)^2\left[\rho''(\alpha')\rho_h(\alpha')-\rho(\alpha')\rho_h''(\alpha')\right]}{4t|t|} + \mathcal{O}(t^{-3}). \tag{296}$$

The contribution due to two particle-hole excitations is more subtle and requires determining the asymptotic behaviour of oscillatory integrals with principal values, whose saddle point falls on the singularity. The general strategy is to write each singularity as

$$\frac{1}{\lambda-\mu} = \frac{t}{2i}\int_{-\infty}^{\infty}\mathrm{sgn}(\xi)e^{it\xi(\lambda-\mu)}d\xi, \tag{297}$$

and then to carry out a regular asymptotic analysis of the multiple oscillatory integrals successively. The $\mathrm{sgn}(\xi)$ factors introduce discontinuities which result in contributions on top of those from the saddle points.

Let us treat the case of $C_{x,t}$ in detail. We write

$$C_{2\alpha't,t} = \frac{t}{2i}\iiint \rho(\alpha'+u)\rho_h(\alpha'+v)\,\mathrm{sgn}(\xi)e^{it[(u-\xi/2)^2-(v-\xi/2)^2]}dudvd\xi, \tag{298}$$

and then apply a saddle point approximation to the $u$ and $v$ integrals using (287) to obtain

$$C_{2\alpha't,t} = \frac{t}{2i}\int_{-\infty}^{\infty}d\xi\,\mathrm{sgn}(\xi)\left[\frac{\pi}{|t|}\rho\left(\alpha'+\frac{\xi}{2}\right)\rho_h\left(\alpha'+\frac{\xi}{2}\right)\right.$$

$$\left. + \frac{\pi i}{4t|t|}\left(\rho''\left(\alpha'+\frac{\xi}{2}\right)\rho_h\left(\alpha'+\frac{\xi}{2}\right)-\rho\left(\alpha'+\frac{\xi}{2}\right)\rho_h''\left(\alpha'+\frac{\xi}{2}\right)\right)\right] + \mathcal{O}(t^{-2}). \tag{299}$$

This can be simplified by performing an integration by parts on the $\xi$ integral of the subleading term

$$
\begin{aligned}
C_{2\alpha' t,t} = i\pi \operatorname{sgn}(t) \int_{-\infty}^{\infty} \rho(\xi)\rho_h(\xi)\operatorname{sgn}(\alpha'-\xi)\mathrm{d}\xi \\
+ \frac{\pi}{2|t|}\big(\rho(\alpha')\rho_h'(\alpha') - \rho'(\alpha')\rho_h(\alpha')\big) + \mathcal{O}(t^{-2}).
\end{aligned}
\tag{300}
$$

Similarly one finds for the $D_{x,t}$ term

$$
\begin{aligned}
D_{2\alpha' t,t} = i\pi \operatorname{sgn}(t) \int_{-\infty}^{\infty} \rho(\xi)\rho_h(\xi)\xi \operatorname{sgn}(\alpha'-\xi)\mathrm{d}\xi \\
+ \frac{\pi}{2|t|}\Big[\alpha'\big(\rho(\alpha')\rho_h'(\alpha') - \rho'(\alpha')\rho_h(\alpha')\big) - \rho(\alpha')\rho_h(\alpha')\Big] + \mathcal{O}(t^{-2}).
\end{aligned}
\tag{301}
$$

To deal with the $A_{x,t}$ term we use that in a distributional sense

$$
\frac{1}{(\lambda-\mu)^2} = -\frac{t^2}{2}\int_{-\infty}^{\infty}|\xi|e^{it\xi(\lambda-\mu)}\mathrm{d}\xi,
\tag{302}
$$

and then carry out a similar analysis to obtain

$$
A_{2\alpha' t,t} = -2\pi|t|\int_{-\infty}^{\infty}\rho(\xi)\rho_h(\xi)|\alpha'-\xi|\mathrm{d}\xi + o(t).
\tag{303}
$$

This leaves us with the $B_{x,t}(\lambda)$ term. It is not possible to determine the asymptotics of $B_{x,t}(\lambda)$ at fixed $\lambda$ and then carry out a saddle point approximation of the resulting integral as the asymptotic expression for $B_{x,t}(\lambda)$ becomes singular at the saddle point $\lambda = \alpha'$. The full contribution involving $B_{x,t}(\lambda)$ to the correlation function is [3]

$$
X_{x,t} \equiv \int \mathrm{d}\mu \fint \mathrm{d}\lambda \mathrm{d}u \mathrm{d}v \frac{e^{it(\lambda^2-\mu^2+u^2-v^2)+ix(\mu-\lambda+v-u)}}{(\lambda-u)(v-u)}(\lambda-\mu)^2\rho(\lambda)\rho_h(\mu)\rho(u)\rho_h(v).
\tag{304}
$$

We rewrite this as a six-fold integral

$$
\begin{aligned}
X_{2\alpha' t,t} = -\frac{t^2}{4}\int\cdots\int \rho(\alpha'+\lambda)\rho_h(\alpha'+\mu)\rho(\alpha'+u)\rho_h(\alpha'+v)\operatorname{sgn}(\xi)\operatorname{sgn}(\zeta) \\
\times (\lambda-\mu)^2 e^{it(\lambda^2-\mu^2+u^2-v^2)}e^{i\xi t(v-u)+i\zeta t(\lambda-u)}\mathrm{d}u\mathrm{d}v\mathrm{d}\lambda\mathrm{d}\mu\mathrm{d}\xi\mathrm{d}\zeta,
\end{aligned}
\tag{305}
$$

and then perform saddle point approximations on the $u, v, \lambda, \mu$ integrals. This gives

$$
\begin{aligned}
X_{2\alpha' t,t} = -\frac{\pi^2}{4}\iint \rho\big(\alpha'-\tfrac{\zeta}{2}\big)\rho_h(\alpha')\rho\big(\alpha'+\tfrac{\zeta+\xi}{2}\big)\rho_h\big(\alpha'+\tfrac{\xi}{2}\big)\frac{\zeta^2}{4}\operatorname{sgn}(\xi)\operatorname{sgn}(\zeta) \\
\times e^{-it\frac{\zeta^2}{2}-it\zeta\frac{\xi}{2}}\,\mathrm{d}\xi\mathrm{d}\zeta[1+o(t^0)].
\end{aligned}
\tag{306}
$$

We now carry out the integral over $\xi$, which does not have saddle point and is dominated by the discontinuity of the integrand at $\xi = 0$ using

$$
\int f(\xi)e^{is\xi}\mathrm{d}\xi = -\frac{f(0+)-f(0-)}{is} - \frac{f'(0+)-f'(0-)}{s^2} + \mathcal{O}(s^{-3}),
\tag{307}
$$

---

[3]Here and in what follows we assume that $\rho_h(\mu)$ is a continuous function of $\mu$ that decays to zero at infinity so that the integral exists. The case where $\rho_h(\mu)$ is the actual hole density is then obtained as a limit of the resulting expression.

where $f$ is a function with discontinuities only at zero. This gives

$$X_{2\alpha't,t} = -\frac{\pi^2 \rho_h(\alpha')^2}{4it} \int_{-\infty}^{\infty} |\zeta| \rho\big(\alpha' - \tfrac{\zeta}{2}\big) \rho\big(\alpha' + \tfrac{\zeta}{2}\big) e^{-it\frac{\zeta^2}{2}} d\zeta [1 + o(t^0)]. \tag{308}$$

This last integral also has a saddle point at zero, but with a coefficient that is not differentiable, so that one cannot apply (287). Approximating $\zeta = 0$ in the $\rho$'s at leading order in $t$, one can integrate the remaining terms to obtain

$$X_{2\alpha't,t} = \frac{\pi^2}{2t^2} \rho(\alpha')^2 \rho_h(\alpha')^2 + o(t^{-2}). \tag{309}$$

The contribution involving $B_{x,t}(\mu)$ is given by

$$Y_{x,t} \equiv \int d\lambda \fint d\mu du dv \frac{e^{it(\lambda^2 - \mu^2 + u^2 - v^2) + ix(\mu - \lambda + v - u)}}{(\mu - u)(v - u)} (\lambda - \mu)^2 \rho(\lambda) \rho_h(\mu) \rho(u) \rho_h(v), \tag{310}$$

and can be analyzed in a similar way. We start by rewriting it as a six-fold integral

$$Y_{2\alpha't,t} = -\frac{t^2}{4} \int \cdots \int \rho(\alpha' + \lambda) \rho_h(\alpha' + \mu) \rho(\alpha' + u) \rho_h(\alpha' + v) \, \mathrm{sgn}(\xi) \, \mathrm{sgn}(\zeta)$$
$$\times (\lambda - \mu)^2 e^{it(\lambda^2 - \mu^2 + u^2 - v^2)} e^{i\xi t(v - u) + i\zeta t(\mu - u)} du dv d\lambda d\mu d\xi d\zeta, \tag{311}$$

and then perform saddle-point approximations on the $\lambda, \mu, u, v$ integrals

$$Y_{2\alpha't,t} \approx -\frac{\pi^2}{16} \iint \mathrm{sgn}(\xi) \zeta |\zeta| \rho(\alpha') \rho_h\big(\alpha' + \tfrac{\zeta}{2}\big) \rho\big(\alpha' + \tfrac{\zeta+\xi}{2}\big) \rho_h\big(\alpha' + \tfrac{\xi}{2}\big) e^{-it\xi\frac{\zeta}{2}} d\xi d\zeta$$
$$-\frac{\pi^2 i}{64t} \iint \mathrm{sgn}(\xi) |\zeta| \Big[ \zeta \rho''(\alpha') \rho_h\big(\alpha' + \tfrac{\zeta}{2}\big) \rho\big(\alpha' + \tfrac{\zeta+\xi}{2}\big) \rho_h\big(\alpha' + \tfrac{\xi}{2}\big)$$
$$- 8\rho'(\alpha') \rho_h\big(\alpha' + \tfrac{\zeta}{2}\big) \rho\big(\alpha' + \tfrac{\zeta+\xi}{2}\big) \rho_h\big(\alpha' + \tfrac{\xi}{2}\big) - \zeta \rho(\alpha') \rho_h''\big(\alpha' + \tfrac{\zeta}{2}\big) \rho\big(\alpha' + \tfrac{\zeta+\xi}{2}\big) \rho_h\big(\alpha' + \tfrac{\xi}{2}\big)$$
$$- 8\rho(\alpha') \rho_h'\big(\alpha' + \tfrac{\zeta}{2}\big) \rho\big(\alpha' + \tfrac{\zeta+\xi}{2}\big) \rho_h\big(\alpha' + \tfrac{\xi}{2}\big) + \zeta \rho(\alpha') \rho_h\big(\alpha' + \tfrac{\zeta}{2}\big) \rho''\big(\alpha' + \tfrac{\zeta+\xi}{2}\big) \rho_h\big(\alpha' + \tfrac{\xi}{2}\big)$$
$$- \zeta \rho(\alpha') \rho_h\big(\alpha' + \tfrac{\zeta}{2}\big) \rho\big(\alpha' + \tfrac{\zeta+\xi}{2}\big) \rho_h''\big(\alpha' + \tfrac{\xi}{2}\big) \Big] e^{-it\xi\frac{\zeta}{2}} d\xi d\zeta. \tag{312}$$

We next perform the integral over $\xi$ in the large $t$ limit using (307). After some rearrangements we obtain

$$Y_{2\alpha't,t} = \frac{i\pi^2}{t} \rho(\alpha') \rho_h(\alpha') \int_{-\infty}^{\infty} |\alpha' - \zeta| \rho(\zeta) \rho_h(\zeta) d\zeta$$
$$- \frac{\pi^2}{4t^2} \int_{-\infty}^{\infty} |\alpha' - \zeta| \Big[ \rho''(\alpha') \rho_h(\alpha') \rho(\zeta) \rho_h(\zeta) - \rho(\alpha') \rho_h''(\alpha') \rho(\zeta) \rho_h(\zeta)$$
$$+ \rho(\alpha') \rho_h(\alpha') \rho''(\zeta) \rho_h(\zeta) - \rho(\alpha') \rho_h(\alpha') \rho(\zeta) \rho_h''(\zeta) \Big] d\zeta$$
$$- \frac{\pi^2}{2t^2} [\rho(\alpha') \rho_h'(\alpha') + 2\rho'(\alpha') \rho_h(\alpha')] \int_{-\infty}^{\infty} \mathrm{sgn}(\alpha' - \zeta) \rho(\zeta) \rho_h(\zeta) d\zeta$$
$$- \frac{\pi^2}{2t^2} \rho(\alpha') \rho_h(\alpha') \int_{-\infty}^{\infty} \mathrm{sgn}(\alpha' - \zeta) [\rho'(\zeta) \rho_h(\zeta) + 2\rho(\zeta) \rho_h'(\zeta)] d\zeta + o(t^{-2}). \tag{313}$$

Putting everything together we arrive at (200).

### C.1.4 GHD predictions

The GHD result for the asymptotics of the density-density correlator is [89, 90]

$$\langle \sigma(x,t)\sigma(0,0)\rangle = \int_{-\infty}^{\infty} \delta\big(x - v(\lambda)t\big)\,\rho(\lambda)\big(1 - \vartheta(\lambda)\big)\big[q^{\mathrm{dr}}(\lambda)\big]^2 \, \mathrm{d}\lambda\,, \tag{314}$$

with $\rho, \vartheta$ defined in (27) and (28), and where the other functions are defined as follows:

$$F(\lambda, v) = \frac{1}{\pi}\arctan\!\left(\frac{\lambda - v}{c}\right) + \int_{-\infty}^{\infty} \frac{2c}{c^2 + (\lambda - \lambda')^2}\,\vartheta(\lambda')\,F(\lambda', v)\frac{\mathrm{d}\lambda'}{2\pi}\,,$$

$$q^{\mathrm{dr}}(\lambda) = 1 - \int_{-\infty}^{\infty} q(\lambda')\,\vartheta(\lambda')\,\partial_\lambda F(\lambda', \lambda)\mathrm{d}\lambda'\,,$$

$$v(\lambda) = \frac{e'(\lambda)}{2\pi\big(\rho(\lambda) + \rho_h(\lambda)\big)}\,,$$

$$e'(\lambda) = 2\lambda - \int_{-\infty}^{\infty} 2v\,\vartheta(v)\partial_\lambda F(v, \lambda)\mathrm{d}v\,. \tag{315}$$

These equations can be straightforwardly solved in a $1/c$-expansion up to order $\mathcal{O}(c^{-2})$

$$\rho(\lambda) + \rho_h(\lambda) = \frac{1 + \frac{2D}{c}}{2\pi}\,, \quad \vartheta(\lambda) = \frac{2\pi\rho(\lambda)}{1 + 2D/c}\,, \quad F(\lambda, \alpha) = \frac{\lambda - \alpha}{\pi c} + \frac{2}{\pi c^2}(\delta - D\alpha)\,,$$

$$v(\lambda) = \frac{2\lambda - 4\delta/c}{1 + 2D/c}\,, \quad q^{\mathrm{dr}}(\lambda) = 1 + \frac{2D}{c}\,, \tag{316}$$

where $D$ and $\delta$ are defined in (29) and (89). Substituting (316) back into (314) precisely recovers the leading contribution in (200).

## C.2 Dynamical structure factor

### C.2.1 Behaviour near the thresholds at zero temperature

We start from the simplified expression of the DSF (203) at zero temperature and will assume $q > 0$ for simplicity. We note that when $z \to \pm\infty$ we necessarily have $Z_+ < Z_-$, so that the only possible region that can lead to a divergence of the integral is the region $z$ close to 0. In this region we first set

$$\omega = q'^2 + 2q'Q - \eta\,, \tag{317}$$

with $\eta > 0$ small, and investigate the values taken by $Z_\pm$. We find for $z$ close to zero

$$Z_+(z) = \begin{cases} \frac{-q'^2 + q'^2 z + \eta}{2q'(1-z)} & \text{if } z > 0\,, \\ \frac{-q'^2 - 4q'Q - q'^2 z + \eta}{2q'(1-z)} & \text{if } z < 0\,, \end{cases}$$

$$Z_-(z) = \begin{cases} \frac{-\eta + q'^2 z}{2q'z} & \text{if } z > \frac{\eta}{2q'^2}\,, \\ \frac{-q'^2 - q'^2 z + \eta}{2q'(1-z)} & \text{if } 0 < z < \frac{\eta}{2q'^2}\,, \\ \frac{-q'^2 - 4q'Q + q'^2 z + \eta}{2q'(1-z)} & \text{if } z < 0\,. \end{cases} \tag{318}$$

We observe that for small $z$ we have $Z_- < Z_+$ if and only if $z < \frac{\eta}{2q'^2}$, in which case $Z_+ - Z_- = \frac{q'|z|}{1-z}$. Substituting this expression back into (203), we find that among the contribution proportional to $\mathbf{1}_{Z_- < Z_+}$ only the term $\frac{Z_+ - Z_-}{z}$ is non-integrable when $z \to 0$. However, its divergent part is

exactly cancelled by the term in the third line of (203) proportional to $\mathbf{1}_{\omega_-<\omega<\omega_+}$. All other terms in the first two lines of (203) give a finite $\mathcal{O}(\eta^0)$ contribution because they are integrable for $z \to 0$. This leaves the contribution proportional to $\mathbf{1}_{\omega_-<\omega<\omega_+}$ for $z > \frac{\eta}{2q'^2}$, which leads to a logarithmic singularity in $\eta$. Setting an arbitrary upper limit in the integral since its modification amounts to a $\mathcal{O}(\eta^0)$ correction we have

$$
\begin{aligned}
S^{(2)}(q,\omega) &= \frac{1}{4\pi^2 q' c^2} \int_{\frac{\eta}{2q'^2}}^{1} \left[ -\frac{2q' \min(|q'z|, 2Q)}{z^2} \right] dz + \mathcal{O}(\eta^0) \\
&= \frac{q'}{2\pi^2 c^2} \log|\eta| + \mathcal{O}(\eta^0).
\end{aligned}
\tag{319}
$$

We now turn to singularities above the upper threshold. Taking $\eta > 0$ to be small and setting

$$
\omega = q'^2 + 2q'Q + \eta,
\tag{320}
$$

we find for $z \approx 0$

$$
\begin{aligned}
Z_+(z) &= \begin{cases} \frac{-q'^2 + q'^2 z - \eta}{2q'(1-z)} & \text{if } z > 0, \\ \frac{\eta + q'^2 z}{2q' z} & \text{if } -\frac{\eta}{2q'(q'+2Q)} < z < 0, \\ \frac{-q'^2 - 4q'Q - q'^2 z - \eta}{2q'(1-z)} & \text{if } z < -\frac{\eta}{2q'(q'+2Q)}, \end{cases} \\
Z_-(z) &= \begin{cases} \frac{\eta + q'^2 z}{2q' z} & \text{if } z > 0, \\ \frac{-q'^2 - 4q'Q + q'^2 z - \eta}{2q'(1-z)} & \text{if } z < 0. \end{cases}
\end{aligned}
\tag{321}
$$

We observe that for $z$ close to zero we have $Z_- < Z_+$ if and only if $z < -\frac{\eta}{2q'(q'+2Q)}$, in which case $Z_+ - Z_- = \frac{q'|z|}{1-z}$. Above the threshold we have $\mathbf{1}_{\omega_-<\omega<\omega_+} = 0$ so that the last term in (203) vanishes. Of the remaining terms only the one proportional to $\frac{Z_+ - Z_-}{z^2}$ diverges near $z = 0$, so that

$$
\begin{aligned}
S^{(2)}(q,\omega) &= \frac{1}{4\pi^2 q' c^2} \int_{-1}^{-\frac{\eta}{2q'(q'+2Q)}} 2q'(1-z)^2 \frac{Z_+ - Z_-}{z^2} dz + \mathcal{O}(\eta^0) \\
&= -\frac{q'}{2\pi^2 c^2} \log|\eta| + \mathcal{O}(\eta^0).
\end{aligned}
\tag{322}
$$

The behaviour near the lower threshold is obtained through a similar analysis.

### C.2.2 Behaviour at small $q, \omega$

We start by writing the two particle-hole contribution as

$$
\begin{aligned}
S^{(2)}(q,\omega) &= \frac{8\pi^2}{c^2} \int_{-\infty}^{\infty} \int_{-\infty}^{\infty} \frac{\rho(q_3)\rho_h(q_4)}{z^2} |1-z| \left[ \rho(q_1)\rho_h(q_2) - \rho\left(\frac{\omega'-q'^2}{2q'}\right)\rho_h\left(\frac{\omega'+q'^2}{2q'}\right) \right] dz\, dp \\
&+ \frac{8\pi^2}{c^2} \int_{-\infty}^{\infty} \int_{-\infty}^{\infty} \frac{\frac{2(\lambda-\mu)^2}{\lambda-\bar{\lambda}} - \frac{(\lambda-\mu)^2}{\mu-\bar{\lambda}} + 3\mu - 2\lambda - \bar{\lambda} + \frac{(\lambda-\mu)^2 - |q'(\lambda-\mu)|}{q'+\lambda-\mu}}{(q'+\lambda-\mu)|q'+\lambda-\mu|} \\
&\qquad\qquad \times \rho(\lambda)\rho_h(\mu)\rho(\bar{\lambda})\rho_h(\bar{\mu})\, d\lambda\, d\mu \equiv \frac{8\pi^2}{c^2}(\Psi_1 + \Psi_2),
\end{aligned}
\tag{323}
$$

where $\Psi_{1,2}$ denote the first and second terms respectively.

The integral for $\Psi_1$ with $q, \omega \to 0$ at fixed $\gamma = \frac{\omega'}{2q'}$ is well-defined and finite. In this limit we have $q_3 = q_4 = \gamma + p(1-z)$ and $q_1 = q_2 = \gamma - pz$. Changing variables to $v = \gamma - pz$ and $u = \gamma + p(1-z)$ we have

$$\Psi_1 = \left( \int_{-\infty}^{\infty} \rho(u)\rho_h(u)|u - \gamma| du \right) \left( \int_{-\infty}^{\infty} \frac{\rho(v)\rho_h(v) - \rho(\gamma)\rho_h(\gamma)}{(v - \gamma)^2} dv \right) + o(q'^0). \quad (324)$$

As for $\Psi_2$, we first perform a change of variables from $\mu$ to $v = q + \lambda - \mu$

$$\Psi_2 = \int_{-\infty}^{\infty} \int_{-\infty}^{\infty} \rho(\lambda)\rho_h(q + \lambda - v)\rho(q' + \lambda - v + q'\tfrac{2\gamma - 2\lambda - q'}{2v})\rho_h(q' + \lambda + q'\tfrac{2\gamma - 2\lambda - q'}{2v})$$

$$\times \frac{1}{v|v|}\left[ \frac{2(v - q')^2}{v - q' - q'\frac{2\gamma - 2\lambda - q'}{2v}} + \frac{(v - q')^2}{q'\frac{2\gamma - 2\lambda - q'}{2v}} + 2q' - 2v \right.$$

$$\left. - q'\frac{2\gamma - 2\lambda - q'}{2v} + \frac{(v - q')^2 - |q'(v - q')|}{v} \right] d\lambda dv. \quad (325)$$

We now observe that the four $\rho$ factors are invariant under the change of variable

$$v' = -q'\frac{2\gamma - 2\lambda - q'}{2v}. \quad (326)$$

We apply this change of variable to all the terms except

$$\frac{(v - q')^2}{q'\frac{2\gamma - 2\lambda - q'}{2v}}, \quad (327)$$

and express the term

$$\frac{2v^2}{v - q' - q'\frac{2\gamma - 2\lambda - q'}{2v}}, \quad (328)$$

as one half of itself plus one half of itself after the change of variables. We obtain for $q' > 0$

$$\Psi_2 = -2 \int_{-\infty}^{\infty} \int_{-\infty}^{\infty} \rho(\lambda)\rho_h(q' + \lambda - v)\rho\left(q' + \lambda - v + q'\tfrac{2\gamma - 2\lambda - q'}{2v}\right)\rho_h\left(q' + \lambda + q'\tfrac{2\gamma - 2\lambda - q'}{2v}\right)$$

$$\times \frac{\operatorname{sgn}(v)}{2\gamma - 2\lambda - q'}\left[ 1 - \frac{q'}{v} + \frac{2v^2}{2v^2 - 2q'(v + \gamma - \lambda) + q'^2} \right. \quad (329)$$

$$\left. - \frac{2v}{2\gamma - 2\lambda - q'}\left( 1 - \left| 1 + \frac{2\gamma - 2\lambda - q'}{2v} \right| \right) \right] d\lambda dv. \quad (330)$$

Since there are no non-integrable divergences in the integrand at small $v$, in this representation one can set $q' = 0$ in the $\rho$ terms as well as in the integrand, at small $q'$. It yields

$$\Psi_2 = -\int_{-\infty}^{\infty} \int_{-\infty}^{\infty} \rho(\lambda)\rho_h(\lambda)\rho(u)\rho_h(u)\frac{\operatorname{sgn}(\lambda - u)}{\gamma - \lambda}\left[ 2 - \frac{\lambda - u}{\gamma - \lambda}\left( 1 - \left| \frac{\gamma - u}{\lambda - u} \right| \right) \right] d\lambda du + o(q^0). \quad (331)$$

We obtain then the claimed result.

### C.2.3 High frequency tail

We start with the representation (184) for the two particle-hole contribution to the DSF expressed as a single double integral. We first decompose the double integral into the two regions $|q' + \lambda - \mu| > \epsilon$ and $|q' + \lambda - \mu| < \epsilon$ and focus on the latter part. Since we have assumed that

$\rho$ decays faster than any power law at infinity $\lambda$ has to remain smaller than any power law of $\omega$ for the integral not to vanish at any order $\mathcal{O}(\omega^{-n})$. Since $|q' + \lambda - \mu| < \epsilon$ the same holds true for $\mu$. But this implies that $\overline{\lambda}$ necessarily grows as $\omega^{1/2}$, which makes this contribution vanish at any order $\mathcal{O}(\omega^{-n})$. Hence at any given order $\mathcal{O}(\omega^{-n})$ we can impose $|q' + \lambda - \mu| > \epsilon$. This removes all poles in the integrand of (184) and one can consider all contributions separately. Moreover, since $\rho$ decays faster than a power law at infinity the term proportional to $\rho(\frac{\omega' - q'^2}{2q'})\rho_h(\frac{\omega' + q'^2}{2q'})$ is negligible at order $\mathcal{O}(\omega^{-n})$.

We then split the $\mu$ integral into the sum of positive and negative $\mu$ parts and perform the change of variables

$$z = \begin{cases} 2\mu - q' - \lambda - \sqrt{2\omega'} & \text{if } \mu > 0 \,, \\ 2\mu - q' - \lambda + \sqrt{2\omega'} & \text{if } \mu < 0 \,. \end{cases} \tag{332}$$

This way the DSF can be brought to the form

$$
\begin{aligned}
S^{(2)}(q,\omega) = &\int_{-\infty}^{\infty} d\lambda \int_{-q'-\lambda-\sqrt{2\omega'}}^{\infty} dz \Big[ \rho(\lambda)\rho\big(z + f_+(z,\lambda,\omega')\big) g_+(z,\lambda,\omega') \\
&\times \rho_h\big(\tfrac{z+q'+\lambda+\sqrt{2\omega'}}{2}\big)\rho_h\big(z + f_+(z,\lambda,\omega') + \tfrac{q'+\lambda-z-\sqrt{2\omega'}}{2}\big) \Big] \\
+ &\int_{-\infty}^{\infty} d\lambda \int_{-\infty}^{-q'-\lambda+\sqrt{2\omega'}} dz \Big[ \rho(\lambda)\rho\big(z + f_-(z,\lambda,\omega')\big) g_-(z,\lambda,\omega') \\
&\times \rho_h\big(\tfrac{z+q'+\lambda-\sqrt{2\omega'}}{2}\big)\rho_h\big(z + f_-(z,\lambda,\omega') + \tfrac{q'+\lambda-z+\sqrt{2\omega'}}{2}\big) \Big] + \dots \,, \quad (333)
\end{aligned}
$$

where the dots indicate subleading corrections that decay faster than any inverse power in $\omega$ and

$$
\begin{aligned}
f_\pm(z,\lambda,\omega) = &\pm \frac{1}{2} \frac{q'^2 + 2q'z + 2q'\lambda - (\lambda - z)^2}{\sqrt{2\omega} \pm (z - q' - \lambda)} \,, \\
g_\pm(z,\lambda,\omega) = &\frac{16\pi^2}{c^2(\lambda + q' - z \mp \sqrt{2\omega})|\lambda + q' - z \mp \sqrt{2\omega}|} \Big[ -4\lambda + \frac{(\lambda - q' - z \mp \sqrt{2\omega})^2}{\lambda + q' - z \mp \sqrt{2\omega}} \\
&+ 2\frac{(\lambda + q' - z \mp \sqrt{2\omega})(\lambda - q' - z \mp \sqrt{2\omega})^2}{4\omega - 8q'\lambda - 4q'^2} + 3(\lambda + q' + z + \sqrt{2\omega}) \\
&+ \frac{-\lambda^2 + 2\lambda q' + q'^2 + z^2 \pm 2\sqrt{2\omega}z}{\lambda + q' - z - \sqrt{2\omega}} \\
&+ 2\frac{(\lambda + q' - z \mp \sqrt{2\omega})(\lambda - q' - z \mp \sqrt{2\omega})^2}{\lambda^2 + q'^2 \pm 2\sqrt{2\omega}z + z^2 + 4\lambda q' - 2\lambda z \mp 2\lambda\sqrt{2\omega}} \Big].
\end{aligned} \tag{334}
$$

We now observe that any part of the integral where the argument of one of the two $\rho$'s grows as a power-law in $\omega$ will give contributions that decay faster than any power-law, since $\rho$ is assumed to decay faster than any power-law at infinity. From the expression of $f_\pm$ one sees that $z$ cannot grow faster than $\omega^{1/4}$. Consequently, with an error that goes to zero faster than any power law in $\omega$ one can replace the limits of the integrals and the arguments of the $\rho_h$'s by $\pm\infty$. This gives

$$
\begin{aligned}
S^{(2)}(q,\omega) = &\frac{1}{4\pi^2} \int_{-\infty}^{\infty} \int_{-\infty}^{\infty} \rho(\lambda)\rho\big(z + f_+(z,\lambda,\omega')\big) g_+(z,\lambda,\omega') dz d\lambda \\
+ &\frac{1}{4\pi^2} \int_{-\infty}^{\infty} \int_{-\infty}^{\infty} \rho(\lambda)\rho\big(z + f_-(z,\lambda,\omega')\big) g_-(z,\lambda,\omega') dz d\lambda + \dots \quad (335)
\end{aligned}
$$

We now expand $f_{\pm}(z, \lambda, \omega')$ and $g_{\pm}(z, \lambda, \omega')$ in Laurent series in $\lambda$, $\lambda - z$ and $\omega'^{-1/2}$, and Taylor expand $\rho(z + f_{\pm}(z, \lambda, \omega'))$. This produces terms of the type $\rho(\lambda)\rho^{(a)}(z)\lambda^b(\lambda - z)^d \omega'^{-e/2}$ with $a, b, e \geq 0$ integers and $d$ a positive or negative integer. We integrate this by parts $a$ times over $z$ so that the integrand involves only $\rho(z)$, and then write the full result $S^{(2)}(q, \omega)$ as one half of itself plus one half of itself after swapping the dummy variables $\lambda$ and $z$. We observe that there remain only positive powers $d \geq 0$, and one obtains the first two terms of the expansion $\omega'^{-7/2}, \omega'^{-9/2}$ stated in the text.

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
