# Peer review of "A systematic $1/c$-expansion of form factor sums for dynamical correlations in the Lieb-Liniger model"

_SciPost Physics, doi:SciPost Phys. 9, 082 (2020)_

## Round 1 · Referee Report · Anonymous (Referee 1) · 2020-9-24

Strengths

  1. Timely and difficult subject.
  2. Nice introduction to the problem.
  3. Systematic approach to the computations.
  4. Rather complete comparison with known results.

Weaknesses

  1. By design, the results are only perturbative.

Report

In this work authors present perturbative computations of the dynamic density-density correlation function in the Lieb-Liniger model. The results give access to correlation functions in the ground state, at finite temperatures but also in non-equilibrium states such as those emerging long-time after a local quench, in the thermodynamically large system. The results are important for both experimental studies of cold-atomic systems, for the understanding of strongly correlated models and non-equilibrium quantum physics. The technique employed involves Lehmann representation of the correlation function with energy levels and form-factors known from the Bethe ansatz methods. As the analytic evaluation of the spectral sum is incomprehensible the authors use the large coupling expansion.

In the large coupling expansion the correlation function is organized in contributions coming from excited states containing different numbers of particle- hole excitations. The authors present computations up to order $1/c^2$ which requires summation over one and two particle-hole excited states. Showing how to perform such summations in the thermodynamic limit is the main technical achievement of this work.

The problem of the computation of dynamic correlation functions in integrable models is long-standing and the present paper constitutes an important step in this direction. This is specifically by clarifying the concept of the representative state and by showing that the thermodynamic limit of the correlation function depends only on the rapidities distribution and not on more detailed microscopic data.

The paper is very well written, the exposition is clear and detailed. It's worth mentioning that authors present in-depth comparison of their computations with known results. I suggest the paper to be published in SciPost Physics after the authors address the minor issues from "Requested changes".

Requested changes

  1. On pg. 35, the authors explain that eq. (163) vanishes. I would like to clarify if this is simply because the sum is over $|\lambda_j| > \Lambda$ or is the argument more involved?

  2. On pg. 36, the authors interpret part of the contribution from two particle-hole excitations as a dressing of contributions from one particle-hole excitations. Would it be possible to do this at the level of form-factors, namely by defining a dressed one particle-hole form factor?

  3. Fig. 6 and Fig. 7, the authors should comment why one particle-hole contribution to the correlation function is larger than the whole correlator. Is this because of including some part of the two particle-hole contribution to the one particle-hole contribution? In any case, this deserves an explanation.

  4. The authors should mention that constants $C_0, C_1, C_3, C_4$ are known from [34] and their result in eq. (210) can be verified. Also in that respect, and in the context of $A_1$ prefactor, ref. https://arxiv.org/abs/1103.4176 should be mentioned.

Some small things:

  • extra apostrophe in the title of section 5.1.8.
  • double "as" at the bottom of pg. 32.
  • wrong spacing after eq. (224).

---

## Round 1 · Referee Report · Anonymous (Referee 2) · 2020-11-1

Strengths

1) Addresses the longstanding problem of calculating dynamical correlation functions of integrable models at finite temperature and out off equilibrium

2) Develops a systematic method to evaluate singular sums of over Bethe roots

3) Good and clear presentation of the context, the physical picture and the technical aspects

Report

The authors contribute to the longstanding problem of the evaluation of dynamical correlation functions of integrable models at finite temperature and out off equilibrium. Contrary to many previous results in the literature they present valid and explicit formulae that can be eventually evaluated numerically. They consider the Lieb-Liniger model in the strong coupling regime. Starting from a form factor expansion of the two-point density correlation function they develop a perturbation theory simultaneously in the inverse coupling 1/c and in the number of particles involved in the "excitation over a given state". They calculate the expansion up to second order. Comparing with known asymptotic results in certain limits they advocate the hypothesis that their expansion is uniform in space and time.

On a technical level the authors develop a technique to deal with sums over Bethe roots that become singular in the thermodynamic limit and that are no longer linear functionals in the Bethe root densities.

All results appear rather plausible to me. The paper is well written. It has a clear outline and provides sufficient details of all calculations.

I recommend publication in the present form.

Here are two minor optional comments. 1) I did not understand the meaning of the function $n(\lambda)$ on the line between equations (58) and (59). 2) In section 7.1.1, where the authors consider the asymptotics of the correlations function at equal times, they refer to paper [33], which appeared 2012, for the nontrivial amplitude $A_1$. An explicit formula for this amplitude had appeared before in the work "Algebraic Bethe ansatz approach to the asymptotic behavior of correlation functions" by N. Kitanine et al in JSTAT 04, 2009 (2009) P04003. I would suggest to mention this work as well. Moreover, I would expect the analysis of the large-$c$ behaviour to be straightforward, as I would expect the series representing the Fredholm determinant in the general answer to terminate after a few terms if it is expanded in $1/c$.

---

## Round 1 · Referee Report · Anonymous (Referee 3) · 2020-11-10

Strengths

  • first exact and unambiguous results on dynamical correlations of finite-temperature states in interacting integrable systems.

Report

I have no comments, the paper is excellent and ready to be published.

---

## Round 2 · Referee Report · Anonymous (Referee 1) · 2020-11-13

Report

I'm satisfied with the new version of the manuscript and consider the paper ready to be published in its present form.

---

## Round 2 · Referee Report · Anonymous (Referee 2) · 2020-11-15

Report

As I said in my report, I was already satisfied with the first version of this manuscript. It has been further improved by taking into account the tiny suggestions of the referees and is ready for publication in its present form.

---

## Round 2 · Author Response

We are grateful to the three referees for their careful reading of the manuscript and helpful comments. We appreciate their very positive reports. In the following we answer the minor points raised by the first and second referees.

— Reply to the first referee —

1- We added the argument mentioned by the referee in paragraph 5.3.1.

2- The referee asks whether the dressing of one-particle-hole excitations by two-particle- hole excitations could be done directly at the level of the form factor. This is indeed a very interesting question. However, this is a rather subtle point that goes beyond the scope of the paper. We are working on higher-order corrections in 1/c and should be able to address this question in future work.

3- The referee asks why some terms contribute negatively to the dynamical structure factor. This is indeed a relevant remark. We added a paragraph at the end of 6.2 to explain this feature. As noted by the referee, this comes from the fact that what we call one and two-particle-hole excitations are terms that result from cross-cancellations of divergent parts in their bare spectral sum, and these resulting pieces need not be positive. The association of the resulting terms with one or two-particle-hole excitations is based on whether they are expressed as double or quadruple integrals over the root densities. As we further note in the added paragraph, negative spectral weights are expected to arise in individual terms of any perturbative expansions of the Lehmann representation.

4- We added the reference mentioned by the referee in Section 7.2.2.

— Reply to the second referee —

1- This n(λ) was a typo and has been changed into θ(λ).

2- We added the reference mentioned by the referee in Section 7.1.1.

---

## Editorial Decision

published